# MD-LSM: An Efficient Tool for Real-time Monitoring Linear Separability of Hidden-layer Outputs of Deep Networks

## Abstract

Many studies have shown that evaluating the linear separability of hidden-layer outputs plays a key role in understanding the working mechanism of deep networks. However, it is still challenging to develop the linear separability measure (LSM) that satisfies all of the following requirements: 1) it should be an absolute measure; 2) it should be insensitive to the outliers; and 3) its computational cost should be low for real-time monitoring the behavior of each hidden layer. In this paper, we propose the Minkowski difference-based linear separability measures (MD-LSMs) that just meet the first two requirements. Moreover, we also introduce an approximate calculation method to significantly decrease their computation costs with only a slight precision sacrifice. As an application example, we conduct the experiments on the real-time monitoring for the hidden-layer behaviors of several popular deep networks, and show that the outputs of the hidden layers adjacent to the output layer have higher linear separability degrees. We also observe that the change of linear separability degree of hidden layers (especially the ones are adjacent to the output layers) are in sync with the change of the training accuracy of the entire network. It implies that the linear separability of some important hidden layers can be treated as a performance criterion to characterize the network's training behavior. The relevant theoretical discussion also validates this finding.

## 1 Introduction

Two point sets are said to be linearly separable if they can be correctly separated by a hyperplane. The concept of linear separability plays an important role in measuring the capability of neural networks (Tajine & Elizondo, 2002; Elizondo, 2004; Elizondo et al., 2010). In the literature, there are two main research issues on the linear separability of a neural network, which trended to be considered as an entire function: 1) whether the current network can achieve all dichotomies, *i.e.,* the mapping capability (Hornik et al., 1989); and 2) how many dichotomies can be recorded by a network with the specific structure, *i.e.,* the memory capability (Cover, 1965). Since neural networks are of multiple-hidden layer stacking structures, the network outputs are produced by the composition of multiple pseudo-linear maps, each of which corresponds to one hidden layer (Vershynin, 2020). It could be hard to infer the working mechanism of a deep network by treating it as an entire function rather than by analyzing the behavior of each hidden layer.

Consider a feed-forward network $\text{net}(\cdot) : \mathbb{R}^N \to \{0, 1\}$ with $N$ input node and $L$ hidden layers. Denote the $l$-th hidden layer as $\text{hid}_l(\cdot)$ and let $\text{hid}_l(\mathcal{X})$ be the set of hidden-layer outputs w.r.t. the input set $\mathcal{X} := \{\mathbf{x}_m\}_{m=1}^M$. Set $\mathbf{V}_l$ as the weights of the $l$-th hidden layer ($1 \le l \le L$), and let $\mathbf{w}$ be the weights of the output layer. Denote $\mathbf{V}_l'$ ($1 \le l \le L$) and $\mathbf{w}'$ as the updated weights provided by a training algorithm implemented on the training set $\mathcal{S} = \{(\mathbf{x}_m, \mathbf{y}_m)\}_{m=1}^M$. The updated network is denoted as $\text{net}'(\cdot)$ with updated weights $\mathbf{V}_1', \cdots, \mathbf{V}_L'$ and $\mathbf{w}'$. Denote $\text{hid}_l'(\cdot)$ as the $l$-th hidden layer with the updated weights $\mathbf{V}_l'$ ($1 \le l \le L$). Under these notations, we obtain the following theorem which motivates the research of this paper:

**Proposition 1.1** (Synchronicity). *Assume that the updated weights $\mathbf{w}'$ achieves the highest classification accuracy on $\mathcal{S}$ when the hidden-layer weights of $\text{net}'(\cdot)$ are updated to be $\mathbf{V}_1', \cdots, \mathbf{V}_L'$. Then, $\text{net}'(\cdot)$ has higher classification accuracy on $\mathcal{S}$ than $\text{net}(\cdot)$ if and only if the linear separability degree of $\text{hid}_L'(\mathcal{X})$ is larger than that of $\text{hid}_L(\mathcal{X})$.*

This proposition demonstrates that there exists the synchronicity between the linear separability degree of the $L$-th hidden-layer outputs and the training performance during the process of training a network, *i.e.,* the change of linear separability degree of the $L$-th hidden-layer outputs is in sync with that of the training accuracy. Although this result cannot explicitly exhibit the relationship between the linear separability degree of the $l$-th hidden-layer outputs ($l < L$) and the training accuracy, the $L$-th hidden-layer outputs actually are determined by the weights $\mathbf{V}'_1, \cdots, \mathbf{V}'_{L-1}$, which also influence the linear separability degree of the relevant hidden-layer outputs. Therefore, the linear separability can be applied to analyze the mapping behaviors of hidden layers and then to understand the working mechanism of deep networks.

In recent years, some pioneering works have been aware of the importance of analyzing deep networks via the layer-wise changes of class separability when they pass through the networks (Schilling et al., 2021; Apicella et al., 2024; Pezzotti et al., 2017; Rauber et al., 2016; Alain & Bengio, 2016; Ben-Shaul & Dekel, 2022; He & Su, 2023; Rangamani et al., 2023). One common opinion of these works is that the linear separability degree of hidden-layer outputs should become layer-wisely stronger if a deep network has been (or is being) trained suitably. Some empirical evidences were also provided to demonstrate this fact. Therefore, the linear separability provides a feasible manner to analyze the mapping behaviors of hidden layers and then to understand the working mechanism of deep networks. Accordingly, a desired linear separability measure (LSM) should meet the following requirements:

(1) **(Efficiency)** It should have a low computational cost, because we would like to layer-wisely examine the linear separability degree of the hidden-layer outputs after each weight-update epoch during the entire training process;

(2) **(Robustness)** It should be insensitive to the outliers, because the stochastic gradient descent methods sometimes cause abnormal hidden-layer outputs;

(3) **(Absoluteness)** It should be an absolute measure, which objectively evaluates the degree of linear separability between two sets. If its value is known, one can directly judge whether two sets are linearly separable or how heavy they are overlapped. In contrast, the relative measure only indicates whether the linear separability between the two current sets becomes stronger (or weaker) than that of the two sets before being transformed. Therefore, it is hard to describe the linear separability of two sets based only on its value. Please refer to Remark 2.8 for an illustration.

## 1.1 BACKGROUND AND RELATED WORKS

Some mathematical terms appearing in the existing works actually can be treated as LSMs of two point sets, *e.g.*, the generalized Rayleigh quotient (GRQ) of linear discriminant analysis (LDA), which is a relative measure:

$$J_{\boldsymbol{\omega}} := \max_{\boldsymbol{\omega}} \left\{ (\boldsymbol{\omega}^T \mathbf{S}_b \boldsymbol{\omega})/(\boldsymbol{\omega}^T \mathbf{S}_w \boldsymbol{\omega}) \right\}, \tag{1}$$

and the sum of slack variables (SSV), which is an absolute measure, in linear support vector machine (L-SVM) with soft margin. Moreover, since $J_{\boldsymbol{\omega}}$ is based on the mean of the point set, it is sensitive to the outliers in the set. Because of the eigenvalue decomposition, the calculation of $J_{\boldsymbol{\omega}}$ could be time-consuming especially when the dimension is high. He & Su (2023) introduced the term $\mathrm{tr}(\mathbf{S}_w \mathbf{S}_b^{\dagger})$, a variant of GRQ, to measure the linear separability degree of hidden-layer outputs, but it is a relative measure and the calculation of the Moore-Penrose inverse $\mathbf{S}_b^{\dagger}$ is time-consuming as well. Similarly, since L-SVM is expressed as a quadratic programming problem, the calculation of SSV sometimes brings a high computational burden when the sample size is large.

Additionally, Ben-Israel & Levin (2006) introduced the linear divisible angle to measure the linear separability degree of two point sets, where the labels of the data are treated as a new attribute to convert the dimension of points from $N$ to $N + 1$, and then LDA is used to compute the GRQ of the converted points. Gabidullina (2013) adopted the smallest thickness of the classified hyperplane as the LSM for the linearly inseparable sets. Since this measure is computed via a minimax optimization problem, its computational cost is high.

By incorporating the intra-class and the inter-class distances, Schilling et al. (2021) introduced the generalized discrimination value (GDV) to measure the class separability among the hidden-layer outputs associated with different labels during the training process. By tracking the behavior of MLP's hidden layers in each training epoch, they detected the synchronicity between the class separability,

measured by GDV, of hidden-layer outputs and the training performance. Since the computation of GDV is time-consuming, for the relatively complicated networks (such as CNN, ResNet, VGG and Inception), they only computed the GDVs of hidden-layer outputs of the trained networks, and then made some statistical analysis between the resultant GDVs and the layer number in order to explore the alteration rule of the class separability of the outputs from different hidden layers.

Apicella et al. (2024) introduced a structural manner to detect the behavior of hidden-layer outputs during the training phase. Specifically, they imposed an auxiliary output layer, called hidden classification layer, into each hidden layer and then combined the loss function of each auxiliary output layer with the loss of the main network to form an entire training objective function. Their experiments have shown some interesting phenomena: 1) the introduction of hidden classification layer can enhance the class separability, measured by GDV, of the corresponding hidden-layer outputs; and 2) when the class separability of each hidden layer increases, the main network gains a higher testing performance. However, it is still challenging to explain them, which also motivates this paper.

Alain & Bengio (2016) used hidden-layer outputs to train a linear classifier, called "probe", and its classification performance is regarded as a measure of the linear separability degree of the hidden layer. Some state-of-the-art classifiers (such as logistic regression or naive Bayes) have the potential to act as feasible "probes" because of their low desired computational complexities. However, if there is no priori knowledge on the data distribution, the efficiency and the performance of these classifiers could be heavily influenced by some unavoidable factors such as the choice of hyperparameters and the setting of termination conditions, and thus their desired complexities are usually hard to be achieved in practice. Consequently, the "probe" method is unsuitable (at least cannot be directly applied) to detecting the mapping behavior of each hidden layer after each training epoch. How to develop an efficient tool for real-time monitoring the status of each hidden layer during the entire training process becomes the main research concern of this paper.

In addition, there are also other works applying the concept of linear separability to study the properties of deep networks, such as the fold of the data manifold in the high-dimensional space via hidden layers of deep networks (Keup & Helias, 2022), and the trade-offs between the representation ability and the depth-size of deep neural networks with rectified linear units (ReLUs) (Arora et al., 2016).

Table 1: Comparison among Different LSMs

| LSM | Efficiency | Robustness | Absoluteness | Reference |
|---|---|---|---|---|
| GRQ | $\times$ | $\times$ | $\times$ | Fisher (1936) |
| $\mathrm{tr}(\mathbf{S}_w\mathbf{S}_b^{\dagger})$ | $\times$ | $\times$ | $\times$ | He & Su (2023) |
| SSV | $\times$ | $\checkmark$ | $\checkmark$ | Cortes & Vapnik (1995) |
| Linear divisible angle | $\times$ | $\times$ | $\times$ | Ben-Israel & Levin (2006) |
| Smallest thickness | $\times$ | $\checkmark$ | $\checkmark$ | Gabidullina (2013) |
| GDV | $\times$ | $\times$ | $\times$ | Schilling et al. (2021) |
| Structural manner | $\times$ | $\checkmark$ | $\times$ | Apicella et al. (2024) |
| "Probe" | $\times$ | $\checkmark$ | $\checkmark$ | Alain & Bengio (2016) |
| $\mathrm{LS}_i$ ($i \in \{*, 0, 1\}$) | $\times$ | $\checkmark$ | $\checkmark$ | Ours |
| $\widehat{\mathrm{LS}}_i$ ($i \in \{*, 0, 1\}$) | $\checkmark$ | $\checkmark$ | $\checkmark$ | Ours |

## 1.2 OVERVIEW OF MAIN RESULTS

In this paper, we mainly concern with two issues: one is how to develop the LSM that satisfies the aforementioned requirements of efficiency, robustness and absoluteness; and the other is what behaviors can be captured via the real-time monitoring for the linear separability of hidden-layer outputs.

First, we introduce Minkowski difference-based LSM (MD-LSM) for evaluating the linear separability degree of hidden-layer outputs. They are absolute measures and insensitive to the outliers. Since their original forms are hard to calculate, we then design an efficient approximation manner whose computational cost is low. The comparative experiments are conducted to demonstrate that the values of the original MD-LSMs slightly differ from those provided by the approximation manner. In Table 1, we make the comparison between the existing LSMs and the proposed MD-LSMs from the viewpoint of whether they meet the requirements of efficiency, robustness and absoluteness.

As an application example, we use the proposed MD-LSMs for real-time monitoring the hidden-layer behaviors of some popular deep networks during their entire training processes, including multilayer perceptron (MLP) (Bishop, 1994), graph neural network (GNN) (Kipf & Welling, 2016), convolutional neural network (CNN) (LeCun et al., 1998), ResNet (He et al., 2016), VGGNet (Simonyan & Zisserman, 2014), AlexNet (Krizhevsky et al., 2017), vision transformer (ViT) (Doso-vitskiy et al., 2020) and GoogLeNet (Szegedy et al., 2015). We verify the effectiveness of MD-LSMs on several real-world datasets including the UCI datasets and the text datasets. We calculate the linear separability of each hidden layer after each training epoch, and observe that when the training sample set passes through a deep network, the outputs of the hidden layers that are closer to the output layer have higher linear separability degrees. This finding not only accords with the common opinion of recent works but also empirically answers the question of why deep networks need a large number of hidden layers. Since one finite-width hidden layer equipped with the usual activation functions (*e.g.,* sigmoid, tanh and ReLU) has limited nonlinear mapping capability, the linear separability degree of its outputs could be slightly higher than that of its inputs. For complicated classification tasks, the composition of multiple hidden layers gradually increases the linear separability degree of outputs of each hidden layer. In this manner, the desired classification accuracy can be achieved.

Moreover, we also find that the changes of linear separability of hidden layers (especially the ones adjacent to the output layer) are in sync with the changes of the training accuracy. This finding implies that detecting linear separability of some important hidden layers potentially becomes a reasonable manner of characterizing the real-time training behavior of the entire network. A relevant theoretical discussion is given to validate this finding as well.

The rest of this paper is organized as follows. Section 2 defines MD-LSMs and then gives the empirical comparison with several representative LSMs. In Section 3, we conduct the numerical experiments on the real-time monitoring for hidden-layer behavior of some popular deep networks. The last section concludes this paper. In the appendix, we give the workflow of finding maximum linearly-separable subsets (part A). We then make the comparisons with GRQ and GDV (parts B & C). Next, we prove the main results (part D). Finally, we present the complete report of the experiments on the real-time monitoring (part E).

## 2 MINKOWSKI DIFFERENCE BASED LINEAR SEPARABILITY MEASURES

In this section, we present the concept of MD-LSM and its alternative versions. Then, we design an approximate manner to calculate them with a low computational cost.

### 2.1 MINKOWSKI DIFFERENCE AND MAXIMUM LINEARLY SEPARABLE SUBSET

The concept of Minkowski difference (MD) has been widely used in many fields such as data classification (Mampaey et al., 2012; Takeda et al., 2013) and collision detection (Ericson, 2004).

**Definition 2.1** (Minkowski Difference). *Let $\mathcal{A} = \{\mathbf{a}_1, \cdots, \mathbf{a}_I\} \subset \mathbb{R}^N$ and $\mathcal{B} = \{\mathbf{b}_1, \cdots, \mathbf{b}_J\} \subset \mathbb{R}^N$ be two point sets. Then, the Minkowski difference between them is defined as*

$$\mathrm{MD}(\mathcal{A}, \mathcal{B}) := \big\{ \mathbf{m}_{ij} := \mathbf{a}_i - \mathbf{b}_j \in \mathbb{R}^N \mid \mathbf{a}_i \in \mathcal{A}, \ \mathbf{b}_j \in \mathcal{B} \big\}.$$

Based on Minkowski difference, we convert the linear separability of two point sets into the relative position relationship between a point set and a hyperplane that passes the origin (*cf.* Fig. 1).

**Theorem 2.2.** *Two points sets $\mathcal{A}, \mathcal{B} \subset \mathbb{R}^N$ are linearly separable if and only if there exists a vector $\boldsymbol{\omega} \in \mathbb{R}^N$ such that all points of $\mathrm{MD}(\mathcal{A}, \mathcal{B})$ locate in one side of the hyperplane $\boldsymbol{\omega}^T \mathbf{m} = 0$, $\mathbf{m} \in \mathbb{R}^N$.*

As shown in the proof of this theorem (*cf.* Appendix D.1), given two linearly separable sets $\mathcal{A}$ and $\mathcal{B}$, the normal vector $\boldsymbol{\omega}$ of any hyperplane that separates the two sets is the one mentioned in the theorem. Additionally, if the two sets $\mathcal{A}$ and $\mathcal{B}$ are linearly inseparable, some points of $\mathrm{MD}(\mathcal{A}, \mathcal{B})$ will lie in one side of the hyperplane and the rest lie in the other side (*cf.* Fig. 1):

**Definition 2.3** (Minor and Major Sides). *Given a hyperplane $\boldsymbol{\omega}^T \mathbf{m} = 0$, if more than half points of $\mathrm{MD}(\mathcal{A}, \mathcal{B})$ lie in one side of $\boldsymbol{\omega}^T \mathbf{m} = 0$, then this side is said to be the major side of the hyperplane; and accordingly, the other side of $\boldsymbol{\omega}^T \mathbf{m} = 0$ is said to be the minor side of the hyperplane.*

Furthermore, we denote $\mathrm{major}_{\boldsymbol{\omega}}(\mathrm{MD}(\mathcal{A}, \mathcal{B}))$ (resp. $\mathrm{minor}_{\boldsymbol{\omega}}(\mathrm{MD}(\mathcal{A}, \mathcal{B}))$) as the subset of $\mathrm{MD}(\mathcal{A}, \mathcal{B})$ that lies in the major (resp. minor) side of $\boldsymbol{\omega}^T \mathbf{m} = 0$ (*cf.* Fig. 1). According to Theorem 2.2, the points $\mathbf{m}_{ij} \in \mathrm{minor}_{\boldsymbol{\omega}}(\mathrm{MD}(\mathcal{A}, \mathcal{B}))$ can be eliminated by removing the relevant points $\mathbf{a}_i$ from $\mathcal{A}$ or $\mathbf{b}_j$ from $\mathcal{B}$, and the rests turn out to be linearly separable.

**Definition 2.4.** *The set $\mathrm{MaxLS}_{\boldsymbol{\omega}}(\mathcal{A}, \mathcal{B})$ is said to be the maximum linearly-separable subset of $\mathcal{A} \cup \mathcal{B}$ w.r.t. the vector $\boldsymbol{\omega}$, if it holds that $\mathrm{MaxLS}_{\boldsymbol{\omega}}(\mathcal{A}, \mathcal{B}) := \mathcal{A}_{\boldsymbol{\omega}} \cup \mathcal{B}_{\boldsymbol{\omega}} = \arg\max_{\mathcal{A}' \subset \mathcal{A}, \mathcal{B}' \subset \mathcal{B}} |\mathcal{A}'| + |\mathcal{B}'|$ such that $\mathcal{A}'$ and $\mathcal{B}'$ can be linearly separated by using the hyperplane with the normal vector $\boldsymbol{\omega}$.*

Namely, $\mathrm{MaxLS}_{\boldsymbol{\omega}}(\mathcal{A}, \mathcal{B})$ is the largest-size subset of $\mathcal{A} \cup \mathcal{B}$ such that $\mathcal{A}_{\boldsymbol{\omega}}$ and $\mathcal{B}_{\boldsymbol{\omega}}$ are linear separable w.r.t. the hyperplane with the normal vector $\boldsymbol{\omega}$. It is noteworthy that $\mathrm{MaxLS}_{\boldsymbol{\omega}}(\mathcal{A}, \mathcal{B})$ could not be unique. The workflow of finding $\mathrm{MaxLS}_{\boldsymbol{\omega}}(\mathcal{A}, \mathcal{B})$ is given in Appendix A.

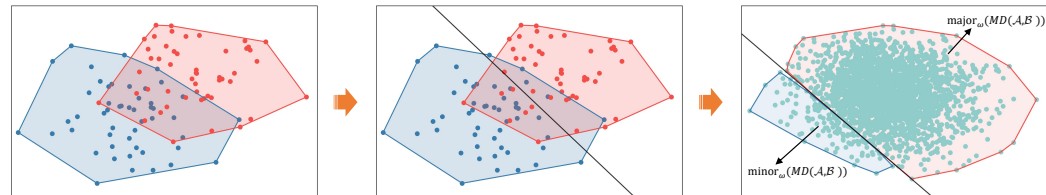

Figure 1: Minor and major sides of the Minkowski difference for two overlapped sets

## 2.2 MD-BASED LINEAR SEPARABILITY MEASURE (MD-LSM)

Following Theorem 2.2, the ratio of the numbers of the points $\mathbf{m}_{ij} \in \mathrm{MD}(\mathcal{A}, \mathcal{B})$ that respectively locate in the two sides of the hyperplane can be treated as a criterion to measure the linear separability degree between $\mathcal{A}$ and $\mathcal{B}$:

$$\mathrm{LS}_*(\mathcal{A}, \mathcal{B}) := \max_{\boldsymbol{\omega} \in \mathbb{R}^N} \left\{ \frac{\sum_{i \leq I, j \leq J} \mathbf{1}(\boldsymbol{\omega}^T \mathbf{m}_{ij} > 0)}{|\mathcal{A}| \cdot |\mathcal{B}|} \right\}, \tag{2}$$

where $|\mathrm{MD}(\mathcal{A}, \mathcal{B})|$ is the cardinality of $\mathrm{MD}(\mathcal{A}, \mathcal{B})$ and $\mathbf{1}(\mathcal{E})$ is the indicator function w.r.t. the event $\mathcal{E}$. It is obvious that $\mathrm{LS}_* \in [0.5, 1]$ is an absolute measure.

Denote $\mathrm{ACC}_{\mathbf{w}, \mathbf{b}}(\mathcal{A}, \mathcal{B})$ as the classification accuracy of the linear model $\mathbf{y} = \langle \mathbf{w}, \mathbf{x} \rangle + \mathbf{b}$ on the point set $\mathcal{A} \cup \mathcal{B}$, and denote $\mathrm{ACC}_{\mathrm{line}}(\mathcal{A}, \mathcal{B}) := \max_{\mathbf{w}, \mathbf{b} \in \mathbb{R}^N} \{\mathrm{ACC}_{\mathbf{w}, \mathbf{b}}(\mathcal{A}, \mathcal{B})\}$ as the maximum classification accuracy of all possible linear models. It is direct that $\mathrm{ACC}_{\mathrm{line}}(\mathcal{A}, \mathcal{B}) = (|\mathcal{A}_{\mathbf{w}}| + |\mathcal{B}_{\mathbf{w}}|)/(|\mathcal{A}| + |\mathcal{B}|)$ and $\mathrm{LS}_*(\mathcal{A}, \mathcal{B}) \geq |\mathrm{MD}(\mathcal{A}_{\mathbf{w}}, \mathcal{B}_{\mathbf{w}})|/|\mathrm{MD}(\mathcal{A}, \mathcal{B})|$. The equality of the latter holds if and only if the sets $\mathcal{A}$ and $\mathcal{B}$ are linearly separable.

**Theorem 2.5.** *Given two point sets $\mathcal{A}$ and $\mathcal{B}$, then it holds that*

$$\sqrt{\frac{|\mathcal{A}_{\boldsymbol{\omega}_*}|^2 + |\mathcal{B}_{\boldsymbol{\omega}_*}|^2}{4|\mathcal{A}| \cdot |\mathcal{B}|}} + \frac{\sqrt{2 \cdot \mathrm{LS}_*(\mathcal{A}, \mathcal{B})}}{2} \geq \mathrm{ACC}_{\mathrm{line}}(\mathcal{A}, \mathcal{B}) \geq \frac{|\mathcal{A}_{\boldsymbol{\omega}_*}| \cdot |\mathcal{B}_{\boldsymbol{\omega}_*}| \cdot \mathrm{LS}_*(\mathcal{A}, \mathcal{B})}{|\mathrm{major}_{\boldsymbol{\omega}_*}(\mathrm{MD}(\mathcal{A}, \mathcal{B}))|}, \tag{3}$$

*where $\boldsymbol{\omega}_*$ stands for the weight vector achieving the maximum operation of $\mathrm{LS}_*(\mathcal{A}, \mathcal{B})$.*

This result implies that the classification accuracy of linear models can be bounded by using $\mathrm{LS}_*(\mathcal{A}, \mathcal{B})$. Based on Eq. (2), replacing the indicator function $\mathbf{1}(\cdot)$ with the sign function $\mathrm{sgn}(\cdot)$ leads to

$$\mathrm{LS}_0(\mathcal{A}, \mathcal{B}) := \max_{\boldsymbol{\omega} \in \mathbb{R}^N} \left\{ \frac{\sum_{i \leq I, j \leq J} \mathrm{sgn}(\boldsymbol{\omega}^T \mathbf{m}_{ij})}{|\mathcal{A}| \cdot |\mathcal{B}|} \right\}, \tag{4}$$

It is obvious that $\mathrm{LS}_0 \in [0, 1]$ is an absolute measure. Let $\boldsymbol{\omega}_0$ be the weight vector achieving the maximum operation of $\mathrm{LS}_0(\mathcal{A}, \mathcal{B})$. It holds that $\mathrm{major}_{\boldsymbol{\omega}_*}(\mathrm{MD}(\mathcal{A}, \mathcal{B})) = \mathrm{major}_{\boldsymbol{\omega}_0}(\mathrm{MD}(\mathcal{A}, \mathcal{B}))$, *i.e.,* the points lying in the major sides of the two hyperplanes $\boldsymbol{\omega}_*^T \mathbf{m} = 0$ and $\boldsymbol{\omega}_0^T \mathbf{m} = 0$ are the same. Unfortunately, it is hard to solve $\mathrm{LS}_0$. Instead, another variant is considered:

$$\mathrm{LS}_1(\mathcal{A}, \mathcal{B}) := \max_{\boldsymbol{\omega}} \left\{ \Big| \sum_{i,j} \boldsymbol{\omega}^T \mathbf{m}_{ij} \Big| \Big/ \sum_{i,j} \Big| \boldsymbol{\omega}^T \mathbf{m}_{ij} \Big| \right\}. \tag{5}$$

The numerator $|\sum_{i,j} \boldsymbol{\omega}^T \mathbf{m}_{ij}|$ is the absolute value of the sum of the directed distances from the points of $\mathrm{MD}(\mathcal{A}, \mathcal{B})$ to the hyperplane $\boldsymbol{\omega}^T \mathbf{m} = 0$. We note that $\mathrm{LS}_1 \in [0, 1]$ is also an absolute measure. If all points of $\mathrm{MD}(\mathcal{A}, \mathcal{B})$ locate in one side of $\boldsymbol{\omega}^T \mathbf{m} = 0$, *i.e.,* the two sets $\mathcal{A}, \mathcal{B}$ are linearly separable, it holds that $\mathrm{LS}_1(\mathcal{A}, \mathcal{B}) = 1$. In contrast, if the value of $\mathrm{LS}_1(\mathcal{A}, \mathcal{B})$ is close to zero, the convex hulls of the two sets $\mathcal{A}, \mathcal{B}$ overlap heavily. Because of the existence of absolute value operation, it is still time-consuming to solve $\mathrm{LS}_1$. Subsequently, we discuss how to approximately calculate $\mathrm{LS}_*, \mathrm{LS}_0$ and $\mathrm{LS}_1$ with a low computation cost.

## 2.3 APPROXIMATE CALCULATION OF MD-LSMs

Set $\widetilde{\mathbf{m}} := \sum_{i \le I, j \le J} \mathbf{m}_{ij}$ and $\mathbf{M} := [\mathbf{m}_{11}, \cdots, \mathbf{m}_{1J}, \cdots, \cdots, \mathbf{m}_{I1}, \cdots, \mathbf{m}_{IJ}]_{N \times (IJ)}$. Making the terms appearing in $\mathrm{LS}_1$ squared leads to a quadratic version:

$$\mathrm{LS}_2(\mathcal{A}, \mathcal{B}) := \max_{\boldsymbol{\omega}} \left\{ \left( \sum_{i,j} \boldsymbol{\omega}^T \mathbf{m}_{ij} \right)^2 \Big/ \sum_{i,j} \left( \boldsymbol{\omega}^T \mathbf{m}_{ij} \right)^2 \right\} = \max_{\boldsymbol{\omega}} \left\{ \frac{\boldsymbol{\omega}^T \widetilde{\mathbf{m}} \widetilde{\mathbf{m}}^T \boldsymbol{\omega}}{\boldsymbol{\omega}^T \mathbf{M} \mathbf{M}^T \boldsymbol{\omega}} \right\}, \qquad (6)$$

which is a relative measure, and its solution is

$$\boldsymbol{\omega}_2 = (\mathbf{M} \mathbf{M}^T)^{-1} \widetilde{\mathbf{m}} \Big/ \sqrt{\widetilde{\mathbf{m}}^T (\mathbf{M} \mathbf{M}^T)^{-1} \widetilde{\mathbf{m}}}.$$

Then, the resultant $\boldsymbol{\omega}_2$ will be substituted into Eqs. (2) - (5) to achieve the approximate calculations of $\mathrm{LS}_*$, $\mathrm{LS}_0$ and $\mathrm{LS}_1$, respectively. It is noteworthy that since the form of Eq. (6) is similar to that of GRQ $J_{\boldsymbol{\omega}}$ (*cf.* Eq. (1)), a comparison between them is given in Appendix B.

**Remark 2.6** (Approximate Calculation of $\mathrm{LS}_*$, $\mathrm{LS}_0$ and $\mathrm{LS}_1$). *In order to efficiently calculate $\boldsymbol{\omega}_2$, we assume that $\mathbf{M} \mathbf{M}^T = \mathbf{I}$ and the solution $\boldsymbol{\omega}_2$ can be simplified as $\widehat{\boldsymbol{\omega}} = \widetilde{\mathbf{m}} / \|\widetilde{\mathbf{m}}\|$, which will be further treated as the maximizers of Eqs. (2) - (5) to approximately calculate $\mathrm{LS}_*$, $\mathrm{LS}_0$ and $\mathrm{LS}_1$, respectively. It seems to be an over-simplification, but our experiments indicate that the approximation is pretty good. For convenience, $\widehat{\mathrm{LS}}_i$ ($i \in \{*, 0, 1, 2\}$) are denoted as the MD-LSMs (including $\mathrm{LS}_*$, $\mathrm{LS}_0$, and $\mathrm{LS}_1$) and the quadratic version $\mathrm{LS}_2$ with $\boldsymbol{\omega}_i$ ($i = *, 0, 1, 2$) being replaced with $\widehat{\boldsymbol{\omega}}$, respectively.*

There naturally arises a question about the discrepancies between $\widehat{\mathrm{LS}}_i$ and $\mathrm{LS}_i$ ($i \in \{*, 0, 1, 2\}$), respectively. In the next section, we conduct comparative experiments to illustrate that the discrepancies are slight and the approximate manner is highly feasible in practice. Consequently, we achieve an efficient tool that is applicable to real-time monitoring the linear separability changes of each hidden layer after each training epoch. We also make the comparison with DGV and GRQ, respectively (*cf.* Appendix B & C).

In addition, we define the MD-LSMs for multiple-class sets:

**Definition 2.7** (MD-LSMs for Multi-class Classification). *Given $S$ point sets $\mathcal{A}_1, \cdots, \mathcal{A}_S$, denote $\mathcal{A}_s^c = \bigcup_{t \in \{1, \cdots, S\} \setminus \{s\}} \mathcal{A}_t$. Then, the MD-LSMs for the $S$ points sets are defined as:*

$$\mathrm{MultiLS}_i(\mathcal{A}_1, \cdots, \mathcal{A}_S) = \left( \sum_s |\mathcal{A}_s| \cdot \mathrm{LS}_i(\mathcal{A}_s, \mathcal{A}_s^c) \right) \Big/ \left( \sum_s |\mathcal{A}_s| \right), \quad \forall i \in \{*, 0, 1\}. \qquad (7)$$

In the one-vs-rest (OvR) way, we break down an $S$-class classification task into $S$ binary classification tasks and then compute the individual $\mathrm{LS}_i(\mathcal{A}_s, \mathcal{A}_s^c)$ of each task. Then, the MD-LSM $\mathrm{MultiLS}_i(\mathcal{A}_1, \cdots, \mathcal{A}_S)$ of the $S$-class sample sets is expressed as a sum of $\mathrm{LS}_i(\mathcal{A}_s, \mathcal{A}_s^c)$ weighted by the ratio of the size of $\mathcal{A}_s$ to the size of all samples.

## 2.4 EMPIRICAL COMPARISON

Here, we empirically compare $\mathrm{LS}_i$ with their approximation $\widehat{\mathrm{LS}}_i$ ($i \in \{*, 0, 1, 2\}$), and then examine the discrepancies among the approximate solution $\widehat{\boldsymbol{\omega}}$ and the solutions to LDA and L-SVM. Moreover, we also consider the discrepancy among different separability measures such as MD-LSMs (including $\mathrm{LS}_*$, $\mathrm{LS}_0$, $\mathrm{LS}_1$), the quadratic version $\mathrm{LS}_2$, GDV and GRQ. For convenience, denote the normal vectors of separating hyperplanes resulted from LDA and L-SVM as $\boldsymbol{\omega}_{\mathrm{LDA}}$ and $\boldsymbol{\omega}_{\mathrm{SVM}}$.

**[Comparison between $\widehat{\boldsymbol{\omega}}$ and $\boldsymbol{\omega}_i$ ($i \in \{*, 0, 1, 2\}$)]** Consider three datasets in the different degrees of linear separability: linearly separable, partly overlapped and heavily overlapped. For each dataset, we solve the optimization problems associated with $\mathrm{LS}_i$ to obtain the optimal (opt.) solutions $\boldsymbol{\omega}_i$ ($i \in \{*, 0, 1, 2\}$), respectively. By the approximate manner (*cf.* Remark 2.6), we also obtain the approximate (appr.) solutions $\widehat{\boldsymbol{\omega}}$ for the three datasets, respectively. As shown in Tabs. 3 & 6, the comparative results demonstrate that the discrepancies among $\boldsymbol{\omega}_i$ ($i \in \{*, 0, 1, 2\}$) and $\widehat{\boldsymbol{\omega}}$ are slight for the datasets in different degrees of linear separability. Their largest relative error is less than $3\%$. This finding supports the effectiveness of the approximate manner.

**[Comparison among $\widehat{\boldsymbol{\omega}}$, $\boldsymbol{\omega}_{\mathrm{LDA}}$ and $\boldsymbol{\omega}_{\mathrm{SVM}}$]** For each of the aforementioned three datasets, we implement LDA and L-SVM to obtain the solution vectors $\boldsymbol{\omega}_{\mathrm{LDA}}$ and $\boldsymbol{\omega}_{\mathrm{SVM}}$, and then make a comparison among the separating lines provided by $\widehat{\boldsymbol{\omega}}$, $\boldsymbol{\omega}_{\mathrm{LDA}}$ and $\boldsymbol{\omega}_{\mathrm{SVM}}$ (*cf.* Tab. 4). The discrepancy between the lines associated with $\widehat{\boldsymbol{\omega}}$ and $\boldsymbol{\omega}_{\mathrm{LDA}}$ (or $\boldsymbol{\omega}_{\mathrm{SVM}}$) is not significant, and their classification performances are comparable.

**[Comparison among separability measures]** In Tab. 2, we compare several kinds of LSMs and their computational costs on four kinds of binary-classification UCI datasets, including Diagnostic (Wolberg et al., 1993), Ionosphere (Sigillito et al., 1989), Maintenance (mai, 2020), and Marketing (Moro et al., 2014). For the sake of fairness, we use the differential evolution (DE) method (Storn & Price, 1997) to solve the unconstrained global optimization problems associated with $\mathrm{LS}_i$ ($i \in \{*, 0, 1, 2\}$) and GRQ. To maintain the efficiency of solving them, we control the DE method's runtime to be around 10 seconds for the first three datasets, and the runtime for the last one is around 20 seconds due to its higher data size/dimension. For each of $\mathrm{LS}_i$ ($i \in \{*, 0, 1, 2\}$) and GRQ, we make ten repeated trials of calculating them. Although the DE method is able to provide the solutions to them with a desired precision regardless of the time cost, this manner does not meet the technical requirement on the real-time monitoring of hidden-layer behaviors during the process of training deep networks. Experimental results show that the approximate manner of calculating MD-LSMs has a high efficiency, and saves at least 90% of the computational cost of exactly solving them. Meanwhile, there is a slight discrepancy between the values of $\mathrm{LS}_i$ and $\widehat{\mathrm{LS}}_i$ ($i \in \{*, 0, 1, 2\}$) .

Table 2: The averaged values of LSMs calculated on UCI datasets and times costs over ten repeated trials

| LSM Dataset | $\mathrm{LS}_*$ | | | | $\mathrm{LS}_0$ | | | |
|---|---|---|---|---|---|---|---|---|
| | opt. | time (s) | appr. | time (s) | opt. | time (s) | appr. | time (s) |
| Diagnostic | $0.9579 \pm 2.10\%$ | $9.5008 \pm 2.14\%$ | 0.9801 | $0.0996 \pm 4.22\%$ | $0.9271 \pm 2.77\%$ | $9.9836 \pm 1.13\%$ | 0.9601 | $0.1006 \pm 4.27\%$ |
| Ionosphere | $0.7732 \pm 3.74\%$ | $9.9466 \pm 15.64\%$ | 0.8608 | $0.1106 \pm 16.91\%$ | $0.5975 \pm 9.87\%$ | $11.0762 \pm 2.66\%$ | 0.7217 | $0.1108 \pm 15.52\%$ |
| Maintenance | $0.7667 \pm 8.23\%$ | $8.6463 \pm 18.82\%$ | 0.8068 | $0.5804 \pm 1.79\%$ | $0.5268 \pm 8.35\%$ | $10.6691 \pm 0.91\%$ | 0.6135 | $0.5746 \pm 1.67\%$ |
| Marketing | $0.7013 \pm 3.72\%$ | $24.0967 \pm 1.06\%$ | 0.8751 | $0.2894 \pm 0.83\%$ | $0.3797 \pm 10.61\%$ | $26.6239 \pm 0.74\%$ | 0.7503 | $0.2867 \pm 0.63\%$ |

| LSM Dataset | $\mathrm{LS}_2$ | | | | $\mathrm{LS}_1$ | | | |
|---|---|---|---|---|---|---|---|---|
| | opt. | time (s) | appr. | time (s) | opt. | time (s) | appr. | time (s) |
| Diagnostic | $36305 \pm 3.94\%$ | $12.9719 \pm 1.49\%$ | 37402 | $0.0992 \pm 4.23\%$ | $0.9849 \pm 0.70\%$ | $13.0579 \pm 0.59\%$ | 0.9932 | $0.0993 \pm 4.23\%$ |
| Ionosphere | $7232 \pm 13.39\%$ | $20.0772 \pm 13.75\%$ | 10221 | $0.1095 \pm 15.71\%$ | $0.8209 \pm 3.84\%$ | $14.6047 \pm 1.78\%$ | 0.8856 | $0.1095 \pm 15.71\%$ |
| Maintenance | $319460 \pm 39.20\%$ | $8.4768 \pm 20.35\%$ | 349939 | $0.5691 \pm 1.72\%$ | $0.5543 \pm 20.40\%$ | $7.0704 \pm 0.31\%$ | 0.6297 | $0.5699 \pm 1.68\%$ |
| Marketing | $46000 \pm 25.13\%$ | $26.8919 \pm 0.77\%$ | 129614 | $0.2833 \pm 1.02\%$ | $0.5748 \pm 12.27\%$ | $26.6174 \pm 0.76\%$ | 0.8990 | $0.2834 \pm 0.99\%$ |

| LSM Dataset | GRQ | | | | GDV | |
|---|---|---|---|---|---|---|
| | opt. | time (s) | appr. | time (s) | value | time (s) |
| Diagnostic | $0.0205 \pm 6.34\%$ | $13.7239 \pm 2.11\%$ | 0.0177 | $0.0996 \pm 4.32\%$ | 5.7181e-4 | $0.6739 \pm 2.15\%$ |
| Ionosphere | $0.0110 \pm 19.09\%$ | $14.9462 \pm 0.41\%$ | 0.0111 | $0.1099 \pm 15.65\%$ | -0.0386 | $0.2570 \pm 2.26\%$ |
| Maintenance | $0.0015 \pm 6.67\%$ | $13.5418 \pm 0.68\%$ | 0.0004 | $0.5644 \pm 1.72\%$ | -0.0081 | $11.8478 \pm 1.00\%$ |
| Marketing | $0.0008 \pm 25.00\%$ | $21.2552 \pm 2.84\%$ | 0.0022 | $0.2831 \pm 1.27\%$ | -0.0005 | $2.5638 \pm 1.89\%$ |

To sum up, the approximate manner, given in Remark 2.6, provides an efficient way of calculating the MD-LSMs $\mathrm{LS}_i$ ($i \in \{*, 0, 1\}$) with only a slight precision sacrifice. Because of its low computation cost, it also brings a reasonable tool for real-time monitoring the behavior of each hidden layer during the entire training process. In contrast, most of the existing LSMs are only applicable to off-line analyzing the hidden-layer characteristics of the trained networks due to their high computation costs.

Table 3: Separating lines and MD-LSM values of datasets in different degrees of linear separability

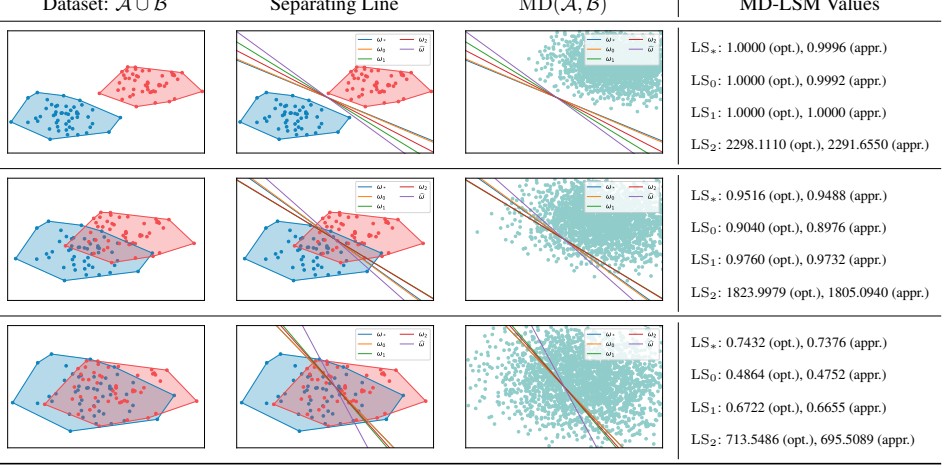

| Dataset: $\mathcal{A} \cup \mathcal{B}$ | Separating Line | $\mathrm{MD}(\mathcal{A}, \mathcal{B})$ | MD-LSM Values |
|---|---|---|---|
| | | | $\mathrm{LS}_*$: 1.0000 (opt.), 0.9996 (appr.) $\mathrm{LS}_0$: 1.0000 (opt.), 0.9992 (appr.) $\mathrm{LS}_1$: 1.0000 (opt.), 1.0000 (appr.) $\mathrm{LS}_2$: 2298.1110 (opt.), 2291.6550 (appr.) |
| | | | $\mathrm{LS}_*$: 0.9516 (opt.), 0.9488 (appr.) $\mathrm{LS}_0$: 0.9040 (opt.), 0.8976 (appr.) $\mathrm{LS}_1$: 0.9760 (opt.), 0.9732 (appr.) $\mathrm{LS}_2$: 1823.9979 (opt.), 1805.0940 (appr.) |
| | | | $\mathrm{LS}_*$: 0.7432 (opt.), 0.7376 (appr.) $\mathrm{LS}_0$: 0.4864 (opt.), 0.4752 (appr.) $\mathrm{LS}_1$: 0.6722 (opt.), 0.6655 (appr.) $\mathrm{LS}_2$: 713.5486 (opt.), 695.5089 (appr.) |

**Remark 2.8.** *As shown in Tab. 3, since the proposed MD-LSMs $\mathrm{LS}_i$ ($i = *, 0, 1$) are absolute measures, their values explicitly describe the linear separability degree of two sets. For example, the*

*fact of* $LS_1 = 1$ *means that the two sets are linearly separable; and when* $LS_1 = 0.6722$*, the two sets overlap heavily. In contrast, the quadratic version* $LS_2$ *is a relative measure. Although its value can be used to compare the linear separability degrees of different datasets, just relying on the value can't even tell whether the two point sets of a dataset are overlapped or linearly separable.*

Table 4: Separating lines of L-SVM, LDA and $\widehat{\boldsymbol{\omega}}^T\mathbf{x} + b = 0$ and classification accuracy (CA)

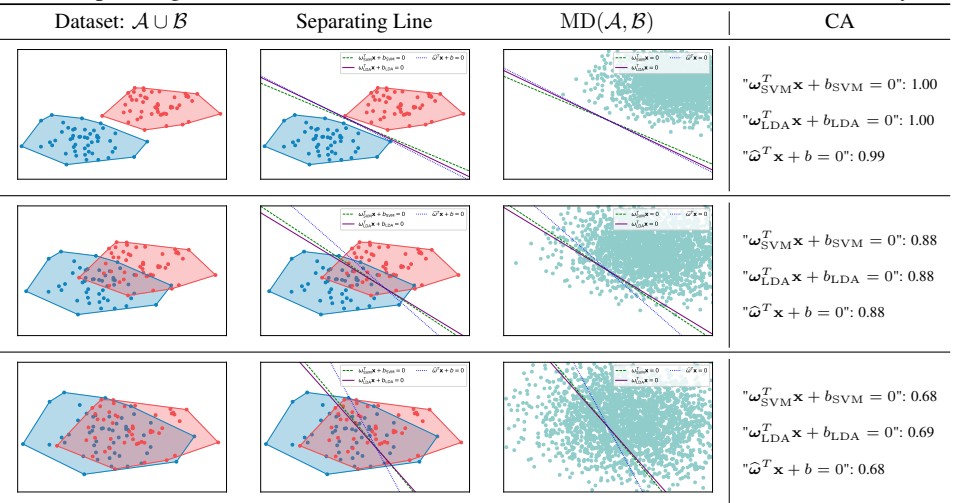

| Dataset: $\mathcal{A} \cup \mathcal{B}$ | Separating Line | $MD(\mathcal{A},\mathcal{B})$ | CA |
|---|---|---|---|
| | | | "$\boldsymbol{\omega}_{SVM}^T\mathbf{x} + b_{SVM} = 0$": 1.00 "$\boldsymbol{\omega}_{LDA}^T\mathbf{x} + b_{LDA} = 0$": 1.00 "$\widehat{\boldsymbol{\omega}}^T\mathbf{x} + b = 0$": 0.99 |
| | | | "$\boldsymbol{\omega}_{SVM}^T\mathbf{x} + b_{SVM} = 0$": 0.88 "$\boldsymbol{\omega}_{LDA}^T\mathbf{x} + b_{LDA} = 0$": 0.88 "$\widehat{\boldsymbol{\omega}}^T\mathbf{x} + b = 0$": 0.88 |
| | | | "$\boldsymbol{\omega}_{SVM}^T\mathbf{x} + b_{SVM} = 0$": 0.68 "$\boldsymbol{\omega}_{LDA}^T\mathbf{x} + b_{LDA} = 0$": 0.69 "$\widehat{\boldsymbol{\omega}}^T\mathbf{x} + b = 0$": 0.68 |

## 3 REAL-TIME MONITORING OF HIDDEN-LAYER BEHAVIORS

In this section, we conduct the numerical experiments to illustrate the application of the proposed MD-LSMs for real-time monitoring hidden-layer behaviors of several popular deep networks. All experiments are processed in the DELL® PowerEdge® T640 Tower Server with two Intel® Xeon® 20-core processors, 128 GB RAM and a NVIDIA® Tesla® V100 32GB GPU.

### 3.1 EXPERIMENT SETTING

Two classes (airplane and automobile) in CIFAR-10 dataset (Krizhevsky, 2012) are selected to form the binary classification task. The SGD method with minibatch is used to update the network weights within 100 training epochs. Since the structures of MLP and CNN are not powerful enough to obtain a good training performance by using all samples of the two classes within the limited epochs, we randomly select 2000 (resp. 1000) samples from the training (resp. testing) data of the two classes for training (resp. testing) them. Moreover, since their network sizes are not large, we directly use the selected 2000 training samples to compute the MD-LSMs of their hidden-layer outputs. In contrast, we use all training (resp. testing) data of the two classes to train (resp. test) ResNet, VGGNet, AlexNet, ViT and GoogLeNet. Since the dimension of hidden-layer outputs of these deep networks is high, in view of the computational burden, we randomly select 500 samples from the two classes to compute MD-LSMs for these networks after each training epoch. In addition, we verify the effectiveness of MD-LSMs on the MLP for solving the binary-classification tasks of the UCI datasets (including Diagnostic, Ionosphere, Maintenance and Marketing) (*cf.* Appendix E.1). We also explore the hidden-layer behaviors of the networks for solving text classification tasks, for example, the MLP and the CNN for the IMDB dataset (*cf.* Appendix E.2), and the graph neural network (GNN) for the Cora dataset (Kipf & Welling, 2016) (*cf.* Appendix E.4).

### 3.2 EXPERIMENTAL RESULTS AND DISCUSSION

In Fig. 2, we illustrate the MD-LSM curves of hidden layers and the training performance curves of the networks. Since $\widehat{LS}_0$, $\widehat{LS}_1$ and $\widehat{LS}_2$ basically have the same experimental results, we only draw the $\widehat{LS}_1$ curves for all hidden layers of MLP and CNN and for the main blocks of AlexNet, GoogLeNet, ResNet, VGGNet and ViT, respectively. The complete experimental report, containing $\widehat{LS}_0$, $\widehat{LS}_1$ and $\widehat{LS}_2$ curves for all hidden layers of these networks, is arranged in Appendix E. Moreover, we also provide the detailed structures of these neural networks with the name of each hidden layer to facilitate the interpretation of experimental results.

**[Binary Classification]** As shown in Figs. 2(a) - 2(j), there is an obvious synchronicity between the $\widehat{\mathrm{LS}}_1$ curves of hidden layers and the accuracy curves: 1) when the training accuracy increases, the $\widehat{\mathrm{LS}}_1$ value of the outputs of each hidden layer (or main block) increases synchronously; 2) when some fluctuations appear in the $\widehat{\mathrm{LS}}_1$ curves, the training and the testing accuracy curves have the fluctuations occurring nearby the corresponding epochs accordingly; 3) especially for the neural networks with relatively shallow structures, such as MLP and CNN (*cf.* Figs. 2(a) - 2(e)), the magnitude of the fluctuations in the $\widehat{\mathrm{LS}}_1$ curves is merely proportional to that of the fluctuations in the training and the testing accuracy curves.

**[Network Depth]** The experimental results, given in Figs. 2(a) - 2(k), also reflect two facts: 1) in most cases, the linear separability of the hidden layers (or blocks) is stronger than that of the original data after a few training epochs; and 2) the hidden layers (or blocks), which are closer to the output layer, have higher linear separability.

**[Multi-class Classification]** We also consider the linear separability of MLP for ten-class classification task. The experiment is conducted by using MLP to classify the MINST dataset (LeCun et al., 1998). We adopt the one-vs-rest (OvR) way to build ten MLPs with the same structure. After each training epoch, we compute the MD-LSMs of all hidden-layer outputs of each CNN in the way mentioned in Definition 2.7. As shown in Fig. 2(k), we obtain the same experiment observations as binary classification tasks and verify the theoretical findings as well.

Because of the low computational cost brought from the approximate manner of calculating MD-LSMs (*cf.* Remark 2.6), there is a reasonable tool of real-time monitoring the training behavior of each hidden layer during the entire training process rather than the post analysis of the hidden-layer characteristics of trained neural networks. The experimental results not only validate the fact that the hidden layers closer to the output layer can provide higher linear separability, but also demonstrate that the linear separability of the hidden layers adjacent to the input layers will remain unchanged (or even decrease) in the middle and late stages of the training processes. The latter finding also suggests that the early stopping of training these hidden layers should be beneficial to improving the training performance. In addition, there also arises another interesting phenomenon that the accuracy change of network outputs is in sync with the linear separability change of hidden layers (especially the ones adjacent to the output layer). This finding implies that the real-time monitoring for linear separability of individual hidden layers potentially becomes a reasonable manner of characterizing the entire network's dynamical behavior instead of only focusing on the training performance evaluated by using the network's outputs during the training process.

## 4 CONCLUSION

Because of multi-layer composite structures, it could be hard to directly analyze the properties of deep networks via the backward inference from the behavior of their outputs. Instead, analyzing the linear separability of hidden-layer outputs becomes a feasible way of understanding the deep networks. However, it is still challenge to develop the LSMs that meet the requirements of robustness, absoluteness, and efficiency. In this paper, we propose the MD-LSMs $\mathrm{LS}_i$ ($i = *, 0, 1$), which meet the first two requirements, and then derive their approximations $\widehat{\mathrm{LS}}_i$ ($i = *, 0, 1$), which meets all of the three requirements. The comparative experiments demonstrate that there is only a slight difference between $\mathrm{LS}_i$ and $\widehat{\mathrm{LS}}_i$ ($i = *, 0, 1$).

Benefited from the low cost of calculating $\widehat{\mathrm{LS}}_i$ ($i = *, 0, 1$), MD-LSMs actually provide a hidden-layer based manner of real-time monitoring the network performance instead of the traditional backward inference from the errors caused by the network outputs. As an application example, we conduct the experiments on the real-time monitoring for the linear separability of each hidden layer after each training epoch of some popular deep networks on different kinds of datasets including the synthetic datasets, the image dataset (*i.e.,* CIFAR-10), the UCI datasets (*i.e.,* Diagnostic, Ionosphere, Maintenance and Marketing), and the text datasets (*i.e.,* IMDB and Cora). First, we demonstrate that when a training sample set passes through a training or trained network, its linear separability degree gradually increases layer-by-layer and the hidden layers that are closer to the output layer will bring higher linear separability degrees. This facts explains why deep networks need a large number of hidden layers. Since one finite-width hidden layer equipped with the usual activation functions (such as sigmoid, tanh or ReLU) only has a limited nonlinear mapping capability, and thus slightly increases the linear separability degree of its inputs. Alternatively, the composition of multiple hidden layers is a feasible way of layer-wisely increasing network nonlinear mapping capability with

acceptable training difficulty. In addition, we find that there exists the synchronicity between the linear separability of hidden layers and the training accuracy in the classification tasks. There is also a theoretical discussion on such a synchronicity phenomenon. This finding implies that the linear separability potentially becomes an applicable tool of layer-wisely exploring the characteristics of deep networks.

The main limitations of this paper lie in the following aspects: 1) There still exists a gap between the setting of Proposition 1.1 and the gradient descent training. 2) When the class number $S$ is large, it is time-consuming to calculate $\mathrm{MultiLS}_i$ ($i = *, 0, 1, 2$) in the OvR way. 3) This paper only focuses on the classification tasks. Our future works will overcome these limitations. Since the quadratic version $\mathrm{LS}_2$ is of a well-defined mathematical form, it is potentially used to theoretically analyze the relationship between the network generalization capability and the network structural parameters such as activation functions and network sizes. In addition MD-LSMs can be treated as the criteria for evaluating the mapping capability of each hidden layer, and thus potentially contribute to achieve the explainable network architecture design or pruning. Since the high-dimensional vectors appearing in the expressions of MD-LSMs are of the inner-product form, we will introduce the kernel trick into them and then develop the tools of evaluating the degree of non-linear separability between two sets.

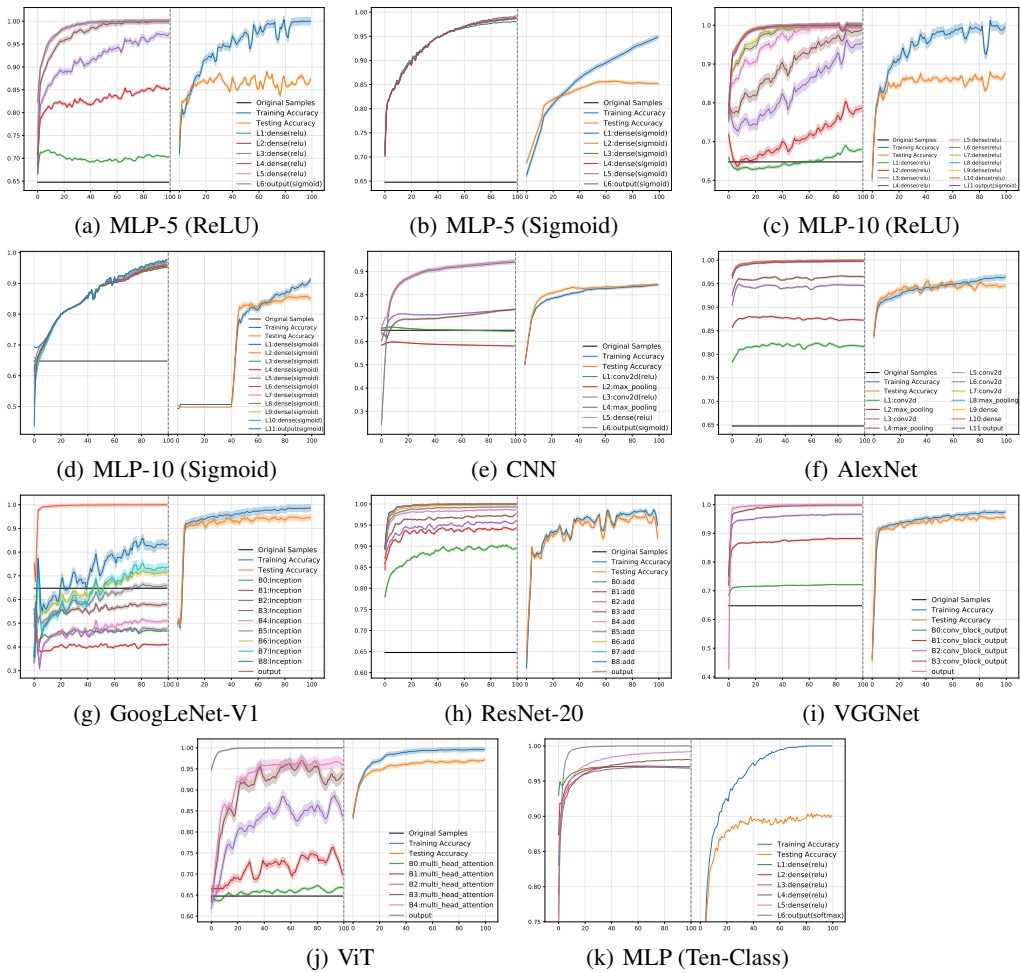

(a) MLP-5 (ReLU)    (b) MLP-5 (Sigmoid)    (c) MLP-10 (ReLU)

(d) MLP-10 (Sigmoid)    (e) CNN    (f) AlexNet

(g) GoogLeNet-V1    (h) ResNet-20    (i) VGGNet

(j) ViT    (k) MLP (Ten-Class)

Figure 2: Real-time monitoring the linear separability, evaluated by using $\widehat{\mathrm{LS}}_1$, of each hidden layer (or block) during the entire training process for different neural networks. The left (resp. right) of each subfigure shows the $\widehat{\mathrm{LS}}_1$ value of hidden-layer outputs (resp. the training and testing accuracy curves) after each training epoch, where L1 and B1 stand for the 1st hidden layer and the 1st block, respectively. The $x$-label of each subfigure stands for the training epoch. Since the ranges of $\mathrm{LS}_1$, training accuracy and testing accuracy are all the interval $[0, 1]$, the corresponding curves share the same $y$-label in each subfigure.

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

## A  WORKFLOW OF FINDING $\mathrm{MaxLS}(\mathcal{A}, \mathcal{B})$

The maximum linearly-separable subset $\mathrm{MaxLS}(\mathcal{A}, \mathcal{B})$ of two sets $\mathcal{A}$ and $\mathcal{B}$ can be obtained in an iterative way:

(1) Build an undirected bipartite graph $G(\mathcal{V}_0, \mathcal{E}_0)$ with

$$\mathcal{V}_0 = \{\text{All points in } \mathcal{A} \text{ and } \mathcal{B} \text{ associated with } \mathrm{minor}_{\boldsymbol{\omega}}(\mathrm{MD}(\mathcal{A}, \mathcal{B}))\},$$

and

$$\mathcal{E}_0 = \{\text{The connected relation of each point in } \mathrm{minor}_{\boldsymbol{\omega}}(\mathrm{MD}(\mathcal{A}, \mathcal{B}))\}.$$

For example, the connected relation of $\mathbf{m}_{ij}$ is denoted as $(\mathbf{a}_i, \mathbf{b}_j)$.

(2) Remove one vertex $v_1$ with the largest degree from $\mathcal{V}_0$ and update $\mathcal{V}_1 = \mathcal{V}_0 \setminus \{v_1\}$.

(3) Eliminate the edges associated with the vertex $v_1$ and update $\mathcal{E}_1 \leftarrow \mathcal{E}_0$.

(4) Repeat the steps (2)-(3) until $\mathcal{E}_t = \emptyset$.

(5) Remove the points in $\mathcal{V}_0 \setminus \mathcal{V}_t$ from the original sets $\mathcal{A}$ and $\mathcal{B}$, and the rest form the desired $\mathrm{MaxLS}(\mathcal{A}, \mathcal{B})$.

In Fig. 3, we illustrate the workflow of determining $\mathrm{MaxLS}(\mathcal{A}, \mathcal{B})$.

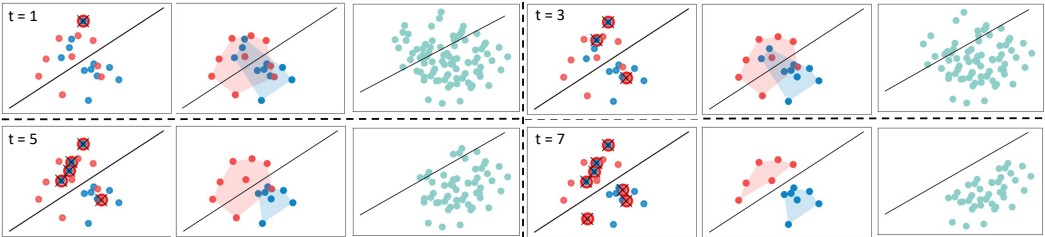

Figure 3: The workflow of obtaining the maximum linearly separable subset. Left: the point marked with a red circle containing a black 'x' has been removed in the first $t$ epochs. Middle: two convex hulls of the rest points in the two sets. Right: Minkowski difference of the rest points.

## B  COMPARISON BETWEEN $\mathrm{LS}_2$ AND GRQ

In this section, we make a comparison between $\mathrm{LS}_2$ and GRQ. Let $\boldsymbol{\mu}_a = \frac{1}{I} \sum_{i=1}^{I} \mathbf{a}_i$ and $\boldsymbol{\mu}_b = \frac{1}{J} \sum_{j=1}^{J} \mathbf{b}_j$ be the centers of the sets $\mathcal{A}$ and $\mathcal{B}$, respectively. Let $\mathbf{A}_c$ (resp. $\mathbf{B}_c$) be the matrix associated with the set $\mathcal{A}$ (resp. $\mathcal{B}$) whose columns consist of the mean shifted data points:

$$\mathbf{A}_c := [\mathbf{a}_1 - \boldsymbol{\mu}_a, \cdots, \mathbf{a}_I - \boldsymbol{\mu}_a] \text{ and } \mathbf{B}_c := [\mathbf{b}_1 - \boldsymbol{\mu}_b, \cdots, \mathbf{b}_J - \boldsymbol{\mu}_b].$$

Denote $\mathbf{S}_w = \mathbf{A}_c \mathbf{A}_c^T + \mathbf{B}_c \mathbf{B}_c^T$ and $\mathbf{S}_b = (\boldsymbol{\mu}_a - \boldsymbol{\mu}_b)(\boldsymbol{\mu}_a - \boldsymbol{\mu}_b)^T$. The GRQ, which is the objective function of LDA, can also be treated as an LSM:

$$J_{\boldsymbol{\omega}} = \max_{\boldsymbol{\omega}} \frac{\boldsymbol{\omega}^T \mathbf{S}_b \boldsymbol{\omega}}{\boldsymbol{\omega}^T \mathbf{S}_w \boldsymbol{\omega}}. \tag{8}$$

The following results show the difference between the optimization problems associated with $\mathrm{LS}_2$ and GRQ.

**Proposition B.1.** *Given two point sets $\mathcal{A} = \{\mathbf{a}_1, \cdots, \mathbf{a}_I\}$ and $\mathcal{B} = \{\mathbf{b}_1, \cdots, \mathbf{b}_J\}$, it holds that*

$$\begin{aligned}
I^2 J^2 \mathbf{S}_b &= \widetilde{\mathbf{m}} \widetilde{\mathbf{m}}^T; \\
\mathbf{S}_w &= \mathbf{A}\mathbf{A}^T + \mathbf{B}\mathbf{B}^T - \left( I(\widehat{\mathbb{E}}\mathbf{a})(\widehat{\mathbb{E}}\mathbf{a})^T + J(\widehat{\mathbb{E}}\mathbf{b})(\widehat{\mathbb{E}}\mathbf{b})^T \right); \\
\mathbf{M}\mathbf{M}^T &= J\mathbf{A}\mathbf{A}^T + I\mathbf{B}\mathbf{B}^T - IJ \cdot \widehat{\mathbb{E}}\{\mathbf{a}\mathbf{b}^T + \mathbf{b}\mathbf{a}^T\},
\end{aligned}$$

*where $\widehat{\mathbb{E}}$ stands for the sample mean with*

$$\widehat{\mathbb{E}}\mathbf{a} \;=\; \frac{1}{I}\sum_{i=1}^{I}\mathbf{a}_i;$$

$$\widehat{\mathbb{E}}\mathbf{b} \;=\; \frac{1}{J}\sum_{j=1}^{J}\mathbf{b}_j;$$

$$\widehat{\mathbb{E}}\{\mathbf{a}\mathbf{b}^T + \mathbf{b}\mathbf{a}^T\} \;=\; \frac{1}{IJ}\sum_{\substack{1\le i\le I \\ 1\le j\le J}}(\mathbf{a}_i\mathbf{b}_j^T + \mathbf{b}_j\mathbf{a}_i^T).$$

As demonstrated above, since $\mathbf{S}_w$ differs from $\mathbf{M}\mathbf{M}^T$, the hyperplane $\boldsymbol{\omega}^T\mathbf{m}=0$ achieving $\mathrm{LS}_2(\mathcal{A},\mathcal{B})$ is different from the one achieving $J_\omega$. When we use the approximate manner to compute the weight for the MD-LSMs (*cf.* Remark 2.6), the corresponding optimization objective function coincides with that of LDA with $\mathbf{S}_b=\mathbf{I}$ (*cf.* Eq. (8)). In spite of the same weight vector $\widehat{\boldsymbol{\omega}}$ derived from the approximated form, the linear separability degree is still evaluated in different forms after substituting $\widehat{\boldsymbol{\omega}}$ into the expressions of MD-LSMs (including $\mathrm{LS}_*$, $\mathrm{LS}_0$, $\mathrm{LS}_1$) and $\mathrm{LS}_2$, and $J_\omega$, respectively.

Table 5: $\widehat{\mathrm{LS}}_0$, $\widehat{\mathrm{LS}}_1$, $\widehat{\mathrm{LS}}_2$ and $\widehat{J}_\omega$ curves of hidden-layer outputs during the process of training different neural networks.

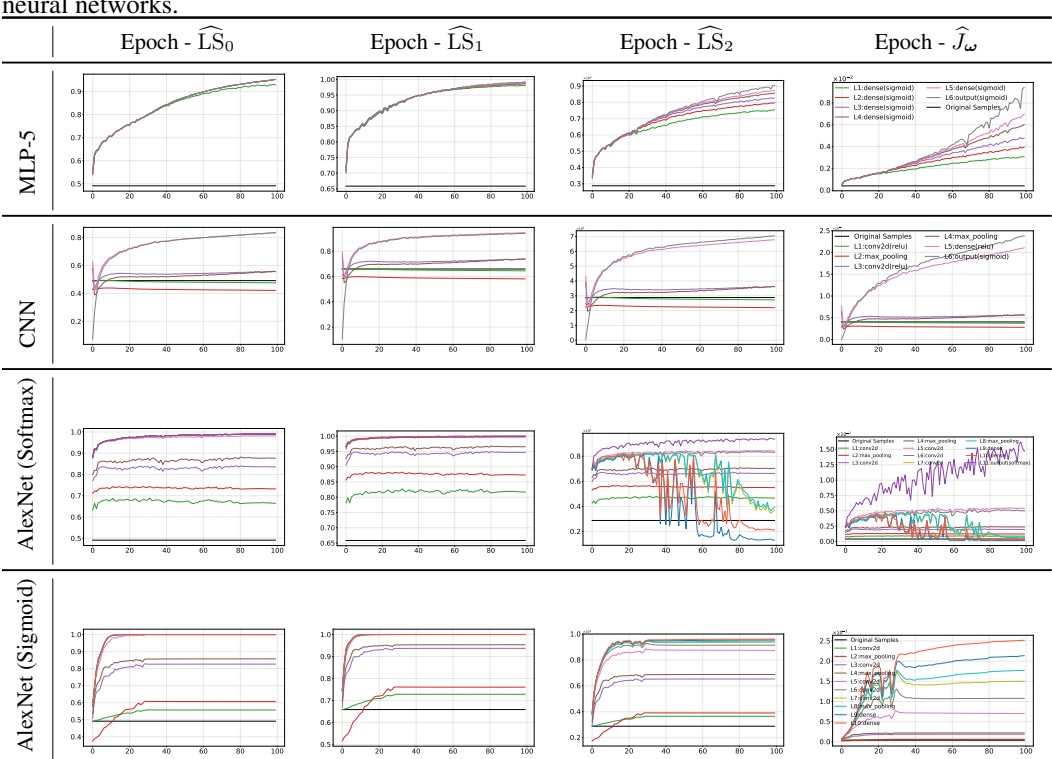

Moreover, $\mathrm{LS}_0$ and $\mathrm{LS}_1$ are absolute measures. The ranges of $\mathrm{LS}_0$ and $\mathrm{LS}_1$ are the interval $(0,1]$; and $\mathrm{LS}_0=\mathrm{LS}_1=1$ holds if and only if the two sets are linearly separable. In contrast, $\mathrm{LS}_2$ and $J_\omega$ are relative measures, and their ranges are the interval $(0,+\infty)$. Since they only provide the relative reference values for the linear separability, it is difficult to estimate the linear separability degree of two sets only based on the values of $\mathrm{LS}_2$ and $J_\omega$. Moreover, as shown in Tab. 5, there are fewer large fluctuations appearing in the curves of $\widehat{\mathrm{LS}}_0$ and $\widehat{\mathrm{LS}}_1$ than in the curves of $\widehat{\mathrm{LS}}_2$ and $\widehat{J}_\omega$, where $\widehat{J}_\omega := \frac{\widehat{\boldsymbol{\omega}}^T\mathbf{S}_b\widehat{\boldsymbol{\omega}}}{\widehat{\boldsymbol{\omega}}^T\mathbf{S}_w\widehat{\boldsymbol{\omega}}}$, *i.e.*, substituting $\widehat{\boldsymbol{\omega}}$ into the right-hide side of Eq. (8). Interestingly, the curve shapes of $\widehat{\mathrm{LS}}_0$, $\widehat{\mathrm{LS}}_1$ and $\widehat{\mathrm{LS}}_2$ are the same, but they significantly differ from that of $\widehat{J}_\omega$. Therefore, we finally

adopt $\widehat{\text{LS}}_0$, $\widehat{\text{LS}}_1$ and $\widehat{\text{LS}}_2$ as the measures of evaluating the linear separability degree of hidden-layer outputs.

## C  COMPARATIVE ANALYSIS BETWEEN GDV AND MD-LSM

Given a dataset with multiple classes $\mathcal{A}_1, \cdots, \mathcal{A}_L \subset \mathbb{R}^N$, let $N_l$ stand for the number of the points belong to the $l$-th class $C_l$ ($1 \le l \le L$). For each class $\mathcal{A}_l$, implement the z-score normalization to all points in it, and denote the resultant points as $\{\mathbf{s}_i^{(l)}\}_{i=1}^{M_l}$ ($1 \le l \le L$). Then, the GDV of the dataset is calculated in the following way (Schilling et al., 2021):

$$\text{GDV} := \frac{1}{\sqrt{N}} \left[ \frac{1}{L} \sum_{l=1}^{L} \bar{d}(\mathcal{A}_l) - \frac{2}{L(L-1)} \sum_{l=1}^{L-1} \sum_{j=l+1}^{L} \bar{d}(\mathcal{A}_l, \mathcal{A}_j) \right], \tag{9}$$

where $\bar{d}(\mathcal{A}_l)$ is the intra-class distances of $\mathcal{A}_l$ with

$$\bar{d}(\mathcal{A}_l) = \frac{2}{M_l(M_l - 1)} \sum_{i=1}^{M_l-1} \sum_{j=i+1}^{M_l} d(s_i^{(l)}, s_j^{(l)}),$$

and $\bar{d}(\mathcal{A}_l, \mathcal{A}_p)$ is the inter-class distances between $\mathcal{A}_l$ and $\mathcal{A}_p$ with

$$\bar{d}(\mathcal{A}_l, \mathcal{A}_p) = \frac{1}{M_l M_p} \sum_{i=1}^{M_l} \sum_{j=1}^{M_p} d(s_i^{(l)}, s_j^{(p)}).$$

Next, we will make the comparison between MD-LSMs and GDV from the viewpoint of whether they meet the requirements of efficiency, robustness and absoluteness.

---

**Algorithm 1** Workflow of calculating GDV [with the computational complexity of each step]

---

1: **Input:** $L$ distinct classes $\mathcal{A}_{l=1,\ldots,L}$, Each $\mathcal{A}_l$ contains $M_l$ points, $\mathbf{x}_{m=1..M_l} = (x_{m,1}, \ldots, x_{m,N})$
   ▶[Initialization, $O(1)$]
2: Each dimension of $\mathbf{x}_m$ in $\mathcal{A}_l$ is separately z-scored, $\mathbf{s}_m = (s_{m,1}, \ldots, s_{m,N})$, where $s_{m,n} = \frac{1}{2} \cdot \frac{x_{m,n} - \mu_n}{\sigma_n}$, $\mu_n = \frac{1}{M_l} \sum_{n=1}^{M_l} x_{m,n}$ and $\sigma_n = \sqrt{\frac{1}{M_l} \sum_{m=1}^{M_l} (x_{m,n} - \mu_n)^2}$.
   ▶[Z-scoring, $O(\sum_{l=1}^{L} M_l \times N)$]
3: Calculate mean intra-class distances for each class $\mathcal{A}_l$:
   $\bar{d}(\mathcal{A}_l) = \frac{2}{M_l(M_l-1)} \sum_{i=1}^{M_l-1} \sum_{j=i+1}^{M_l} d(\mathbf{s}_i^{(l)}, \mathbf{s}_j^{(l)})$.
   ▶[Intra-class, $O(\sum_{l=1}^{L} M_l(M_l - 1) \times N)$]
4: Calculate mean inter-class distances for each combination of $\mathcal{A}_l$ and $\mathcal{A}_p$:
   $\bar{d}(\mathcal{A}_l, \mathcal{A}_p) = \frac{1}{M_l M_p} \sum_{i=1}^{M_l} \sum_{j=1}^{M_p} d(s_i^{(l)}, s_j^{(p)})$.
   ▶[Inter-class, $O(\sum_{l=1}^{L-1} \sum_{p=l+1}^{L} M_l M_p \times N)$]
5: Calculate GDV:
   $\text{GDV} = \frac{1}{\sqrt{N}} \left[ \frac{1}{L} \sum_{l=1}^{L} \bar{d}(\mathcal{A}_l) - \frac{2}{L(L-1)} \sum_{l=1}^{L-1} \sum_{p=l+1}^{L} \bar{d}(\mathcal{A}_l, \mathcal{A}_p) \right]$.
   ▶[GDV calculation, $O(L^2)$]
6: **Output:** GDV
   ▶[Final output, $O(1)$]

---

Originally, MD-LSMs are designed to evaluate the degree of linear separability between two sets, while the GDV aims to measure the degree of the separability among multiple classes. For the sake of fairness, we consider the case of multiple classes and compute $\text{MultiLS}_i$ ($i \in \{*, 0, 1, 2\}$) (*cf.* Definition 2.7) via the approximate manner mentioned in Remark 2.6. Letting $M = \sum_{l=1}^{L} M_l$, the complexity of computing GDV (*cf.* Alg. 1) is

$$O \left( \sum_{l=1}^{L} M_l \times N + \sum_{l=1}^{L} M_l(M_l - 1) \times N + \sum_{l=1}^{L-1} \sum_{p=l+1}^{L} M_l \times M_p \times N + L^2 \right),$$

and the complexity of computing $\text{MultiLS}_i$ ($i \in \{*, 0, 1, 2\}$) (*cf.* Alg. 2) is

$$O\left(2 \times \sum_{l=1}^{L} M_l(M - M_l) \times N + L \times \sum_{l=1}^{L} M_l(M - M_l)\right).$$

Their computational complexities are comparable. By using GDV, Schilling et al. (2021) detected the class separability of hidden-layer outputs of the trained deep network (such as VGG, Xception and Inception), while the diagrams of experimental results have too many fluctuations to comprehensively capture the relatedness information between the class separability and different hidden layers (Schilling et al., 2021, Figs. 10 - 11). In Section 3, we have used the proposed MD-LSMs to demonstrate the synchronicity between the training performance and the linear separability of hidden-layer outputs in each training epoch, and the complete experimental report is given in Appendix E.

---

**Algorithm 2** Workflow of calculating $\text{MultiLS}_i$ ($i \in \{*, 0, 1, 2\}$) [with the computational complexity of each step]

---

1: **Input:** $L$ distinct classes $\mathcal{A}_{l=1,\dots,L}$, each $\mathcal{A}_l$ contains $M_l$ points, $\mathbf{x}_{n=1..M_l} = (x_{m,1}, \dots, x_{m,N})$.

2: Calculate Minkowski difference for each class $\mathcal{A}_l$: $\text{MD}(\mathcal{A}_l, \mathcal{A}_{l=1,\dots,l-1,l+1,\dots,L}) := \{\mathbf{m}_{ij} := \mathbf{a}_i - \mathbf{b}_j \mid \mathbf{a}_i \in \mathcal{A}_l, \ \mathbf{b}_j \in \mathcal{A}_{l=1,\dots,l-1,l+1,\dots,L}\}$.
$\qquad\qquad\qquad\qquad\qquad\qquad$ ▶[Minkowski difference, $O(\sum_{l=1}^{L} M_l(M - M_l) \times N)$]

3: Approximately calculate $\widehat{\boldsymbol{\omega}}$ for each class $\mathcal{A}_l$ (*cf.* Remark 2.6): $\widehat{\boldsymbol{\omega}}_{\mathcal{A}_l} = \frac{\widetilde{\mathbf{m}}}{\|\widetilde{\mathbf{m}}\|}$, where $\widetilde{\mathbf{m}} := \sum_{i \leq M_l, j \leq M - M_l} \mathbf{m}_{ij}$ ($\mathbf{m}_{ij} \in \text{MD}(\mathcal{A}_l, \mathcal{A}_{l=1,\dots,l-1,l+1,\dots,L})$).
$\qquad\qquad\qquad\qquad\qquad\qquad$ ▶[Calculation of $\widehat{\boldsymbol{\omega}}$, $O(\sum_{l=1}^{L} M_l(M - M_l) \times N)$]

4: Calculate $\text{MultiLS}_i$ ($i \in \{*, 0, 1, 2\}$):

$\quad$ 1. $\text{MultiLS}_*\big(\mathcal{A}_1, \cdots, \mathcal{A}_L\big) = \sum_{l=1}^{L} \frac{|\mathcal{A}_l| \cdot \widehat{\text{LS}}_*(\mathcal{A}_l, \mathcal{A}_l^c)}{\sum_{l=1}^{S} |\mathcal{A}_l|}$,
$\qquad\qquad\qquad\qquad\qquad\qquad$ ▶[Calculation of $\widehat{\text{LS}}_*$, $O(L \times \sum_{l=1}^{L} M_l(M - M_l))$]

$\qquad$ where $\widehat{\text{LS}}_*(\mathcal{A}, \mathcal{B}) := \max\left\{\frac{\sum_{i \leq I, j \leq J} \mathbf{1}(\widehat{\boldsymbol{\omega}}^T \mathbf{m}_{ij} > 0)}{|\text{MD}(\mathcal{A}, \mathcal{B})|}, \frac{\sum_{i \leq I, j \leq J} \mathbf{1}(\widehat{\boldsymbol{\omega}}^T \mathbf{m}_{ij} < 0)}{|\text{MD}(\mathcal{A}, \mathcal{B})|}\right\}$.

$\quad$ 2. $\text{MultiLS}_0\big(\mathcal{A}_1, \cdots, \mathcal{A}_L\big) = \sum_{l=1}^{L} \frac{|\mathcal{A}_l| \cdot \widehat{\text{LS}}_0(\mathcal{A}_l, \mathcal{A}_l^c)}{\sum_{l=1}^{S} |\mathcal{A}_l|}$,

$\qquad$ where $\widehat{\text{LS}}_0(\mathcal{A}, \mathcal{B}) := \max\left\{\frac{\sum_{i \leq I, j \leq J} \text{sgn}(\widehat{\boldsymbol{\omega}}^T \mathbf{m}_{ij})}{|\text{MD}(\mathcal{A}, \mathcal{B})|}, \frac{\sum_{i \leq I, j \leq J} \text{sgn}(-\widehat{\boldsymbol{\omega}}^T \mathbf{m}_{ij})}{|\text{MD}(\mathcal{A}, \mathcal{B})|}\right\}$.
$\qquad\qquad\qquad\qquad\qquad\qquad$ ▶[Calculation of $\widehat{\text{LS}}_0$, $O(L \times \sum_{l=1}^{L} M_l(M - M_l))$]

$\quad$ 3. $\text{MultiLS}_1\big(\mathcal{A}_1, \cdots, \mathcal{A}_L\big) = \sum_{l=1}^{L} \frac{|\mathcal{A}_l| \cdot \widehat{\text{LS}}_1(\mathcal{A}_l, \mathcal{A}_l^c)}{\sum_{l=1}^{S} |\mathcal{A}_l|}$,

$\qquad$ where $\widehat{\text{LS}}_1(\mathcal{A}, \mathcal{B}) := \max\left\{\left|\sum_{i \leq I, j \leq J} \widehat{\boldsymbol{\omega}}^T \mathbf{m}_{ij}\right| \Big/ \sum_{i \leq I, j \leq J} |\widehat{\boldsymbol{\omega}}^T \mathbf{m}_{ij}|\right\}$.
$\qquad\qquad\qquad\qquad\qquad\qquad$ ▶[Calculation of $\widehat{\text{LS}}_1$, $O(L \times \sum_{l=1}^{L} M_l(M - M_l))$]

$\quad$ 4. $\text{MultiLS}_2\big(\mathcal{A}_1, \cdots, \mathcal{A}_L\big) = \sum_{l=1}^{L} \frac{|\mathcal{A}_l| \cdot \widehat{\text{LS}}_2(\mathcal{A}_l, \mathcal{A}_l^c)}{\sum_{l=1}^{S} |\mathcal{A}_l|}$,

$\qquad$ where $\widehat{\text{LS}}_2(\mathcal{A}, \mathcal{B}) := \max\left\{\left(\sum_{i \leq I, j \leq J} \widehat{\boldsymbol{\omega}}^T \mathbf{m}_{ij}\right)^2 \Big/ \sum_{i \leq I, j \leq J} \left(\widehat{\boldsymbol{\omega}}^T \mathbf{m}_{ij}\right)^2\right\}$.
$\qquad\qquad\qquad\qquad\qquad\qquad$ ▶[Calculation of $\widehat{\text{LS}}_2$, $O(L \times \sum_{l=1}^{L} M_l(M - M_l))$]

5: **Output:** $\text{MultiLS}_i$ ($i \in \{*, 0, 1, 2\}$) $\qquad\qquad\qquad\qquad$ ▶[Final output, $O(1)$]

---

Consider eight datasets with different distribution characteristics, denoted as Case-$i$ ($i = 1, \cdots, 8$) respectively. For each one, we calculate the values of $\widehat{\text{LS}}_i$ (resp. $\text{LS}_i$) ($i \in \{*, 0, 1, 2\}$) and $\widehat{J}_{\boldsymbol{\omega}}$ (resp. $J_{\boldsymbol{\omega}}$) as well as the value of GDV. As shown in Tab. 6, the values of $\widehat{\text{LS}}_i$ (resp. $\text{LS}_i$) ($i \in \{*, 0, 1, 2\}$) and $\widehat{J}_{\boldsymbol{\omega}}$ (resp. $J_{\boldsymbol{\omega}}$) are consistent with the visual observations on the data distributions. Since the calculation of GDV is based on the average of intra-class and inter-class distances, some distribution

characteristics of the datasets might be eliminated after such a calculation process. Therefore, the value of GDV sometimes does not accord with the intuitive conclusion.

Table 6: Separating lines for the datasets with different distribution characteristics

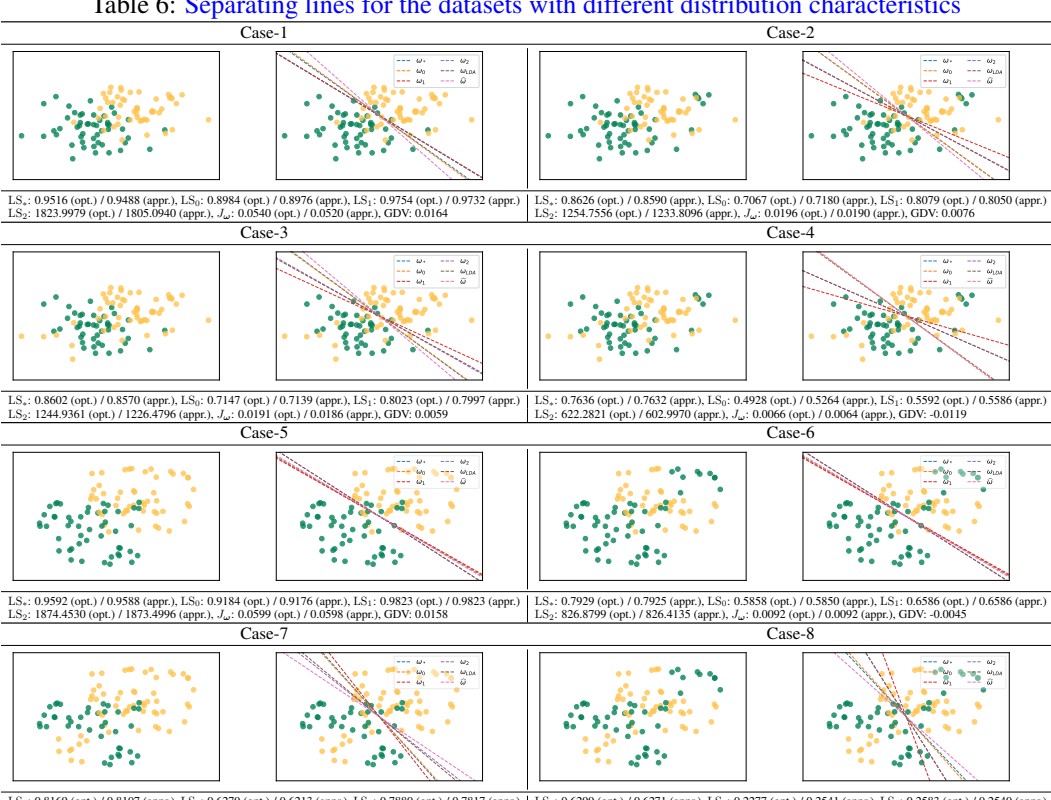

| Case-1 | Case-2 |
|---|---|
| $LS_*$: 0.9516 (opt.) / 0.9488 (appr.), $LS_0$: 0.8984 (opt.) / 0.8976 (appr.), $LS_1$: 0.9754 (opt.) / 0.9732 (appr.) $LS_2$: 1823.9979 (opt.) / 1805.0940 (appr.), $J_\omega$: 0.0540 (opt.) / 0.0520 (appr.), GDV: 0.0164 | $LS_*$: 0.8626 (opt.) / 0.8590 (appr.), $LS_0$: 0.7067 (opt.) / 0.7180 (appr.), $LS_1$: 0.8079 (opt.) / 0.8050 (appr.) $LS_2$: 1254.7556 (opt.) / 1233.8096 (appr.), $J_\omega$: 0.0196 (opt.) / 0.0190 (appr.), GDV: 0.0076 |
| Case-3 | Case-4 |
| $LS_*$: 0.8602 (opt.) / 0.8570 (appr.), $LS_0$: 0.7147 (opt.) / 0.7139 (appr.), $LS_1$: 0.8023 (opt.) / 0.7997 (appr.) $LS_2$: 1244.9361 (opt.) / 1226.4796 (appr.), $J_\omega$: 0.0191 (opt.) / 0.0186 (appr.), GDV: 0.0059 | $LS_*$: 0.7636 (opt.) / 0.7632 (appr.), $LS_0$: 0.4928 (opt.) / 0.5264 (appr.), $LS_1$: 0.5592 (opt.) / 0.5586 (appr.) $LS_2$: 622.2821 (opt.) / 602.9970 (appr.), $J_\omega$: 0.0066 (opt.) / 0.0064 (appr.), GDV: -0.0119 |
| Case-5 | Case-6 |
| $LS_*$: 0.9592 (opt.) / 0.9588 (appr.), $LS_0$: 0.9184 (opt.) / 0.9176 (appr.), $LS_1$: 0.9823 (opt.) / 0.9823 (appr.) $LS_2$: 1874.4530 (opt.) / 1873.4996 (appr.), $J_\omega$: 0.0599 (opt.) / 0.0598 (appr.), GDV: 0.0158 | $LS_*$: 0.7929 (opt.) / 0.7925 (appr.), $LS_0$: 0.5858 (opt.) / 0.5850 (appr.), $LS_1$: 0.6586 (opt.) / 0.6586 (appr.) $LS_2$: 826.8799 (opt.) / 826.4135 (appr.), $J_\omega$: 0.0092 (opt.) / 0.0092 (appr.), GDV: -0.0045 |
| Case-7 | Case-8 |
| $LS_*$: 0.8169 (opt.) / 0.8107 (appr.), $LS_0$: 0.6279 (opt.) / 0.6213 (appr.), $LS_1$: 0.7880 (opt.) / 0.7817 (appr.) $LS_2$: 1102.6562 (opt.) / 1099.4835 (appr.), $J_\omega$: 0.0152 (opt.) / 0.0151 (appr.), GDV: -0.0104 | $LS_*$: 0.6299 (opt.) / 0.6271 (appr.), $LS_0$: 0.2277 (opt.) / 0.2541 (appr.), $LS_1$: 0.2583 (opt.) / 0.2540 (appr.) $LS_2$: 121.4865 (opt.) / 117.9541 (appr.), $J_\omega$: 0.0010 (opt.) / 0.0010 (appr.), GDV: -0.0212 |

# D PROOFS OF MAIN RESULTS

In this section, we give the proofs of Theorem 2.2, Theorem 2.5, Proposition 1.1, and Proposition B.1, respectively.

## D.1 PROOF OF THEOREM 2.2

**Proof of Theorem 2.2:** "$\Longrightarrow$": If $\mathcal{A}$ and $\mathcal{B}$ are linearly separable, there exists a vector $\boldsymbol{\omega} \in \mathbb{R}^N$ and a constant $c \in \mathbb{R}$ such that the relation $\boldsymbol{\omega}^T \mathbf{a} + c > \boldsymbol{\omega}^T \mathbf{b} + c$ holds for all $\mathbf{a} \in \mathcal{A}$ and $\mathbf{b} \in \mathcal{B}$. Then, we arrive at $\boldsymbol{\omega}^T(\mathbf{a} - \mathbf{b}) > 0$ ($\forall \, \mathbf{a} \in \mathcal{A}, \mathbf{b} \in \mathcal{B}$). Namely, all points of the Minkowski difference $\mathrm{MD}(\mathcal{A}, \mathcal{B})$ lie above the hyperplane $\boldsymbol{\omega}^T \mathbf{m} = 0$.

"$\Longleftarrow$": Assume that all points of $\mathrm{MD}(\mathcal{A}, \mathcal{B})$ lie in one side of the hyperplane $\boldsymbol{\omega}^T \mathbf{m} = 0$. Without loss of generality, we consider a vector $\boldsymbol{\omega} \in \mathbb{R}^N$ such that $\boldsymbol{\omega}^T(\mathbf{a} - \mathbf{b}) > 0$ holds for any $\mathbf{a} \in \mathcal{A}$ and any $\mathbf{b} \in \mathcal{B}$. Define $\mathbf{a}^* := \arg\min_{\mathbf{a} \in \mathcal{A}}\{\boldsymbol{\omega}^T \mathbf{a}\}$ and $\mathbf{b}^\dagger := \arg\max_{\mathbf{b} \in \mathcal{B}}\{\boldsymbol{\omega}^T \mathbf{b}\}$. Then, for all $\mathbf{a} \in \mathcal{A}$ and $\mathbf{b} \in \mathcal{B}$, it holds that

$$\boldsymbol{\omega}^T \mathbf{a} - \frac{\boldsymbol{\omega}^T \mathbf{a}^* + \boldsymbol{\omega}^T \mathbf{b}^\dagger}{2} > 0 > \boldsymbol{\omega}^T \mathbf{b} - \frac{\boldsymbol{\omega}^T \mathbf{a}^* + \boldsymbol{\omega}^T \mathbf{b}^\dagger}{2}.$$

Namely, the hyperplane $\boldsymbol{\omega}^T \mathbf{m} - \dfrac{\boldsymbol{\omega}^T \mathbf{a}^* + \boldsymbol{\omega}^T \mathbf{b}^\dagger}{2} = 0$ ($\mathbf{m} \in \mathbb{R}^N$) separates the set $\mathcal{A}$ from the set $\mathcal{B}$. This completes the proof. ∎

## D.2 PROOF OF THEOREM 2.5

**Proof of Theorem 2.5:** (1) First, we prove the second inequality. It follows from $\mathcal{A}_{\boldsymbol{\omega}_*} \subseteq \mathcal{A}$ and $\mathcal{B}_{\boldsymbol{\omega}_*} \subseteq \mathcal{B}$ that

$$\frac{1}{\frac{1}{|\mathcal{A}|} + \frac{1}{|\mathcal{B}|}} \geq \frac{1}{\frac{1}{|\mathcal{A}_{\boldsymbol{\omega}_*}|} + \frac{1}{|\mathcal{B}_{\boldsymbol{\omega}_*}|}}$$

$$\Longleftrightarrow \frac{|\mathcal{A}_{\boldsymbol{\omega}_*}| + |\mathcal{B}_{\boldsymbol{\omega}_*}|}{|\mathcal{A}| + |\mathcal{B}|} \geq \frac{|\mathcal{A}_{\boldsymbol{\omega}_*}| \cdot |\mathcal{B}_{\boldsymbol{\omega}_*}|}{|\mathcal{A}| \cdot |\mathcal{B}|}$$

$$\Longleftrightarrow \frac{|\mathcal{A}_{\boldsymbol{\omega}_*}| + |\mathcal{B}_{\boldsymbol{\omega}_*}|}{|\mathcal{A}| + |\mathcal{B}|} \geq \frac{|\mathcal{A}_{\boldsymbol{\omega}_*}| \cdot |\mathcal{B}_{\boldsymbol{\omega}_*}|}{|\mathrm{major}_{\boldsymbol{\omega}_*}(\mathrm{MD}(\mathcal{A}, \mathcal{B}))|} \cdot \frac{|\mathrm{major}_{\boldsymbol{\omega}_*}(\mathrm{MD}(\mathcal{A}, \mathcal{B}))|}{|\mathcal{A}| \cdot |\mathcal{B}|}$$

$$\Longleftrightarrow \frac{|\mathcal{A}_{\boldsymbol{\omega}_*}| + |\mathcal{B}_{\boldsymbol{\omega}_*}|}{|\mathcal{A}| + |\mathcal{B}|} \geq \frac{|\mathcal{A}_{\boldsymbol{\omega}_*}| \cdot |\mathcal{B}_{\boldsymbol{\omega}_*}| \cdot \mathrm{LS}_*(\mathcal{A}, \mathcal{B})}{\mathrm{major}_{\boldsymbol{\omega}_*}(\mathrm{MD}(\mathcal{A}, \mathcal{B}))}.$$

The last step is due to the definition of $\mathrm{LS}_*(\mathcal{A}, \mathcal{B})$ and the fact that $|\mathrm{MD}(\mathcal{A}, \mathcal{B})| = |\mathcal{A}| \cdot |\mathcal{B}|$.

(2) Since $\mathrm{LS}_*(\mathcal{A}, \mathcal{B}) \geq \frac{|\mathcal{A}_{\boldsymbol{\omega}_*}| \cdot |\mathcal{B}_{\boldsymbol{\omega}_*}|}{|\mathcal{A}| \cdot |\mathcal{B}|}$, we have

$$\begin{aligned}
\mathrm{ACC}_{\mathrm{line}}^2(\mathcal{A}, \mathcal{B}) &= \frac{(|\mathcal{A}_{\boldsymbol{\omega}_*}| + |\mathcal{B}_{\boldsymbol{\omega}_*}|)^2}{(|\mathcal{A}| + |\mathcal{B}|)^2} \\
&\leq \frac{(|\mathcal{A}_{\boldsymbol{\omega}_*}| + |\mathcal{B}_{\boldsymbol{\omega}_*}|)^2}{4|\mathcal{A}| \cdot |\mathcal{B}|} \\
&= \frac{|\mathcal{A}_{\boldsymbol{\omega}_*}|^2 + |\mathcal{B}_{\boldsymbol{\omega}_*}|^2 + 2|\mathcal{A}_{\boldsymbol{\omega}_*}| \cdot |\mathcal{B}_{\boldsymbol{\omega}_*}|}{4|\mathcal{A}| \cdot |\mathcal{B}|} \\
&\leq \frac{|\mathcal{A}_{\boldsymbol{\omega}_*}|^2 + |\mathcal{B}_{\boldsymbol{\omega}_*}|^2}{4|\mathcal{A}| \cdot |\mathcal{B}|} + \frac{\mathrm{LS}_*(\mathcal{A}, \mathcal{B})}{2}.
\end{aligned}$$

Then, it follows from the fact $\sqrt{a + b} \leq \sqrt{a} + \sqrt{b}$ ($\forall a, b \geq 0$) that

$$\mathrm{ACC}_{\mathrm{line}}(\mathcal{A}, \mathcal{B}) \leq \sqrt{\frac{|\mathcal{A}_{\boldsymbol{\omega}_*}|^2 + |\mathcal{B}_{\boldsymbol{\omega}_*}|^2}{4|\mathcal{A}| \cdot |\mathcal{B}|}} + \frac{\sqrt{2 \cdot \mathrm{LS}_*(\mathcal{A}, \mathcal{B})}}{2}.$$

This completes the proof. ∎

### D.3   PROOF OF PROPOSITION 1.1

**Proof of Proposition 1.1:** (1) "$\Longleftarrow$" If the linear separability degree of the $L$-th hidden-layer outputs increases after updating the hidden-layer weights $\mathbf{V}_1, \cdots, \mathbf{V}_L$ to be $\mathbf{V}'_1, \cdots, \mathbf{V}'_L$ respectively, it means that there exists a hyperplane $\mathbf{w}^T \mathbf{s} + \mathbf{b} = 0$ such that more $L$-th hidden-layer outputs can be correctly separated. Since the hyperplane $(\mathbf{w}')^T \mathbf{s} + b' = 0$ can provide the highest training classification accuracy, the training performance of $\text{net}'(\cdot)$ is better than that of $\text{net}(\cdot)$.

"$\Longrightarrow$" If the classification accuracy increases, it means that more $L$-th hidden-layer outputs can be correctly separated by the hyperplane $(\mathbf{w}')^T \mathbf{s} + b' = 0$. Namely, the linear separability of hidden-layer outputs increases. This completes the proof. ∎

### D.4   PROOF OF PROPOSITION B.1

**Proof of Proposition B.1:** Denote $\mathbf{A} = [\mathbf{a}_1, \cdots, \mathbf{a}_I]$ and $\mathbf{B} = [\mathbf{b}_1, \cdots, \mathbf{b}_J]$. Let $\mathbf{1} = (1, \cdots, 1)^T$ be the vector whose components are all ones. Since $\boldsymbol{\mu}_a = \frac{1}{I}\mathbf{A}\mathbf{1}$ and $[\boldsymbol{\mu}_a, \cdots, \boldsymbol{\mu}_a] = \frac{1}{I}\mathbf{A}\mathbf{1}\mathbf{1}^T$, we have

$$
\begin{aligned}
\mathbf{A}_c\mathbf{A}_c^T &= \left(\mathbf{A} - \frac{1}{I}\mathbf{A}\mathbf{1}\mathbf{1}^T\right)\left(\mathbf{A} - \frac{1}{I}\mathbf{A}\mathbf{1}\mathbf{1}^T\right)^T \\
&= \mathbf{A}\mathbf{A}^T - \frac{1}{I}\mathbf{A}\mathbf{1}\mathbf{1}^T\mathbf{A}^T - \frac{1}{I}\mathbf{A}\mathbf{1}\mathbf{1}^T\mathbf{A}^T + \frac{1}{I^2}\mathbf{A}\mathbf{1}\mathbf{1}^T\mathbf{1}\mathbf{1}^T\mathbf{A}^T \\
&= \mathbf{A}\mathbf{A}^T - \frac{1}{I}\mathbf{A}\mathbf{1}\mathbf{1}^T\mathbf{A}^T - \frac{1}{I}\mathbf{A}\mathbf{1}\mathbf{1}^T\mathbf{A}^T + \frac{1}{I}\mathbf{A}\mathbf{1}\mathbf{1}^T\mathbf{A}^T \\
&= \mathbf{A}\mathbf{A}^T - \frac{1}{I}\mathbf{A}\mathbf{1}\mathbf{1}^T\mathbf{A}^T.
\end{aligned}
$$

In the similar way, we also have $\mathbf{B}_c\mathbf{B}_c^T = \mathbf{B}\mathbf{B}^T - \frac{1}{J}\mathbf{B}\mathbf{1}\mathbf{1}^T\mathbf{B}^T$. Thus, the matrices $\mathbf{S}_w$ and $\mathbf{S}_b$ can be rewritten as

$$
\begin{aligned}
\mathbf{S}_w &= \mathbf{A}_c\mathbf{A}_c^T + \mathbf{B}_c\mathbf{B}_c^T \\
&= \mathbf{A}\mathbf{A}^T + \mathbf{B}\mathbf{B}^T - \frac{1}{I}\mathbf{A}\mathbf{1}\mathbf{1}^T\mathbf{A}^T - \frac{1}{J}\mathbf{B}\mathbf{1}\mathbf{1}^T\mathbf{B}^T; \\
\mathbf{S}_b &= (\boldsymbol{\mu}_a - \boldsymbol{\mu}_b)(\boldsymbol{\mu}_a - \boldsymbol{\mu}_b)^T \\
&= \frac{1}{I^2}\mathbf{A}\mathbf{1}\mathbf{1}^T\mathbf{A}^T + \frac{1}{J^2}\mathbf{B}\mathbf{1}\mathbf{1}^T\mathbf{B}^T - \frac{1}{IJ}\mathbf{A}\mathbf{1}\mathbf{1}^T\mathbf{B}^T - \frac{1}{IJ}\mathbf{B}\mathbf{1}\mathbf{1}^T\mathbf{A}^T.
\end{aligned}
$$

Denote

$$
\begin{aligned}
\mathrm{D}(\mathbf{A}; J) &:= \underbrace{[\mathbf{a}_1, \cdots, \mathbf{a}_I, \cdots, \mathbf{a}_1, \cdots, \mathbf{a}_I]}_{J \text{ groups of } \{\mathbf{a}_1, \cdots, \mathbf{a}_I\}} \in \mathbb{R}^{N \times IJ}; \\
\mathrm{D}(\mathbf{B}; I) &:= \underbrace{[\mathbf{b}_1, \cdots, \mathbf{b}_J, \cdots, \mathbf{b}_1, \cdots, \mathbf{b}_J]}_{I \text{ groups of } \{\mathbf{b}_1, \cdots, \mathbf{b}_J\}} \in \mathbb{R}^{N \times IJ}.
\end{aligned}
$$

Since $\mathbf{m}_{ij} = \mathbf{a}_i - \mathbf{b}_j$, $\mathbf{M}$ can be rewritten as

$$
\begin{aligned}
\mathbf{M} &= [\mathbf{m}_{11}, \cdots, \mathbf{m}_{1J}, \cdots, \mathbf{m}_{i1}, \cdots, \mathbf{m}_{iJ}, \cdots, \mathbf{m}_{I1}, \cdots, \mathbf{m}_{IJ}]_{N \times IJ} \\
&= \mathrm{D}(\mathbf{A}; J) - \mathrm{D}(\mathbf{B}; I).
\end{aligned}
$$

Then, we have

$$
\begin{aligned}
\widetilde{\mathbf{m}}\widetilde{\mathbf{m}}^T &= \left(\sum_{ij}\mathbf{m}_{ij}\right)\left(\sum_{ij}\mathbf{m}_{ij}\right)^T \\
&= [\mathrm{D}(\mathbf{A}; J)\mathbf{1} - \mathrm{D}(\mathbf{B}; I)\mathbf{1}] \cdot [\mathrm{D}(\mathbf{A}; J)\mathbf{1} - \mathrm{D}(\mathbf{B}; I)\mathbf{1}]^T \\
&= \mathrm{D}(\mathbf{A}; J)\mathbf{1}\mathbf{1}^T\mathrm{D}^T(\mathbf{A}; J) + \mathrm{D}(\mathbf{B}; I)\mathbf{1}\mathbf{1}^T\mathrm{D}^T(\mathbf{B}; I) \\
&\qquad\qquad -\mathrm{D}(\mathbf{B}; I)\mathbf{1}\mathbf{1}^T\mathrm{D}^T(\mathbf{A}; J) - \mathrm{D}(\mathbf{A}; J)\mathbf{1}\mathbf{1}^T\mathrm{D}^T(\mathbf{B}; I) \\
&= J^2\mathbf{A}\mathbf{1}\mathbf{1}^T\mathbf{A}^T + I^2\mathbf{B}\mathbf{1}\mathbf{1}^T\mathbf{B}^T - IJ\mathbf{B}\mathbf{1}\mathbf{1}^T\mathbf{A}^T - IJ\mathbf{A}\mathbf{1}\mathbf{1}^T\mathbf{B}^T \\
&= I^2J^2\left(\frac{1}{I^2}\mathbf{A}\mathbf{1}\mathbf{1}^T\mathbf{A}^T + \frac{1}{J^2}\mathbf{B}\mathbf{1}\mathbf{1}^T\mathbf{B}^T - \frac{1}{IJ}\mathbf{A}\mathbf{1}\mathbf{1}^T\mathbf{B}^T - \frac{1}{IJ}\mathbf{B}\mathbf{1}\mathbf{1}^T\mathbf{A}^T\right).
\end{aligned}
$$

It is direct that

$$I^2 J^2 \mathbf{S}_b = \widetilde{\mathbf{m}} \widetilde{\mathbf{m}}^T,$$

which implies that the eigenvectors of the two matrices $\mathbf{S}_b$ and $\widetilde{\mathbf{m}} \widetilde{\mathbf{m}}^T$ have the same direction.

Moreover, let $\mathbf{a}$ and $\mathbf{b}$ stand for the random variables obeying the probability distributions on the sets $\mathcal{A}$ and $\mathcal{B}$, respectively. Since $\mathbf{A1} = I \cdot \widehat{\mathbb{E}} \mathbf{a} = \sum_{i=1}^{I} \mathbf{a}_i$ and $\mathbf{B1} = J \cdot \widehat{\mathbb{E}} \mathbf{b} = \sum_{j=1}^{J} \mathbf{b}_j$, we have

$$
\begin{aligned}
\mathbf{S}_w &= \mathbf{A}\mathbf{A}^T + \mathbf{B}\mathbf{B}^T - \frac{1}{I}\mathbf{A}\mathbf{1}\mathbf{1}^T\mathbf{A}^T - \frac{1}{J}\mathbf{B}\mathbf{1}\mathbf{1}^T\mathbf{B}^T \\
&= \mathbf{A}\mathbf{A}^T + \mathbf{B}\mathbf{B}^T - \left[ I(\widehat{\mathbb{E}}\mathbf{a})(\widehat{\mathbb{E}}\mathbf{a})^T + J(\widehat{\mathbb{E}}\mathbf{b})(\widehat{\mathbb{E}}\mathbf{b})^T \right].
\end{aligned}
$$

Since $\sum_{i,j}(\mathbf{a}_i\mathbf{b}_j^T + \mathbf{b}_j\mathbf{a}_i^T) = IJ \cdot \widehat{\mathbb{E}}\{\mathbf{a}\mathbf{b}^T + \mathbf{b}\mathbf{a}^T\}$, we have

$$
\begin{aligned}
\mathbf{M}\mathbf{M}^T &= \left[ \mathrm{D}(\mathbf{A};J) - \mathrm{D}(\mathbf{B};I) \right] \left[ \mathrm{D}(\mathbf{A};J) - \mathrm{D}(\mathbf{B};I) \right]^T \\
&= J\mathbf{A}\mathbf{A}^T + I\mathbf{B}\mathbf{B}^T - \sum_{i,j}(\mathbf{a}_i\mathbf{b}_j^T + \mathbf{b}_j\mathbf{a}_i^T) \\
&= J\mathbf{A}\mathbf{A}^T + I\mathbf{B}\mathbf{B}^T - IJ \cdot \widehat{\mathbb{E}}\{\mathbf{a}\mathbf{b}^T + \mathbf{b}\mathbf{a}^T\}.
\end{aligned}
$$

This completes the proof. ∎

# E  COMPLETE EXPERIMENTAL REPORT ON REAL-TIME MONITORING

In this part, we provide the experimental results of three kinds of MD-LSMs: $LS_0$, $LS_1$ and $LS_2$. In view of the complicated structures of VGGNet, ResNet-20, GoogLeNet-V1 and ViT, we also draw the structure diagrams to denote their hidden layers or main blocks. In Tab. 10, we show the arrangement of the structure diagrams and the experimental results. We note that the $x$-label of all figures stands for the training epoch.

## E.1  MLP

First, we layer-wisely examine the linear separability of the MLPs with five hidden layers, denoted as MLP-5, and ten hidden layers, denoted as MLP-10, respectively. The hidden nodes of MLPs are activated by using Sigmoid functions (denoted as Sigmoid) and ReLU functions (denoted as ReLU), respectively. In Figs. 4 - 7, we illustrate the experimental results of MLPs in the binary classification tasks. Moreover, we also conduct the experiments of the MLP-5 (ReLU) on the binary-classification UCI datasets (including Diagnostic, Marketing, Ionosphere, and Maintenance), and obtain the similar experimental results with the aforementioned ones. In addition, we also consider the linear separability of MLPs in ten-class classification task, where the network has five hidden layers and its hidden nodes are activated by using ReLU (*cf.* Fig. 8).

In Tab. 8, we illustrate the MD-LSM curves of the hidden-layer outputs of the MLPs with varying numbers (from 1 to 5) of hidden layers. For the hidden layer that is closer to the output layer, its outputs have the stronger linear separability, and this experimental phenomenon is in accordance with the intuitive explanation, mentioned in the existing works (Alain & Bengio, 2016; Apicella et al., 2024; He & Su, 2023; Schilling et al., 2021), to the working mechanism of deep networks. Interestingly, in the MLPs with multiple hidden layers, the linear separability of the hidden layer that is closest to the input layer could become degraded in the middle and late stages of the training process, *i.e.,* this layer could become helpless to improve the network's classification accuracy. This phenomenon has called the feature freezing in the recent literature (Bär et al., 2024). Our experimental results demonstrate the existence of this phenomenon, and the proposed MD-LSMs could become the potential tool of analyzing the issue on this phenomenon.

Table 7: $\widehat{LS}_0$, $\widehat{LS}_1$, $\widehat{LS}_2$ curves and the accuracy curves during the process of training MLP-5 (ReLU) networks on UCI datasets

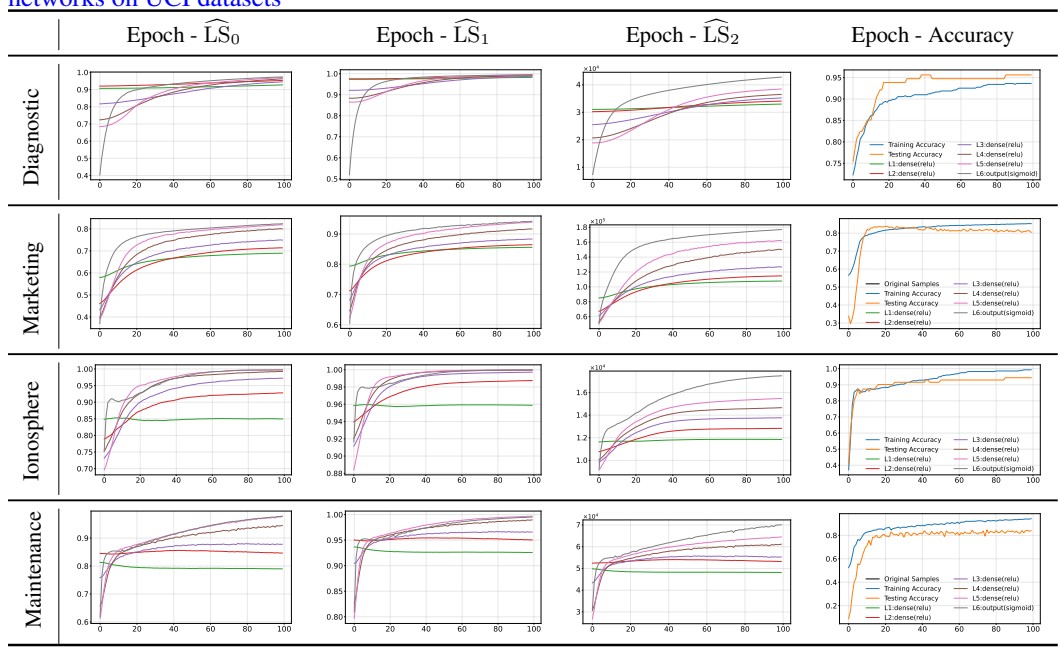

Table 8: $\widehat{LS}_0$, $\widehat{LS}_1$, $\widehat{LS}_2$ curves and the accuracy curves during the process of training MLP (ReLU) networks whose numbers of hidden layers range from 1 to 5.

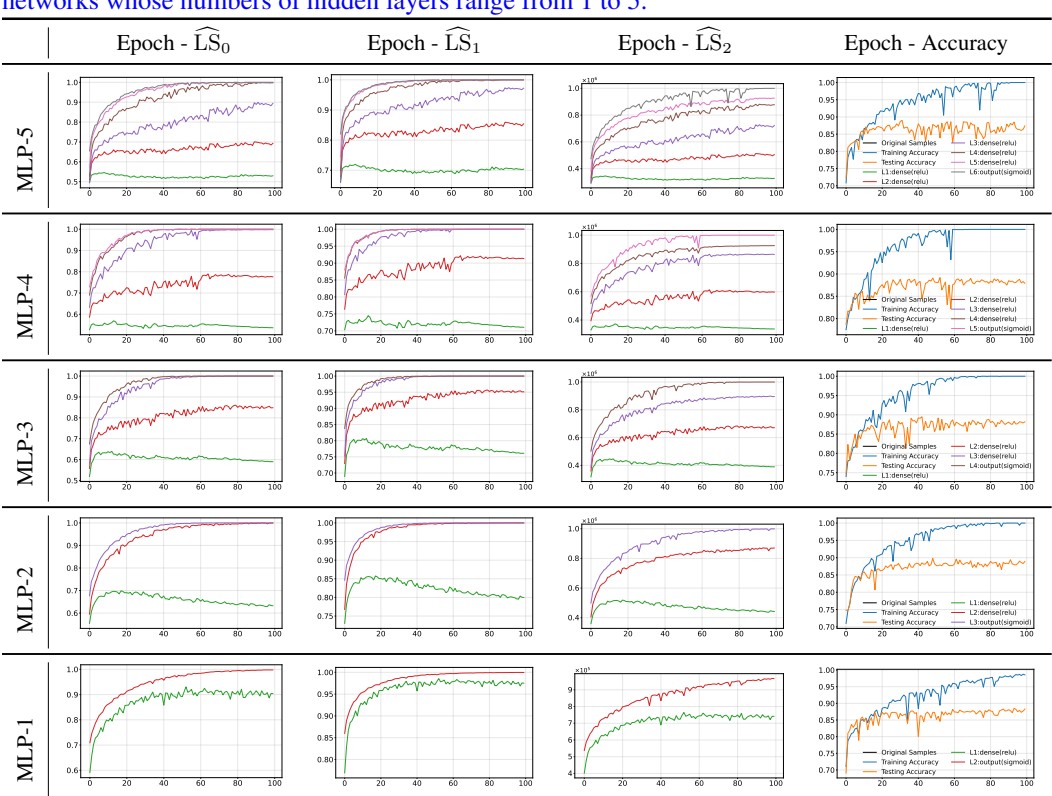

### E.2    CNN, ALEXNET AND DBN

Moreover, we examine the linear separability of CNNs with two convolution layers and two pooling layers in binary classification task. All hidden nodes of CNNs are activated by using ReLU (*cf.* Fig. 9). Moreover, the linear separability of AlexNet and DBN is also considered in the same task (*cf.* Figs. 10 - 12). It is noteworthy that we consider two kinds of AlexNets that have different output activation functions: one is Softmax, denoted as AlexNet (Softmax), and the other is Sigmoid, denoted as AlexNet (Sigmoid). We also simplify the process of training AlexNet (Sigmoid), where the tricks of learning rate decay and data augmentation are not used. Since the binary classification is much simpler than the ImageNet classification task for which AlexNet was originally designed, the simplified training process is enough to provide a good performance. Thus, the curves of AlexNet (Sigmoid) are smoother than those of AlexNet (Softmax), especially for the $\widehat{\mathrm{LS}}_2$.

In addition, we also conduct the experiment on the IMDB dataset, which is a text classification task. When utilizing neural networks to process text data, the embedding layer is employed to map textual information into vector spaces. For MLPs, the outputs of the embedding layer are typically flattened to be compatible with the subsequent dense layer, and this manner could results in the loss of spatial information. In contrast, benefited from the specific convolutional structure, CNNs are able to capture the spatial information encoded in the outputs of the embedding space, and thus to improve the representation capability of the embedding layer. As illustrated in Tab. 9, the linear separability of CNN's embedding layer gradually increases during its training process, and the experiment results demonstrate the embedding layer of CNN plays a more important role in processing text data than that of MLP.

Table 9: $\widehat{\mathrm{LS}}_0$, $\widehat{\mathrm{LS}}_1$, $\widehat{\mathrm{LS}}_2$ curves and the accuracy curves during the process of training MLP-5 (ReLU) and CNN on the IMDB dataset.

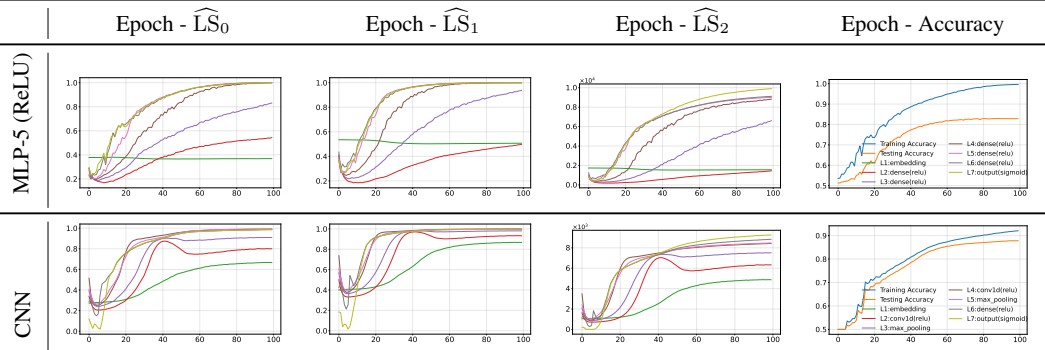

### E.3    VGGNET, GOOGLENET, RESNET AND VIT

Here, we consider the linear separability of the deep networks with complicated hidden-layer structures, including VGGNet, GoogLeNet-V1, ResNet-20 and ViT. Since the structures of these networks can be split into some individual blocks, we first examine the linear separability of the outputs of their main blocks, and then illustrate the MD-LSMs of hidden layers of these networks.

Table 10: Numerical Experiment Results

| Deep Networks | Structure Diagram | Main Blocks | Hidden Layers |
|---|---|---|---|
| MLP-5 (ReLU) | | | Fig. 4 |
| MLP-5 (Sigmoid) | | | Fig. 5 |
| MLP-10 (ReLU) | | | Fig. 6 |
| MLP-10 (Sigmoid) | | | Fig. 7 |
| MLP (Ten-Class) | | | Fig. 8 |
| CNN | | | Fig. 9 |
| AlexNet (Softmax) | | | Fig. 10 |
| AlexNet (Sigmoid) | | | Fig. 11 |
| DBN | | | Fig. 12 |
| VGGNet | Fig. 14 | Fig. 13 | Fig. 15 |
| GoogLeNet-V1 | Fig. 16 | Fig. 17 | Fig. 18 |
| ResNet-20 | Fig. 19 | Fig. 20 | Fig. 21 |
| ViT | Fig. 22 | Fig. 23 | Fig. 24 |

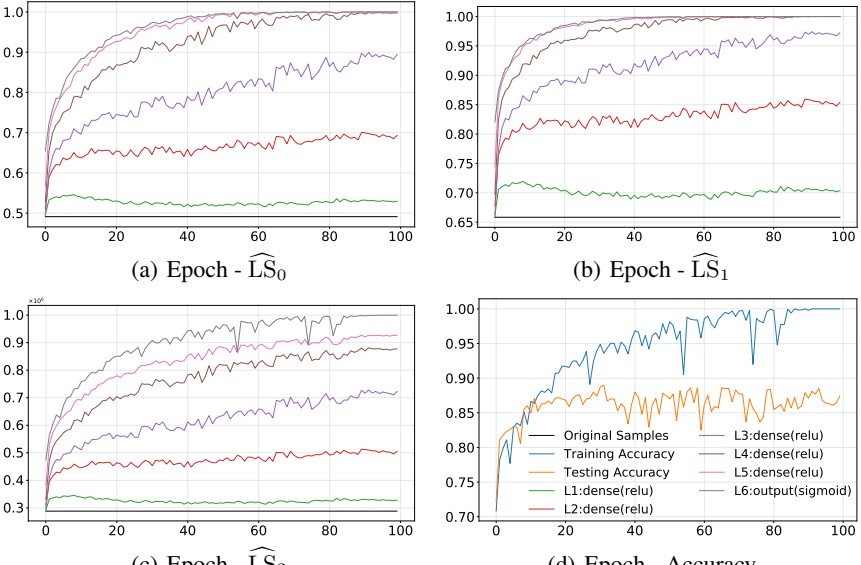

(a) Epoch - $\widehat{LS}_0$

(b) Epoch - $\widehat{LS}_1$

(c) Epoch - $\widehat{LS}_2$

(d) Epoch - Accuracy

Figure 4: MD-LSM and Accuracy Curves of Hidden Layers of MLP-5 (ReLU)

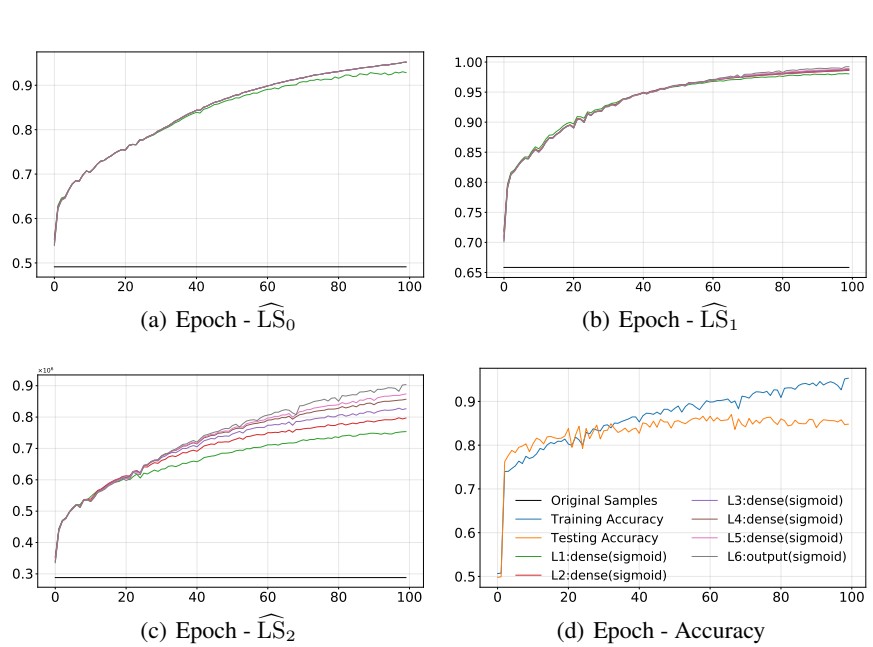

Figure 5: MD-LSM and Accuracy Curves of Hidden Layers of MLP-5 (Sigmoid)

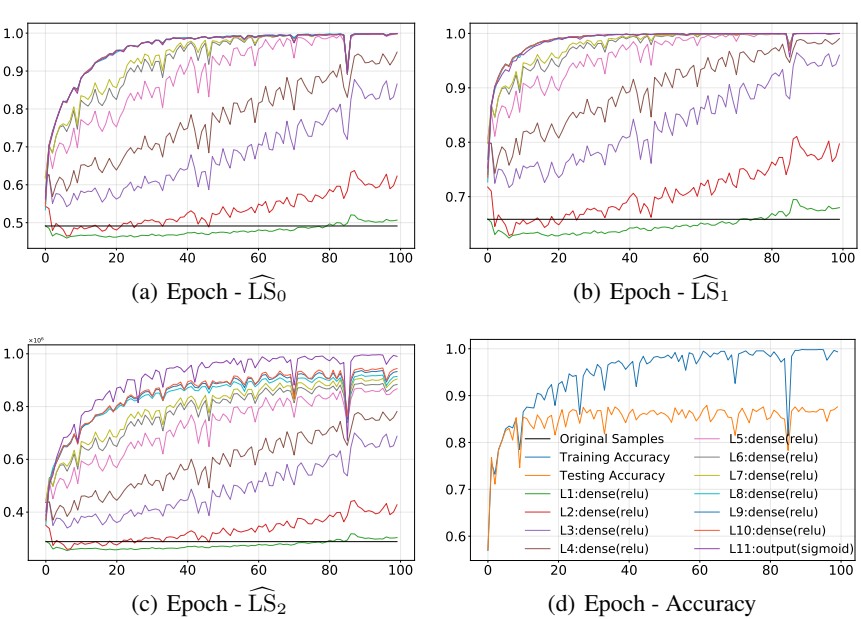

Figure 6: MD-LSM and Accuracy Curves of Hidden Layers of MLP-10 (ReLU)

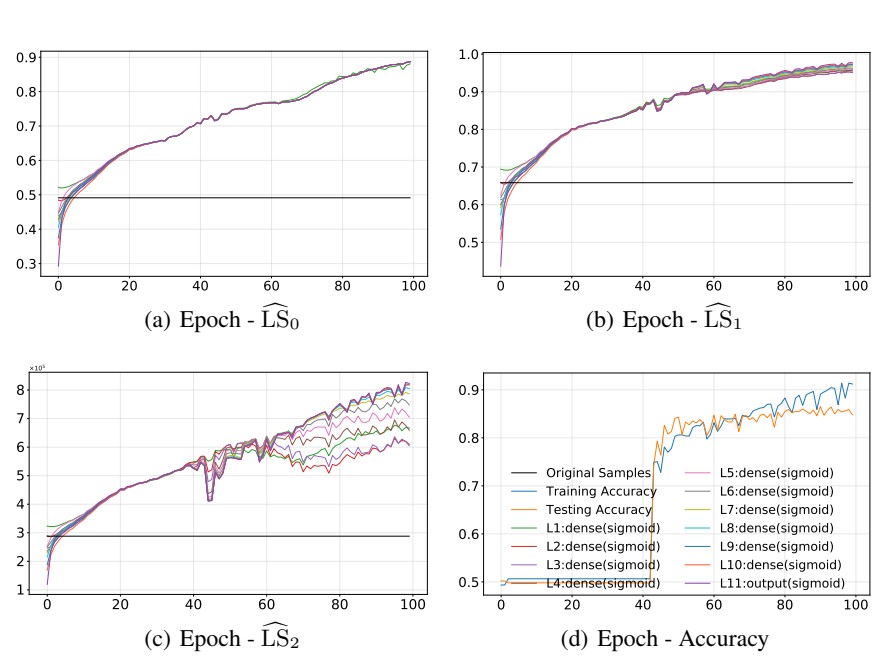

Figure 7: MD-LSM and Accuracy Curves of Hidden Layers of MLP-10 (Sigmoid)

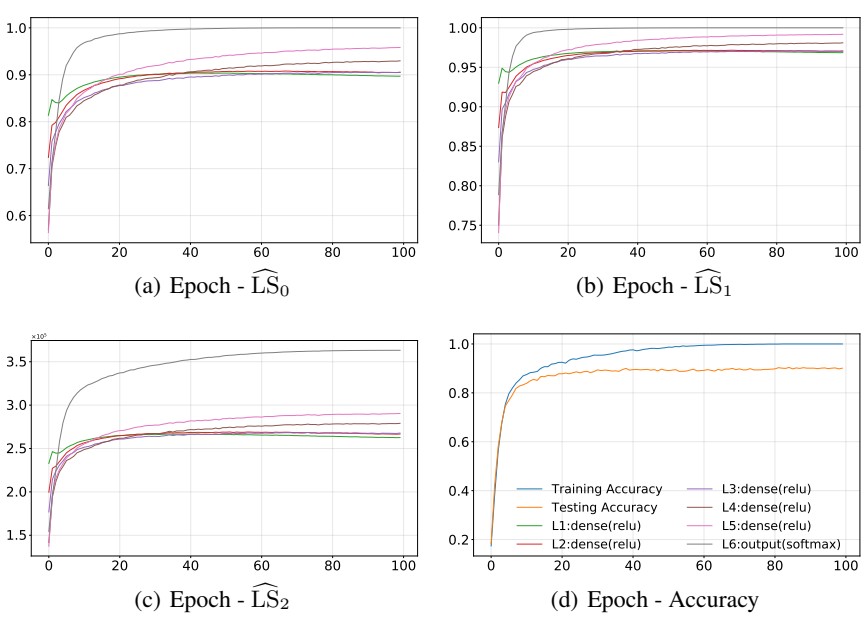

Figure 8: MD-LSM and Accuracy Curves of Hidden Layers of MLP (Ten-Class)

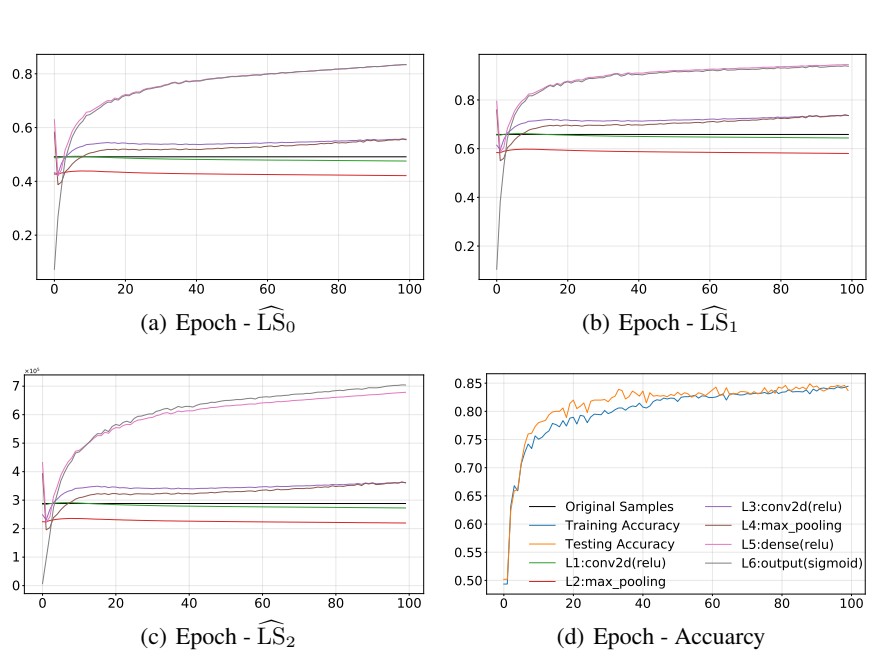

Figure 9: MD-LSM and Accuracy Curves of CNN's Hidden Layers

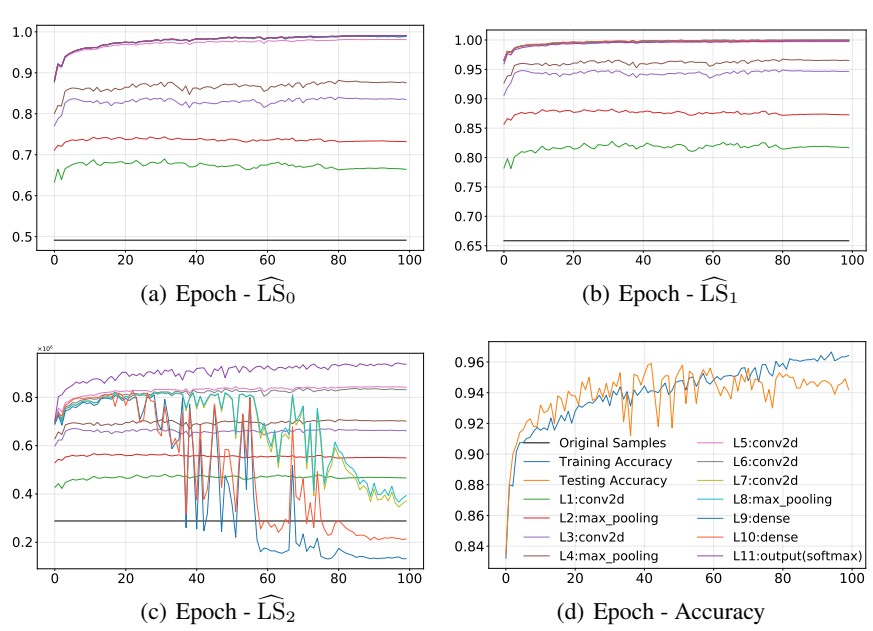

Figure 10: MD-LSM and Accuracy Curves of Hidden Layers of AlexNet (Softmax)

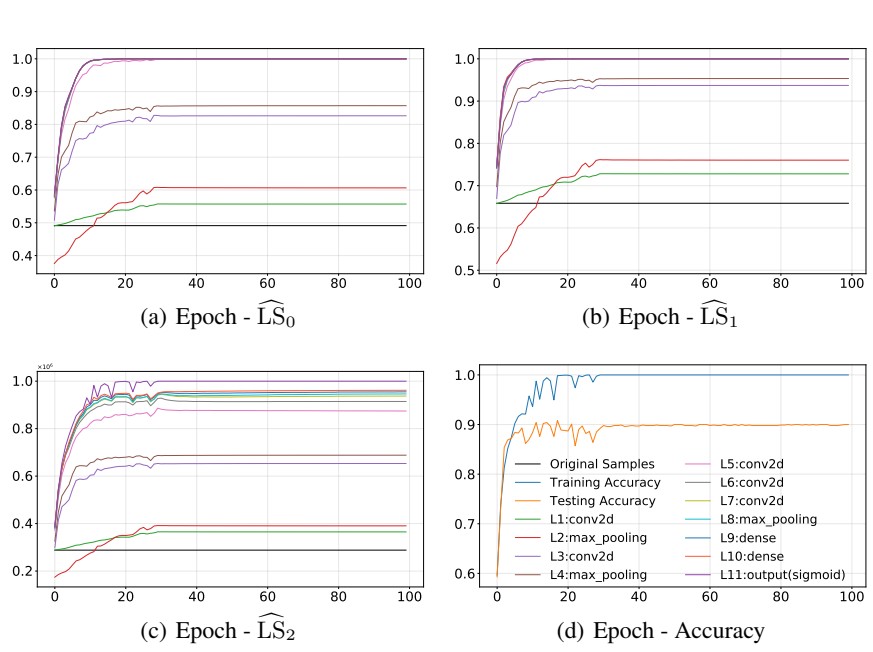

Figure 11: MD-LSM and Accuracy Curves of Hidden Layers of AlexNet (Sigmoid)

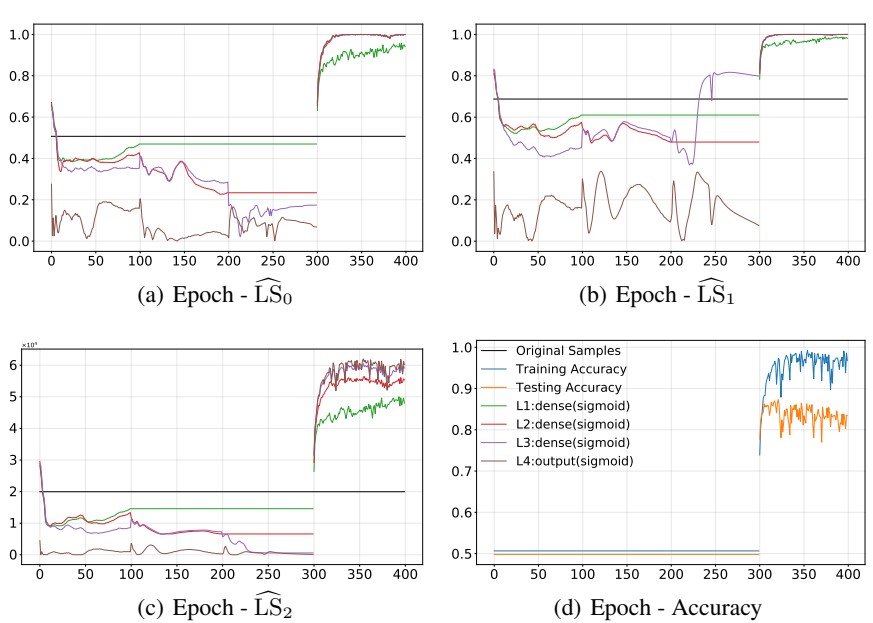

Figure 12: MD-LSM and Accuracy Curves of DBN's Hidden Layers

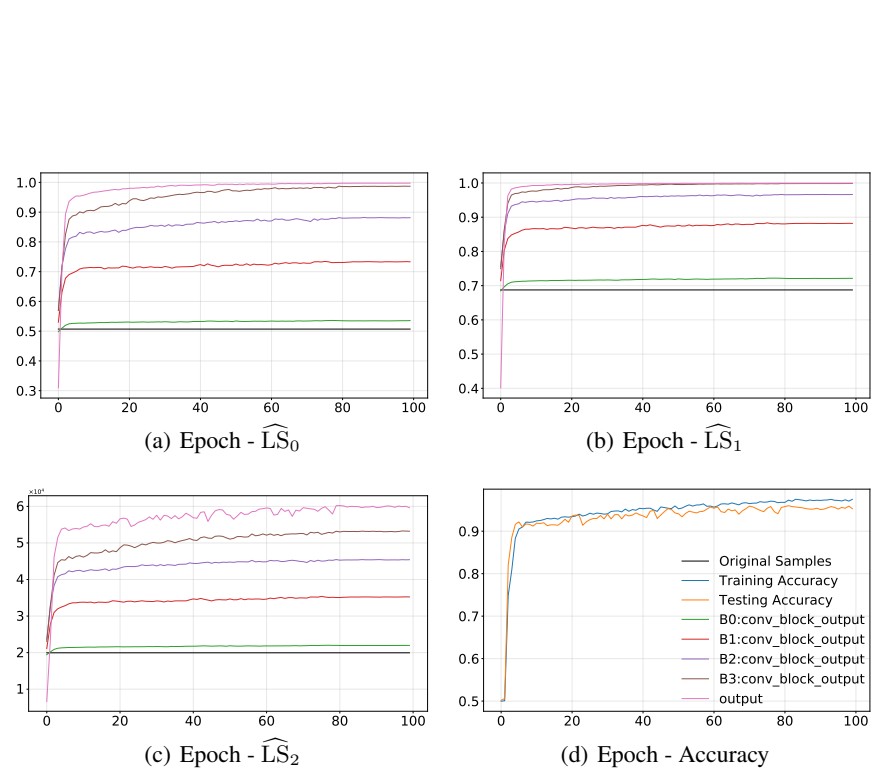

Figure 13: MD-LSM and Accuracy Curves of VGGNet's Main Blocks

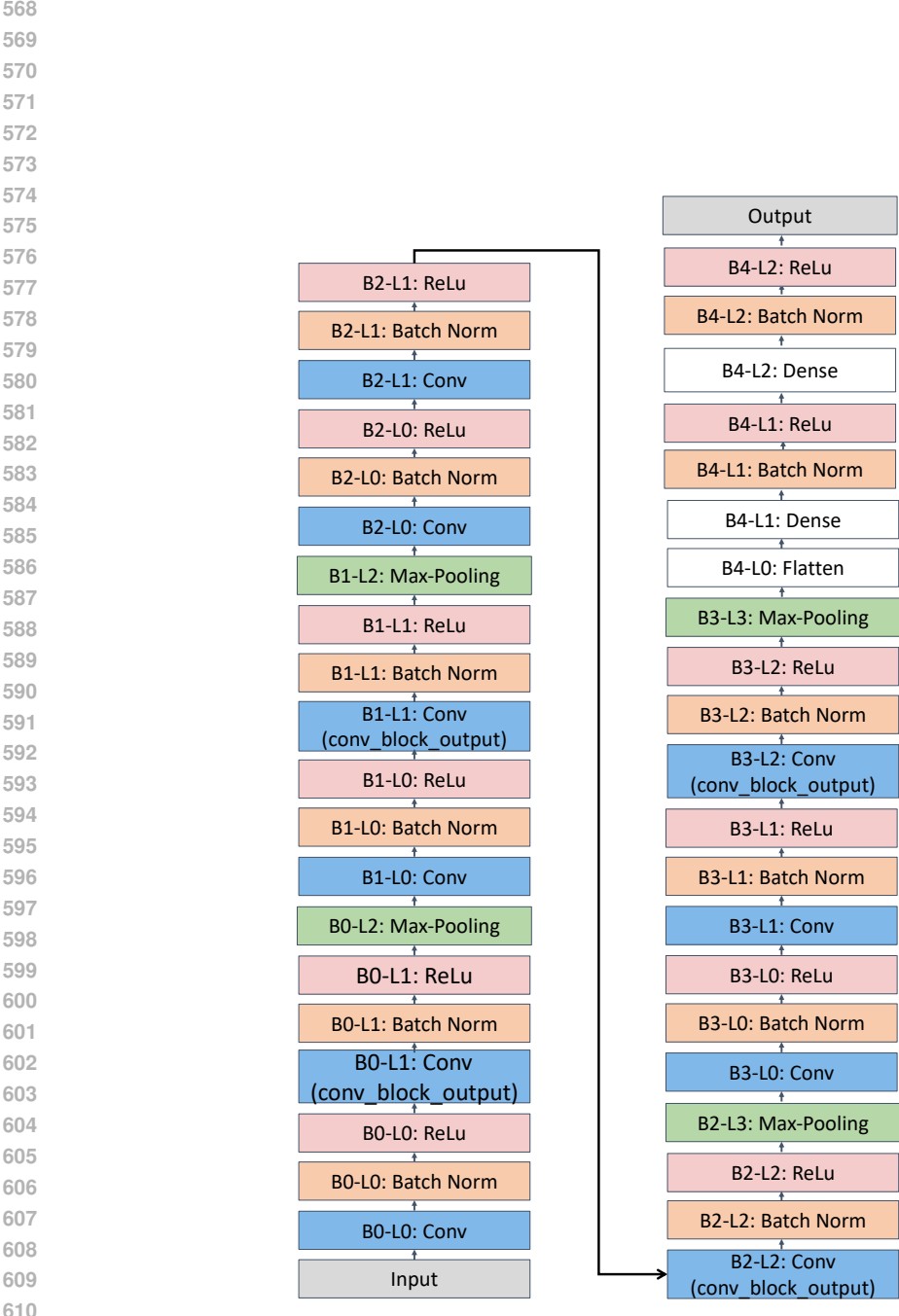

Figure 14: Structure of VGGNet

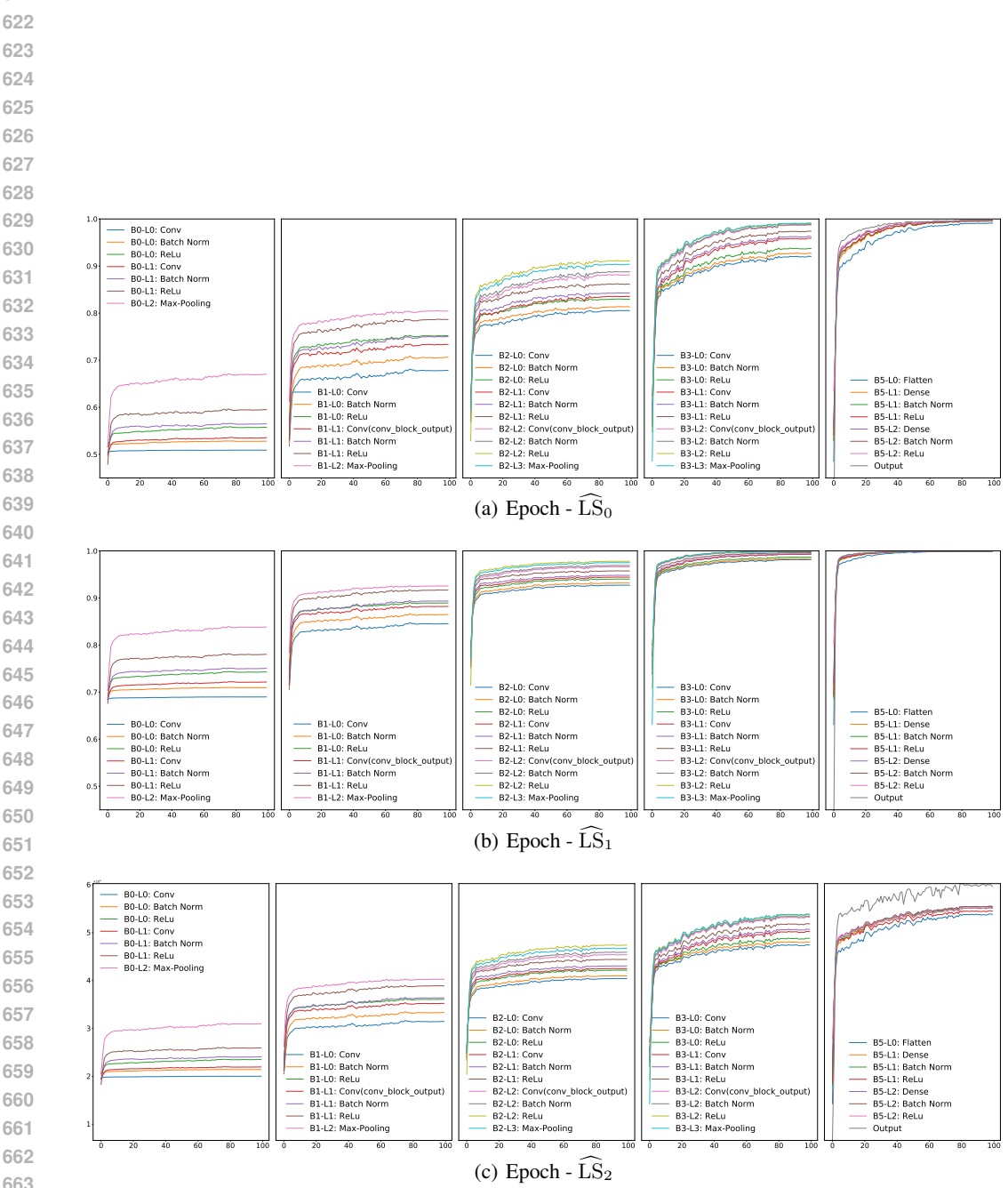

(a) Epoch - $\widehat{\text{LS}}_0$

(b) Epoch - $\widehat{\text{LS}}_1$

(c) Epoch - $\widehat{\text{LS}}_2$

Figure 15: MD-LSM and Accuracy Curves of VGGNet's Hidden Layers

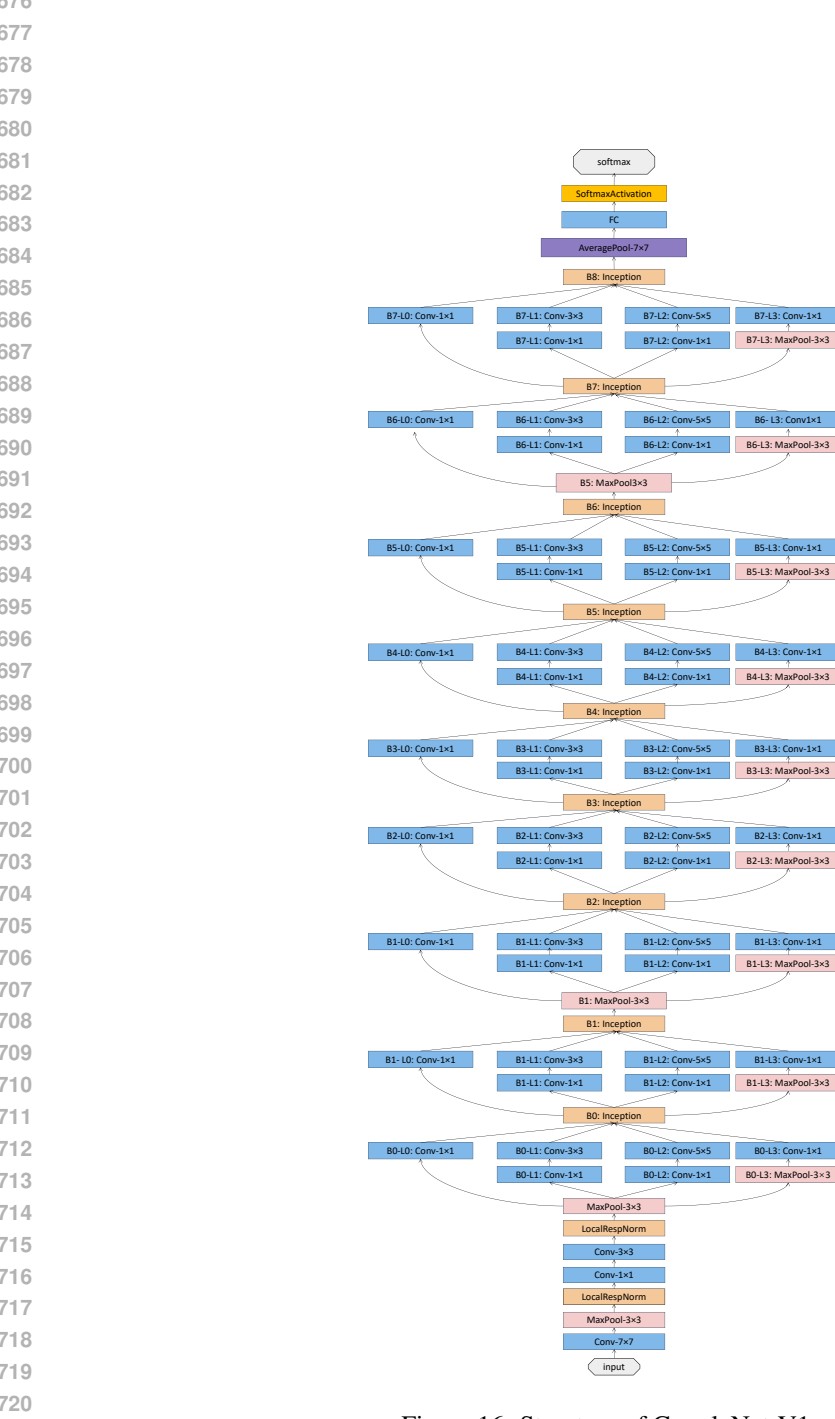

Figure 16: Structure of GoogleNet-V1

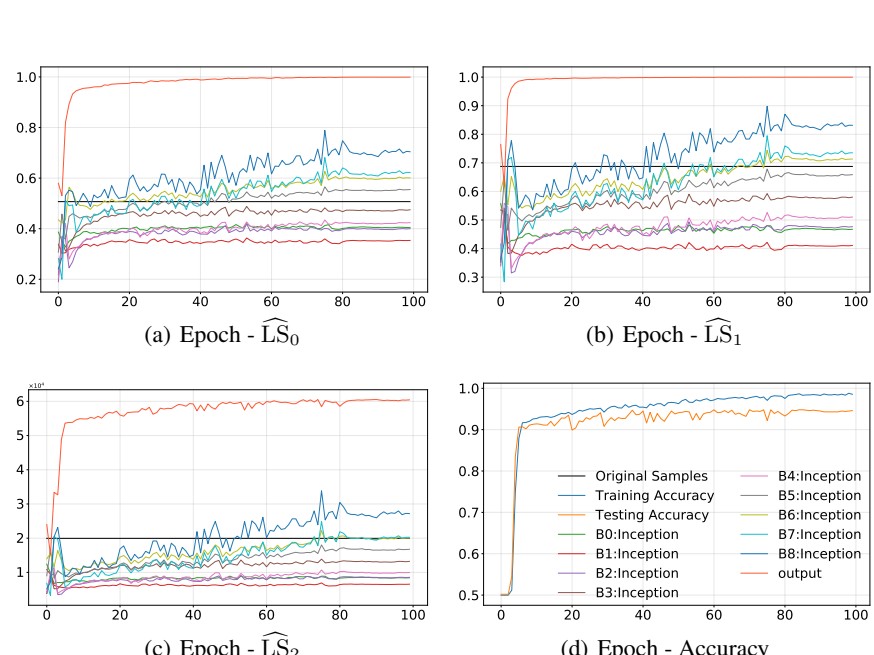

Figure 17: MD-LSM and Accuracy Curves of GoogleNet's Main Blocks

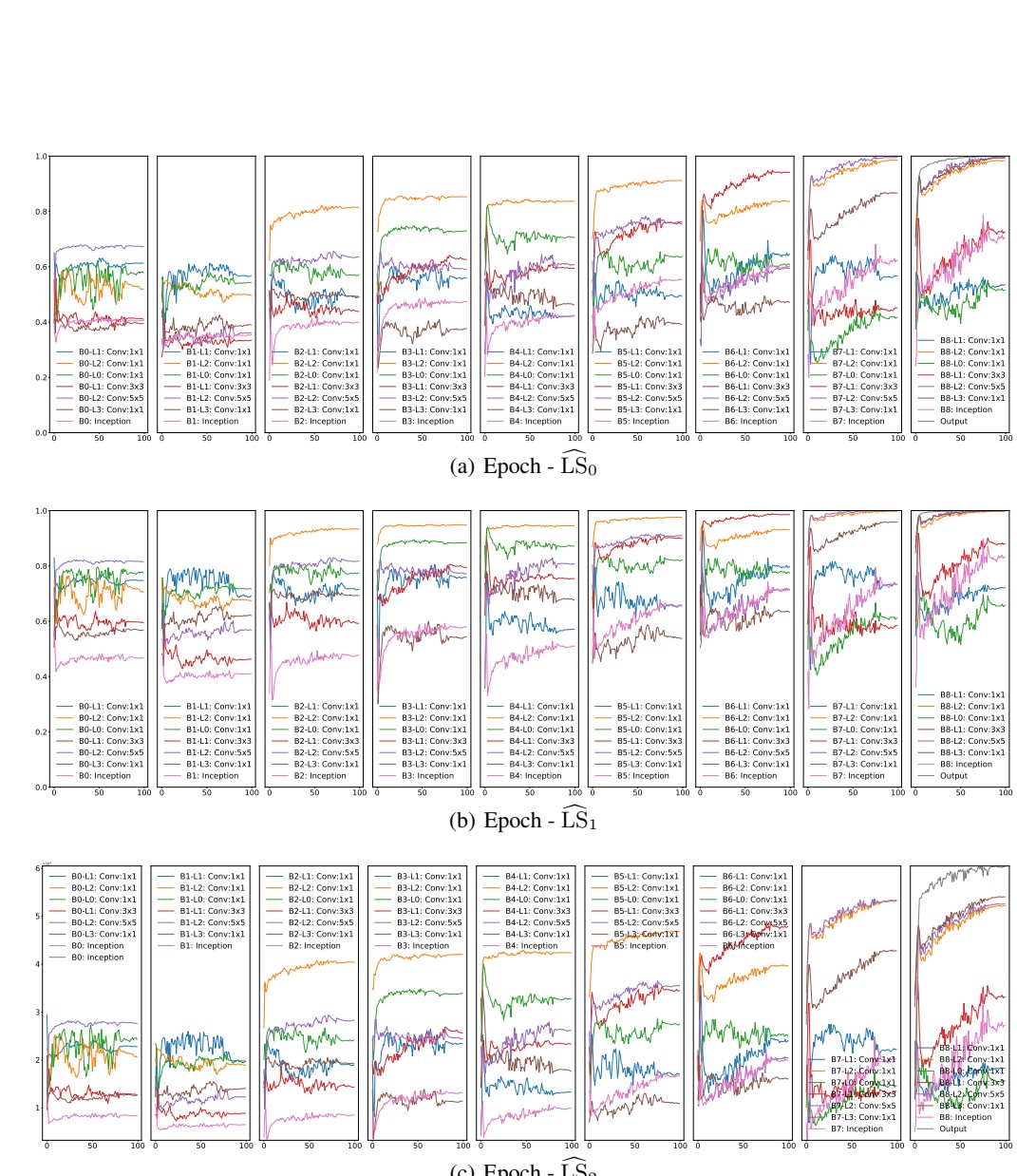

Figure 18: MD-LSM and Accuracy Curves of GoogLeNet's Hidden Layers

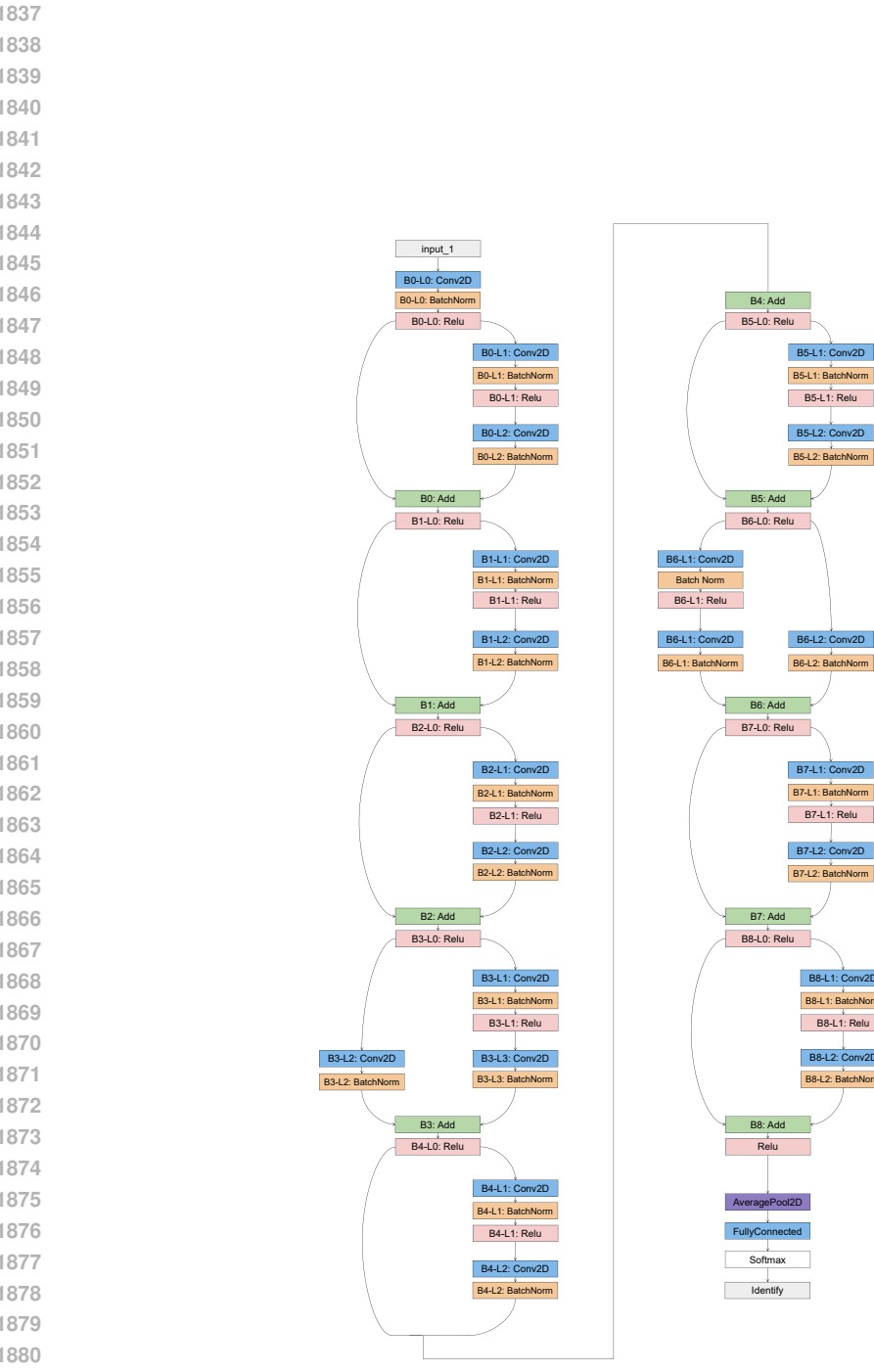

Figure 19: Structure of ResNet-20

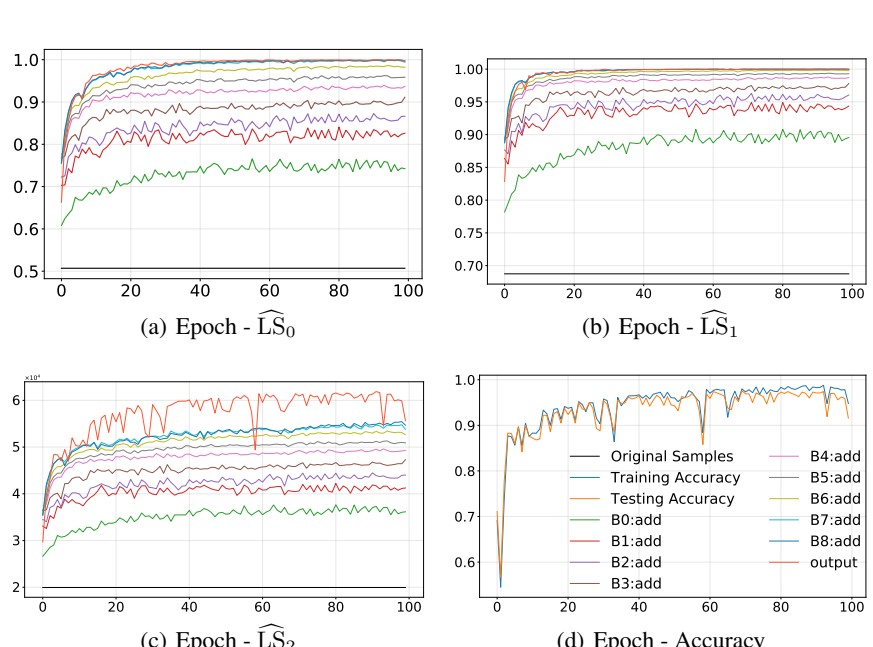

Figure 20: MD-LSM and Accuracy Curves of ResNet's Main Blocks

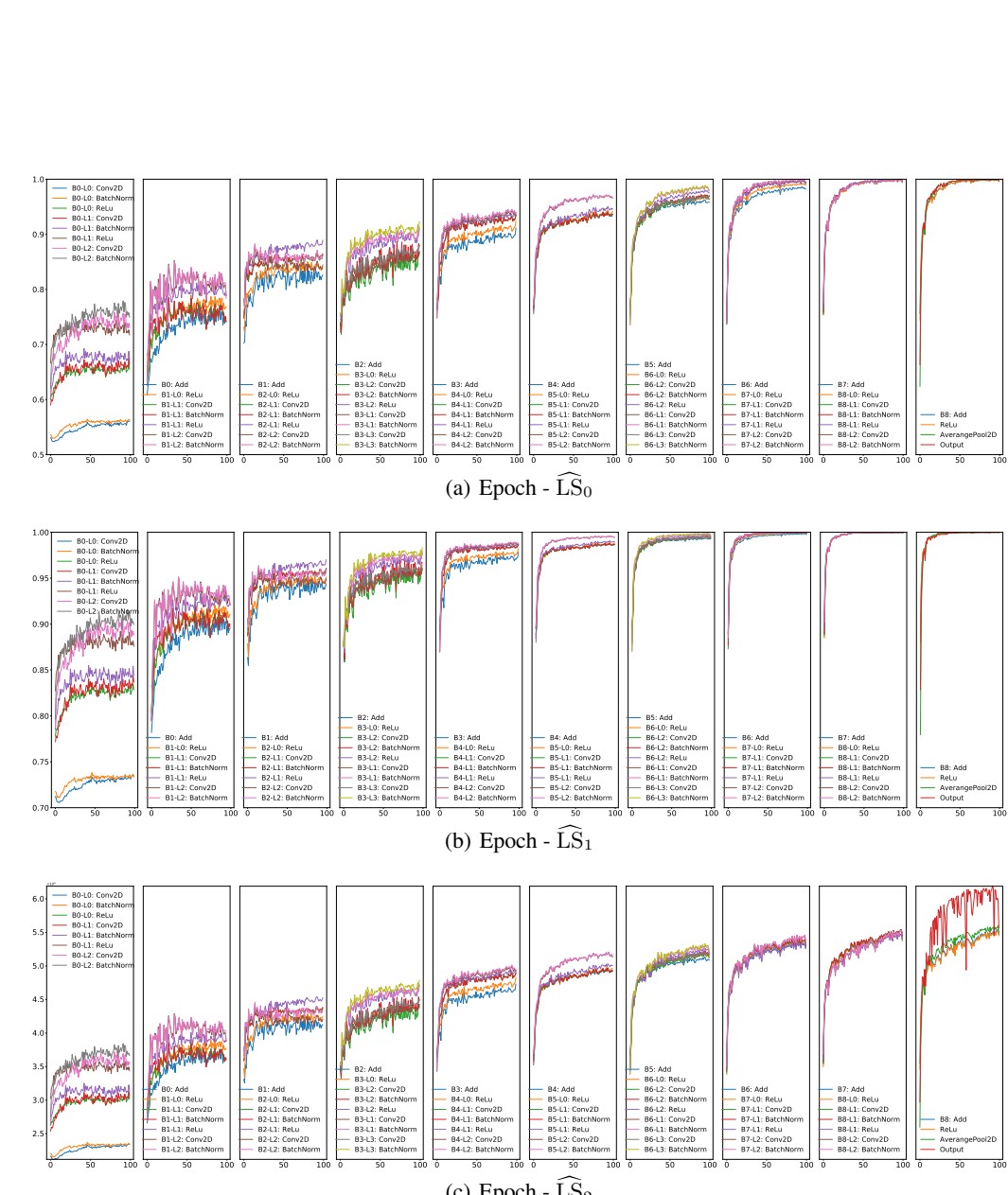

Figure 21: MD-LSM and Accuracy Curves of ResNet's Hidden Layers

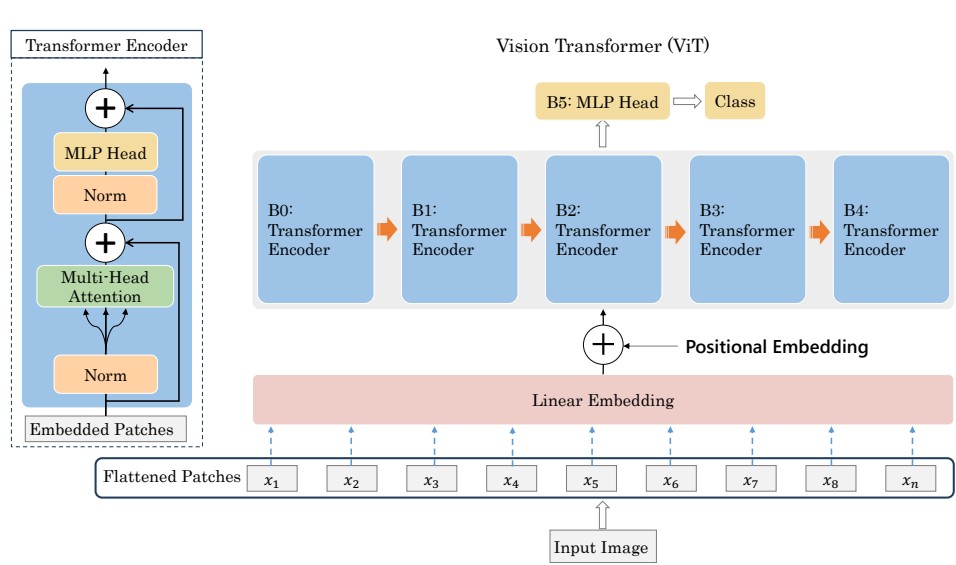

Figure 22: Structure of ViT

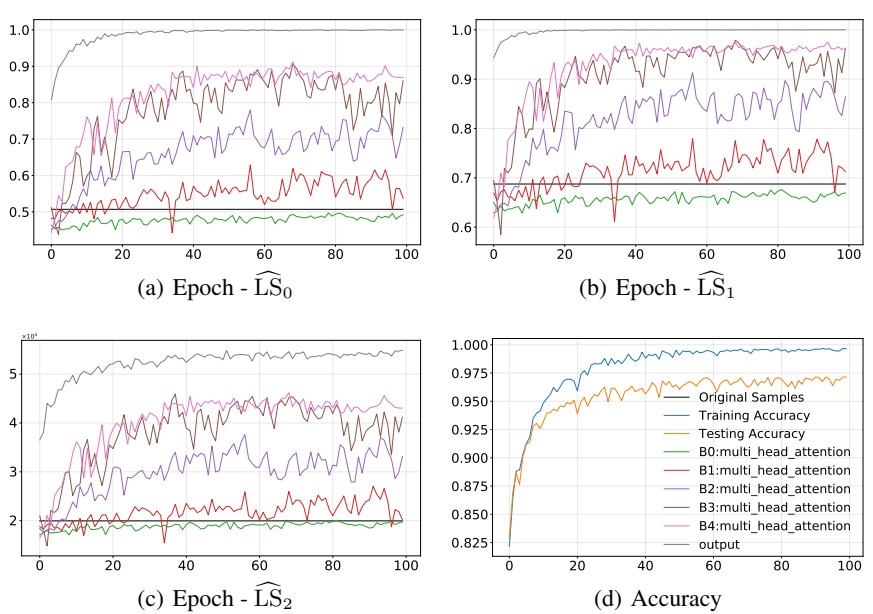

(a) Epoch - $\widehat{\mathrm{LS}}_0$

(b) Epoch - $\widehat{\mathrm{LS}}_1$

(c) Epoch - $\widehat{\mathrm{LS}}_2$

(d) Accuracy

Figure 23: MD-LSM and Accuracy Curves of ViT's Main Blocks

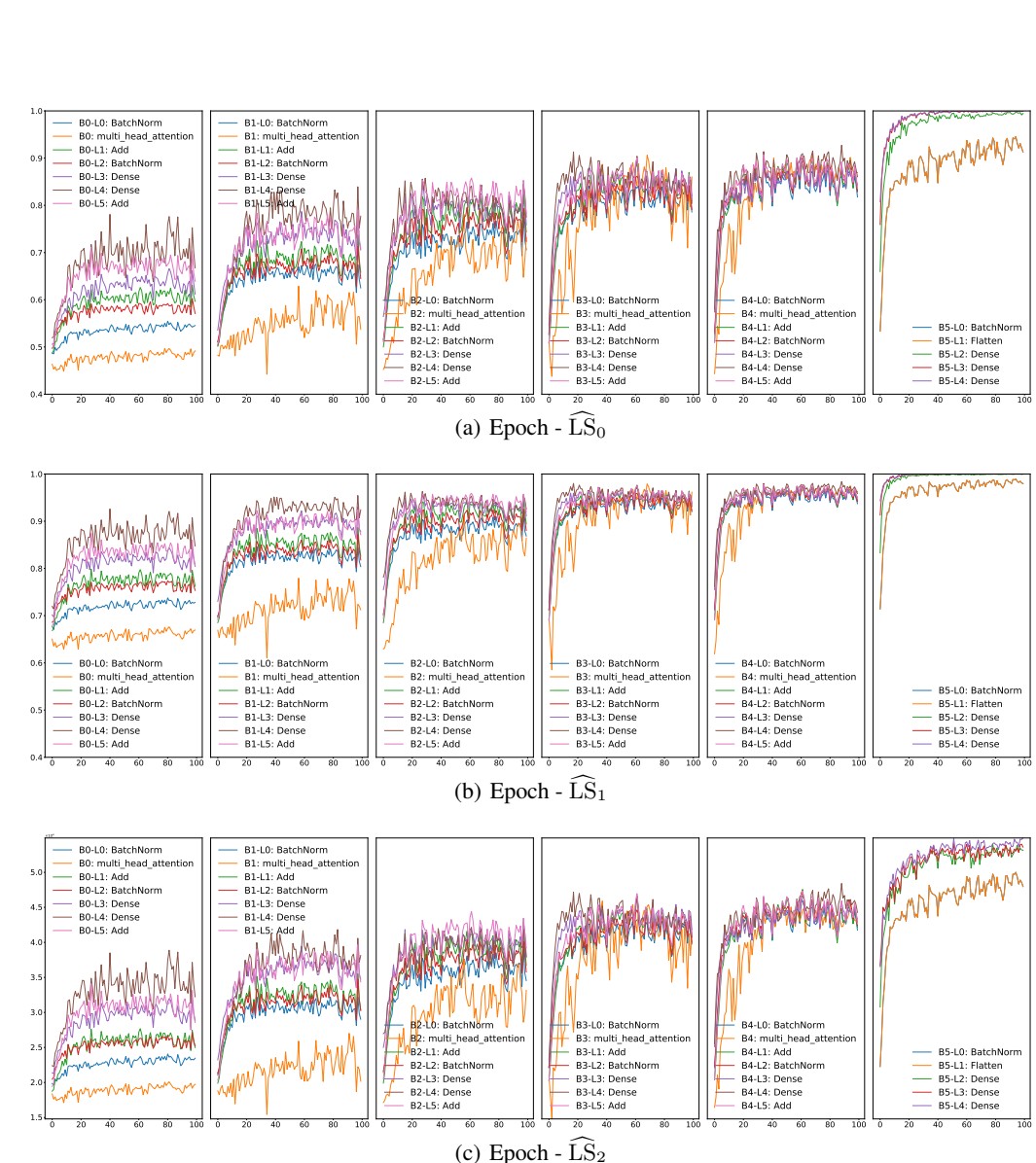

Figure 24: MD-LSM and Accuracy Curves of ViT's Hidden Layers

### E.4 GRAPH NEURAL NETWORKS

Graph neural networks (GNNs) have been successfully employed to deal with the graph-structured data, such as the Cora dataset (Kipf & Welling, 2016). However, there could arise the "over-smoothing" issue in the application of the classical GNN framework, where the discrepancy among the node features tends to become less significant as the network depth increases and thus cause the indistinguishable representations of nodes (Chen et al., 2020; Keriven, 2022). As illustrated in Fig. 11, we find that 1) the discrepancy among the MD-LSM curves of different graph convolutional layers becomes smaller when the number of graph convolutional layers increases; and 2) more interestingly, the linear separability degrees of the layers close to the input layer are higher than those of the layers close to the output layer, *i.e.,* the graph convolutional layers close to the input layer have better node representations. The latter finding is in accordance with the aforementioned "over-smoothing" issue.

Table 11: $\widehat{\text{MultiLS}}_0$, $\widehat{\text{MultiLS}}_1$, $\widehat{\text{MultiLS}}_2$ curves and the accuracy curves during the process of training GNNs with different number of graph convolution layers on the Cora dataset.

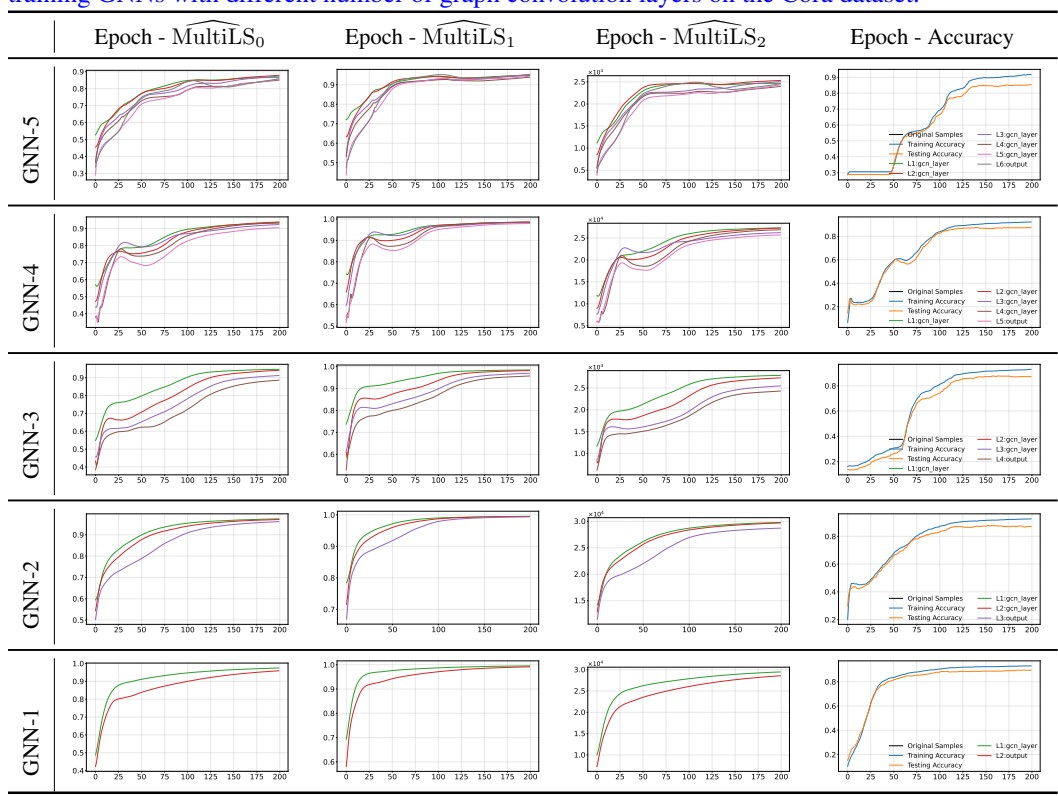

