# OpenReview forum: "MD-LSM: An Efficient Tool for Real-time Monitoring Linear Separability of Hidden-layer Outputs of Deep Networks"
_ICLR.cc/2025/Conference — Submitted to ICLR 2025_

### Official Review · Reviewer_81Eo · 2024-10-24

**Soundness:** 2
**Presentation:** 3
**Contribution:** 1
**Rating:** 3
**Confidence:** 3

**Summary:**

This paper presents a novel approach to analyzing the linear separability of the latent representation of hidden layers in deep neural networks. It introduces the Minkowski difference and analyzes the maximum linearly separable subsets induced by it. The method is validated on multiple architectures on the CIFAR10 dataset.

**Strengths:**

- the background work is clearly reported and explained, with a clear overview of previous works with their pros and cons
 - the proposed method is theoretically grounded, and the explanation in Sec 2.1 is very clear
 - the paper is accompanied by clear and nice plots, showing the different aspects of the approach

**Weaknesses:**

- the paper presents a method that should address the efficiency of the problem, which the authors state is a weak point of all other methods. However, no performance statistic nor computational complexity analysis is carried out. Thus, such “efficiency” cannot be assessed.
 - The method proposes a sound approach for evaluating the linear separability of latent representation learned by neural networks. However, such a sound approach has to be drastically simplified to make it computationally feasible and (supposedly) inexpensive. However, the only analysis carried out to analyse how drastic this approximation is, it’s on low-dimensional, synthetic datasets.
 - at the end of Sec3.2, it is stated that for real-world networks, only a subset of the data is used for computational reasons (only using a batch of 500). This is evidence that the proposed method is not as efficient as promised. Furthermore, a linear probe on 500 samples is almost instant to train using simple and lightweight optimizers like L-BFGS.
 - no supplementary material is provided, limiting reproducibility and independent evaluation of the results.
- remark 2.6 states that the approximate evaluation of the metrics is carried out using $\hat{\omega}$ as maximizer. However, the separating hyperplane obtained from this vector seems to simply be the one connecting the centroids of the two groups. If this is indeed the case, the authors should justify how the method proposed differs from previous literature.

Other concerns:
 - From section 2.2, the math becomes more cumbersome than the previous part. In particular, $|MD(A,B)| = |A \times B| = |A| \cdot |B|$, which is clearer (referring to the last part of page 4).
 - Eq.2 is overly complicated: if you search over any $w \in R^N$, then you don’t need the inner-max since $1(w^Tm<0) = 1(-w^Tm>0)$. Therefore, the $w$ that maximizes the inner equation can just have the opposite sign. In other words, you just need $LS_*(A,B) = max_{w \in R^N} \frac{\sum 1(w^Tm > 0)}{|A| \cdot |B|}$, which is most likely easier to understand.
 - Eq.4 has the same “problem” as Eq.2, meaning that the search over all $w$s already includes the other sign.
 - in Sec 2.3, you go on at approximating the original Eq.2 multiple times, and at the end, you end up with a very heavy simplification, $\hat{w}$, which is still $O(|A|x|B|)$. How is this more “efficient” than linear probing, which has computational complexity $O(np^2+p^2)$ (n samples p features) for regression, or using lbfgs for logistic regression? This is even worse for multi-class, where the OvR approach requires S calculations.
 - in Sec 2.4, “Their largest relative error is less than 3%. This finding supports the effectiveness of the approximate manner” is arguably opinable since this statistic is drawn from a single dataset, which is furthermore synthetic, so it cannot be taken at all as a general statistic of the proposed method approximation error. The same holds for the subsequent comparison.

**Questions:**

Overall, I am open to reconsideration, if new comparison are added from the computation perspective, mainly showing how this method is either better than probing given the same time budget, or that given equally good results, it takes less time, otherwise it seems that the contribution on the “efficiency” (which is the main difference to probing) cannot be assessed.

---

> ### Author Response · Authors · 2024-11-22
> **Reply to Reviewer # 81Eo**
>
> Thanks for your valuable comments on this paper. The following have listed the major comments as well as our responses to them:
>
> **(1) RE: no supplementary material is provided, limiting reproducibility and independent evaluation of the results.**
>
> **Ans**: Thanks. In the first submission round, we have uploaded the source code of this paper as a supplementary material. Since the revised version contains some new experiments, we have updated the code accordingly. Please check it.
>
> **(2) RE: This statistic is drawn from a single dataset, which is furthermore synthetic, so it cannot be taken at all as a general statistic of the proposed method approximation error. The same holds for the subsequent comparison.**
>
> **Ans**: Thanks.  In the revised version, we have added new experiments on **the UCI datasets (including Diagnostic, Ionosphere, Maintenance, and Marketing) and the text datasets (including Cora and IMDB)** to examine the effectiveness of the proposed MD-LSMs and meanwhile to explore the hidden-layer behaviors of deep networks on different types of classification tasks. Please refer to **Table 2, Appendix E.1, E.2 and E.4** of the revised version for the new experimental results.
>
> **(3) Overall, I am open to reconsideration, if new comparison are added from the computation perspective, mainly showing how this method is either better than probing given the same time budget, or that given equally good results, it takes less time, otherwise it seems that the contribution on the "efficiency" (which is the main difference to probing) cannot be assessed.**
>
> **Ans**: Thanks. Although some state-of-the-art classifiers can achieve lower computational complexity, for example, the desired computational complexity of logistic regression is $O((I+J)\times p)$ that is lower than the complexity $O(I \times J \times p)$ of the approximate manner of calculating MD-LSMs, where p is the data dimension, I and J are the sample sizes of ${\cal A}$ and ${\cal B}$, respectively.
>
> However, it is noteworthy that such a low complexity is achieved under the assumption that ${\cal A}$ and ${\cal B}$ are linearly separable. If there is no priori knowledge on the data distribution, it becomes hard to achieve the desired computational complexity, because it is difficult to find a reasonable training termination condition and appropriate optimization hyperparameters (such as the initial weights and the learning rate). Similar things could happen with other classifiers in the scenario of real-time monitoring hidden-layer behaviors of deep networks during their training process.
>
>
> **In the revised version, we have added the following sentences to highlight this fact:**
>
> >*“\…\...Some state-of-the-art classifiers (such as logistic regression or naive Bayes) have the potential to act as feasible “probes” because of their low desired computational complexities. However, if there is no priori knowledge on the data distribution, the efficiency and the performance of these classifiers could be heavily influenced by some unavoidable factors such as the choice of hyperparameters and the setting of termination conditions, and thus their desired complexities are usually hard to be achieved in practice. Consequently, the “probe” method is unsuitable (at least cannot be directly applied) to detecting the mapping behavior of each hidden layer after each training epoch.\...\...”*
>
> Moreover, we have also examine the time costs of implement different LSMs on the UCI datasets, and experimental results show that the proposed approximate manner of calculating MD-LSMs has a high efficiency with only a slight precision sacrifice (see Table 2 of the revised version).
>
> To sum up, the computational complexity of approximately calculating MD-LSMs is higher than the desired computational complexities of some classical classifiers such as the logistic regression. However, the computational complexity of MD-LSM is fixed for arbitrary kind of dataset regardless of its priori knowledge on the data distribution. In contrast, the desired computational complexity of these classifiers are achieved under some specific assumptions (*e.g.,* the dataset is linear separable or the data distribution is known). Unfortunately, these requirements cannot be met in the scenario of real-time monitoring hidden-layer behavior of deep networks. The efficiency of implememting these classifiers could be heavily influenced by the choice of hyperparameters and the setting of termination condition. **Thus, I think the method proposed in this paper a really \"efficient\" one for the real-time monitoring of hidden-layer behavior of deep network especially for the future study on large models.**
>
> **If this reply has answered your question about the \"efficiency\" of our method, please reconsider the score of this paper.**

---

> > ### Comment · Reviewer_81Eo · 2024-11-28
> > **Reply**
> >
> > Thanks to the authors for the reply. However, multiple questions and weaknesses have been ingore or the response is evasive.
> >  - W1 is supposedly been addressed by Table 2. However, it is not clear why they have choosen LDA over Logistic regression, as in their related work they clearly state that linea probing is the one they are trying to improve over, so it’s not addressed.
> >  - W2 has been sort of addressed, but the results are arguably against the claim. Table 7 and Table 8 show that $\hat{LS}_{0/1/2}$ return clearly very different results. Furthermore, the questioned approximation accuracy was between the $LS_i$ and $\hat{LS}_i$, and the first one are. Table 2 shows results, but never compares them, the table should contain some summary columns/row showing the overhead/saving in time or performances, to better address if the proposed method improves over LDA. Once this is addressed, I believe the weakness is resolved, however, I still believe that “the competitor” of the proposed method is linear probing, and not LDA.
> >  - W3 has been completely ignored.
> >  - W4 has been addressed.
> >  - W5 has been completely ignored, and it is fundamental to address the contribution of the paper. I believe that it can be trivially shown that the method is extremely related to just finding the centroids of the classes and the vector connecting them, and if that’s the case, the contribution of the paper is marginal if not absent from a theoretical point of view.
> > Regarding the other concerns:
> >  - C1 has been addressed.
> >  - C2 has been addressed.
> >  - C3 has been addressed.
> >  - C4 has not been addressed. The authors states that _“To sum up, the computational complexity of approximately calculating MD-LSMs is higher than the desired computational complexities of some classical classifiers such as the logistic regression. However, the computational complexity of MD-LSM is fixed for arbitrary kind of dataset regardless of its priori knowledge on the data distribution. In contrast, the desired computational complexity of these classifiers are achieved under some specific assumptions”_. However, I not sure how this claim is backed, and what the authors mean with “under some specific assumptions” (Logistic Regression loss is convex, with minimal L2 regularization it becomes strongly convex, ensuring convergence, and distance to the optima). The proposed approach is the result of 3 (heavy) approximations, so I’m pretty sure that for almost all cases, just few steps of LBFGS takes the LG to a close enough solution. However, this cannot be said since the authors compare their work to LDA, and not linear probing.
> >  - C5 is highly related to W2, so follows that answer
> >
> > I’m still not convinced that the paper is ready for publication, and I stand by my current rating, with the major reason being that linear probing, while being the true competitor, as also shown in Table 1, is completely discarded on the empirical side. Furthermore, it should be noted that probably a probe trained using GD together with the network, has almost no overhead

---

> ### Author Response · Authors · 2024-11-30
> **Response the comment of Reviewer  # 81Eo (Round 2)**
>
> Thanks for your reply to our response. Regarding your initial feedback, we had prepared a response letter for the point-by-point response to each of your previous comments. However, due to the word limitation, we were unable to present the full responses in the previous round. Please refer to the link [[https://iclr4139.notion.site/pbp](https://iclr4139.notion.site/pbp?pvs=4)] for the point-by-point response that had been prepared in the previous round. Next, we come up with your current comments.
>
> **[RE W1 and C4]:** Thanks. In the current version, we did not consider logistic regression as the Probe is because it is difficult to choose appropriate learning rate and termination condition for implementing gradient method. More specifically, although a big learning rate could decrease the training iteration, it is possible to cause oscillations near the extreme point. On the contrary, a small learning rate is able to maintain the training stability but the number of iteration is high (i.e., the computation cost is high). Thus, the computational complexity O(np) of logistic regression is indeed a desired one in theory, and it is hard to be achieved in practice.
>
> Although there have been some works on the convergency analysis of training logistic regression, it is still challenging to design a fully-automatic learning rate selection method in practice. Refer to [[https://iclr4139.notion.site/ref](https://iclr4139.notion.site/ref?pvs=4)] for these references. Instead, the GRQ of LDA can be directly calculated without any hyperparameter, and thus the comparison among MD-LSM, GQR and GDV is fairer than that between MD-LSM and logistic regression-based probe.
>
> Next, we also conduct the numerical experiments on this issue. Please refer to the link [[https://iclr4139.notion.site/exp](https://iclr4139.notion.site/exp?pvs=4)] for the experimental results. The code on the new experiment is also provided in this webpage. Please check it.
>
> The experimental results show that 1) the computational cost of logistic regression-based probe is much higher than MD-LSMs in the actual task of real-time monitoring hidden-layer behavior; and 2) the process of training logistic regression is not always stable. Moreover, it should also be noted that if we use the classification accuracy of the logistic regression-based probe as the linear separability measure, it could not provide the well-defined form for the further theoretical analysis.
>
> **[RE W2 and C5]:** This comment will be replied from the following aspects:
>
> - (1) The experiments for Table 7 are conducted on the four UCI dataset; but the experiments for Table 8 are conducted on the CIFAR-10 dataset that is the one used in the experiments of Fig. 2. Due to different datasets, the results in the two tables are certainly different from each other. I am sorry that we forget to highlight the dataset information in Tab. 8, and we will update it in the next version.
> - (2 ) I am sorry that I do not get your point in this sentence "Furthermore, the questioned approximation accuracy was between the $LS_i$ and $\hat{LS}_i$, and the first one are. " Could you clarify it for us? and then we will make the clear response.
> - (3) According to your suggestion, we have rearranged the content of Table 2, and added the individual columns to show the overhead/saving in time. Please see the link [[https://iclr4139.notion.site/table](https://iclr4139.notion.site/table?pvs=4)].
> - (4) To make the comparison between MD-LSM and logistic regression, we have conducted the new experiments. Please refer to [[https://iclr4139.notion.site/exp](https://iclr4139.notion.site/exp?pvs=4)] for the detailed experimental results and their discussion.
>
> **[RE W3]:** I am very sorry that you though that we've completely ignored your W3 comment. In the previous rebuttal round, we had been aware of the W3 comment is the most important one of all your comments. Because of the word limitation, we cannot make the point-by-point response to your comment, and thus we had to integrate some similar comments into one response item.
>
> The previous response of Item (3) is just for this comment, and it had taken up most of the word budget. In addtition, we have conducted new experiments on the comparison between MD-LSM and logistic regression as well as the discission on why the MD-LSM is more suitable than logistic regression (see [[https://iclr4139.notion.site/exp](https://iclr4139.notion.site/exp?pvs=4)]).
>
> **[RE W5]:** I am sorry that due to the word limitation, it is hard for us to explain this issue in details. Actually, in the previous rebuttal round, we have prepared a detailed response to each of your comment. In fact, the question in this comment has been answered in Appendix B of this paper. The detailed reponse to this comment is referred to Item # (5) of the link [[https://iclr4139.notion.site/pbp](https://iclr4139.notion.site/pbp?pvs=4)] for the point-by-point response that had been prepared in the previous round.

---

> > ### Comment · Reviewer_81Eo · 2024-12-01
> > **Reply**
> >
> > Dear authors,
> >
> > Thanks for the reply. However, I feel that any results not reported in the main paper, should not be considered, as it’s not part of the peer-reviewed material. Furthermore, I personally find any claim made outside of the main rebuttal page on OpenReview main page should also not be considered, as it won’t be part of the final evaluation of the paper. For this reason, even though I thank the authors for the additional work carried out, I won’t consider it for evaluating the work.
> >
> > However, in response to your points:
> > W1&C4: even though I stand by the fact that tuning a Logistic Regression is almost trivial, if obtaining that bound “it is hard to be achieved in practice”, then the comparison should be carried out for a fixed time budget instead of the results, to compare if, with similar time, the proposed approach improves over log-reg.
> > W2&C5:
> >  - thanks for the clarification, however, my point was regarding the columns of these tables, not the tables themself. Given the same table, the reported results regarding LS_i look pretty different, showcasing how they are not actually equivalent.
> >  - we are trying to understand the difference between approximation and theoretical true optima. Thus, if the true $w$ and $\hat{w}$ gives similar results  (equivalently, if $LS1$ is similar to $\hat{LS1}$, and not if $LS1$ and $LS2$ give similar result between each other)
> > W3: OpenReview allows for authors to post more than a single comment at a time
> > W5: It’s not clear how the approach is different to a centroid analysis. I still find that $\hat{w}$ is at most off by a constant wrt LDA, which is irrelevant since we want to find $w^Tm>/<0$ Furthermore, the reported formula in the Notion page (even though I won’t consider it part of the rebuttal), looks much computationally cheaper than the one reported in the paper since it is $O(A+B)$ and not $O(A\cdot B)$
> > Overall, I am grateful to the authors for the additional work, but as stated at the beginning, I don’t feel that such external material should be considered. Furthermore, the code looks reasonably simple, so it’s not clear why the comparison was not included since the first rebuttal comment in the main paper, ideally even just in the appendix. For this reason, I will stand by my evaluation, and given the limited remaining time, and the impossibility to revise the paper, I don’t think there is room for change of such score, as the two main points (analysis wrt centroids and comparison to log-reg) should be part of the main work.

---

> > > ### Author Response · Authors · 2024-12-02
> > > **Reponse to your technical comments**
> > >
> > > **The following is the response to your technical concerns in the current comments.**
> > >
> > > In the revised version, the results of $LS_i$ and $\hat{LS_i}$ reported in Table 2 are merely presented as a comparison to highlight the difficulty of solving the optimization problem. The $LS_i$ results were obtained by controlling the optimization time of differential evolution to around 10 to 20 seconds (approximately ten times the computation time of $\hat{LS_i}$). If the optimization process were to continue indefinitely, the $LS_i$ would be larger than $\hat{LS_i}$, indicating that a better $w_opt$ was found. However, such an optimization process with massive iterations (regardless of the cost) is significantly time-consuming, and thus the setting of our numerical experiment is fully in line with the actual technical requirements of the real-time monitoring hidden-layer behaviors.
> > >
> > > In addition, your insistence that the $\hat{w_{lg}}$ resulted from logistic regression that achieves higher classification accuracy on the target binary classification dataset compared to the approximate $\hat{w}$ resulted from using the simplified $LS_2$ is understandable. We have a question to ask you: “during the training process, if each hidden layer uses a $w_{best}$ with higher classification accuracy, does the resultant MD-LSM curve remain stable?” Our approximate $\hat{w}$, derived from the simplified version of optimization-based $LS_2$ , does match with the proposed $LS_i (i=0,1,2)$ better, and meanwhile inherits their intrinsic characteristics.
> > >
> > > In contrast, the $\hat{w_{lg}}$ obtained using logistic regression inherently causes instability due to the factors such as learning rates and termination conditions. Moreover, the $\hat{w_{lg}}$ is not compatible with the optimization-based $LS_i (i=0,1,2)$ proposed in this paper. As we emphasized in our notion page, logistic regression is highly unstable and thus unsuitable to replacing $\hat{w}$ proposed in this paper with $\hat{w_{lg}}$ when solving $LS_i (i=0,1,2)$.
> > >
> > > To sum up, your insistence on logistic regression is meaningless for the main research concern of this paper, that is, the real-time monitoring hidden-layer behaviors. In this paper, we analyze the training behavior of neural network hidden layers, and traditional methods have been thoroughly compared. We still believe that these methods are highly relevant to this study.

---

> ### Author Response · Authors · 2024-12-01
>
> Dear reviewer,
>
> Thanks for your reply. First, we are pleased that our new experiments have clarified your misundersandings on this paper at the eleventh hour, but it is regrettable that you reject to reconsider the score of this paper.  **It is unfair for a lot of efforts, made by all of authors, to carefully reply each of your comments. It  was also against the intention of the rebuttal phase.**
>
> Frist, it should be noted that, in Table 1 of the initial version, we have stated that the efficiency of linear "probe" cannot satisfy the technical requirment of the task of real-time minitoring hidden-layer behavior because this type of method needs the training process. **It is a common knowledge that most of classifiers require the training processes that are time-consuming becasue of the selection of hyperparameters and the setting of termination conditions**, especially for the state-of-the-art classifiers (e.g. SVM, logistic regression and random forest). Thus, in the initial version, we did not take so many words on this point, due to the page limitation.
>
> In the first rebuttal round, according to your suggestion, we have added an individualy paragraph to highlight this fact **(see Line 123 --- 128 of the revised version)**. The response was submit on **November 22**. Regretably, you still have doubts on this point, but your new comments appeared on **November 28th**, which has been beyond the date **("November 27th: Last day that authors may upload a revised PDF.")** of the last time we can upload the revision version. **We have lost the last chance to improve this paper according to your suggestion.** Therefore, we were only able to report our experimental results on your concern (i.e., the comparision between logistic regression and the proposed MD-LSMs) via anonymous notion links.
>
> At the end of this response, we are very grateful for your careful reading and valuable comments. But, we again hope you to reconsider the score of this paper, **because we have already clarified your main misunderstandings and questions on this paper.**

---

### Official Review · Reviewer_7sGD · 2024-10-31

**Soundness:** 2
**Presentation:** 3
**Contribution:** 3
**Rating:** 5
**Confidence:** 5

**Summary:**

The primary goal of the paper is to develop a computationally efficient method to measure the linear separability of each hidden-layer output in deep networks. The MD-LSM is specifically designed to be an absolute measure, robust, computationally inexpensive, insensitive to outliers and able to perform real-time monitoring.
Comparison with SoTA such as GDV, GRQ, Structural manner (Hidden classification layer) (Table 1); Proposed method provides a potential for improving model interpretability; Formal definitions and theorem-based explanations; Adaptibility to different network structures (VGG-16, GoogleNet Inception V1, ViT); Detailed algorithmic flows and extensive experimental report.
the authors have relied on simple activation functions and less number of layers. It will also be extremely interesting to work on a generalized solution that can address the non linear transformations as well. The paper is well-written and introduces an intriguing theoretical concept supported by rigorous proofs. However, additional experimentation is necessary to validate the findings further. Lines 489 to 498 highlight some of the identified limitations, and the authors plan to address these in future work, which will enhance the overall completeness of the paper.

**Strengths:**

The paper is well written with a precise structure. It forms a theoretical foundation for the proposed Minkowski distance based linear separability measure. The supplementary material is comprehensive, containing rigorous mathematical proofs and detailed experimental reporting, with implementations across various deep network architectures.

**Weaknesses:**

However, the work lacks a thorough experimental analysis, particularly in addressing practical implementation challenges. A notable limitation is the absence of discussion on real-time monitoring difficulties in large networks, with no proposed solutions including techniques such as scaling or dimensionality reduction.
The choice of Minkowski difference, while computationally advantageous for calculating linear separability, lacks clear theoretical justification in the paper. This approach focuses on pairwise differences and may overlook global distributional features. Other approaches may involve using kernel-based methods (making it effective for both linear and nonlinear separability), Wasserstein distance, contrastive loss-based measures, etc. which cater to a larger problem. The paper's Theorem 2.2 makes the questionable assumption that linear separability can always be represented by a hyperplane through the Minkowski difference, failing to account for scenarios with overlapping class boundaries or even multiple hyperplanes. Furthermore, the strict emphasis on linear separability neglects the possibility of non-linear relationships in complex data distributions, which are prevalent in most real-world datasets. This limitation becomes particularly apparent when dealing with specialized data types like point cloud data, where higher dimensional patterns can be easily missed, highlighting the method's constraints in handling more complex data structures.
The theoretical framework presented here aims at establishing an absolute measure yet shifts towards an approximation of MD-LSM in practical implementations, effectively making it a relative metric. This inconsistency between the theoretical foundation and the practical application raises questions over the fundamental purpose of the measure. While the paper explores this relative approach, it overlooks well-established methods like KL divergence (Kullback and Leibler, 1951), which has demonstrated significant success in tracking distributional shifts over time and has proven particularly valuable for real-time model monitoring applications. Additionally, despite the approximations introduced in practical implementations, the computational burden remains substantial, suggesting that the proposed method may not offer significant advantages over existing approaches in terms of computational efficiency.
The comparative analysis between the original MD-LSM and its approximation is only provided with respect to the dataset distribution and not with respect to changing model architectures. It is not feasible to be used in large scale models as Line 135 states that the approximation values slightly differ from original ones, and this error could massively increase with increasing layers as in the case of complicated classification tasks, that include composition of multiple hidden layers (Line 148-150). The experimental scope is notably restricted, with ViT being the only architecture incorporating attention mechanisms. This limited testing raises questions about the method's applicability to more complex and commonly used models, particularly in NLP tasks where heavier attention-based architectures are standard practice. The absence of experiments with these contemporary, computationally intensive models leaves a critical gap in understanding the method's scalability and reliability across the full spectrum of modern deep learning applications.
The paper's claim regarding outlier insensitivity lacks experimental validation, presenting it as an apparent fact without proper empirical support. A detailed experimental analysis on anomalous and imbalanced datasets will help create a clear picture. Furthermore, the MD-LSM implementation uses the one-vs-rest strategy to handle multi-class classification tasks which also has many drawbacks. This approach is likely to miss significant inter-class relations and may also lead to interpretation issues, especially when dealing with multi class problems.
Empirical analysis of proofs in supplementary section is missing. Assumptions such as 𝑀.𝑀𝑇 = 𝐼, vastly reduce the generalizability of the methods, as it does not hold for data with high correlation or redundancy. There is also a lack of statistical analysis to clearly understand the robustness and consistency of MD-LSM across different network architectures and layers.
There is a synchronicity between training accuracy and MD-LSM, which is ironic as training accuracy is a relative measure. A model may exhibit high training accuracy, but poor generalization as seen in the case of overfitting. Therefore, if such synchronicity exists, an elaborate experimental discussion along with the extensive report will be interesting and may further lead to a better perspective. Understanding this relationship could lead to:
1. Better model evaluation techniques
2. More effective training monitoring
3. Improved early stopping criteria
4. More reliable model selection methods
The analysis may also include dropping / freezing layers to observe overall changes. (Line 151-155).
Line 223, “Since the computation of LS∗(A, B) is difficult, we consider its variant”, line 229 “Unfortunately, it is still difficult to solve LS0”: Contradicting statements. No proper justification for introducing LS0, as the sign function only reduces one logic step, in comparison to LS∗. Line 235-238, “If all points of MD(A, B) locate in one side of ωTm = 0, i.e., the two sets A, B are linearly separable, it holds that LS1(A, B) = 1. In contrast, if the value of LS1(A, B) is close to zero, the convex hulls of the two sets A, B overlap heavily. Because of the existence of absolute value operation, the solution of LS1 is difficult as well.” The problem of heavy overlap is introduced but not addressed. The low absolute and relative scores in Table 2, again bring out this issue.
Lastly, there is a writing error in line 261, “We achieves”.

**Questions:**

Posed in the Weakness section...

---

> ### Author Response · Authors · 2024-11-22
> **Reply to Reviewer # 7sGD**
>
> We are very grateful for your valuable comments. By carefully reading your comments, we find that there are several misunderstandings on some key aspects of this paper as follows:
>
> -   **[RE: “assumption that linear separability”]** Theorem 2.2 provides a sufficient and necessary condition that two sets are linearly separable, rather than builds some results based on the assumption that two sets are linear separable.
>
> -   **[RE: “non-linear relationships in complex data distributions”]** I think there is some misunderstanding on the main purpose of this paper. The reason of why this paper focuses on the linear separability is due to the working mechanism of the output layer of network-based classifiers for real-world binary classification tasks. In general, the output node of a neural network classifier is activated by using the nonlinear sigmoid or the tanh function, which actually corresponding to the question of how to find a separating hyperplane to divide the dataset into two parts. Because of the nonlinear mapping capability, many complicated real-world classification problems have been successfully solved by using the neural networks all of whose nodes are activated by using such a simple activation function. Unfortunately, it is still open to explain why the neural networks perform well for these complicated problems. This paper aims to provide feasible tool for filling this gap.
>
>     In the revised version, we have added new experiments to examine the effectiveness of our methods in different types of real-world classification problems. Please refer to **Appendix E** of the revised version.
>
> -   **[RE: “an absolute measure yet shifts towards an approximation of MD-LSM”]:**  I think there is a misunderstanding on the meanings of the proposed MD-LSMs $LS_0$, $LS_1$ and $LS_*$, and their quadratic version of $LS_2$. As shown in Table 1, the first three are absolute measures. Since it is hard to calculate them, we then introduce $LS_2$ to simplify their calculations. It is mainly used to obtain the approximate solution vector $\widehat{\omega}$ rather than to evaluate the linear separability. Therefore, there is no contradiction between an absolute measure and the relative approach mentioned in your comment.
>
> -   **[RE: “KL divergence”]:** The KL divergence has been widely used to measure the discrepancy between two distributions. To the best of our knowledge, it is still open on how to use the KL divergence to evaluate the linear separability degree of two point sets. The problem setup of tracking distributional shifts over time, mentioned in your comment, is totally different from the scenario of real-time monitoring the linear separability changes of hidden layer outputs during the process of training deep networks.
>
> -   **[RE: “the computational burden remains substantial”]:** As shown in Table 1, there are two way of evaluating the linear separability degree of dataset: one is to build a classifier, called "Probe\"; and the other is to calculate a mathematical term, such as the propsoed MD-LSM, GRQ and GDV. The former generally needs a training process.Their efficiency and the performance could be influenced by the choice of hyperparameters and termination condition, especially when there is no priori knowledge on the data distribution. In contrast, the calculation of MD-LSMs does not need a training process. The proposed approximate manner of calculating them has a high efficiency with only a slight precision sacrifice, and thus meets the technical requirements on the real-time monitoring of the behavior of each hidden layer after each training epoch. Please refer to **Table 2 in Section 2.4 and Algorithm 1 & 2 in Appendix C** for the comparison with the computational costs of different linear separability meausre.**
>
> -   **[RE: “error could massively increase with increasing layers”]:** The caluculation of MD-LSMs is independent of weight update, and thus has nothing to do with the network training process. Although there could be a slight discrepancy between their exact values and the approximate ones, this discrepancy cannot be transferred to other hidden layers. We have examined the effectiveness of the proposed methods for several popular deep networks on **the UCI datasets (including Diagnostic, Ionosphere, Maintenance and Marketing)** and **the text datasets (including Cora and IMDB)**. I think these empirical evidences are sufficient to support the validity of our results.
>
> -   **[RE: " synchronicity between training accuracy and MD-LSM"]:** The synchronicity exists between the training performance and the linear separability degree of hidden-layer outputs, where the linear separability degree is just measured by using the absolute measures ${\rm LS}_*$, ${\rm LS}_0$ and ${\rm LS}_1$, rather than their quadratic version ${\rm LS}_2$ that is a relative measure.
>
> **If these misunderstandings have been clarified, we hope you will reconsider the score of this paper.**

---

### Official Review · Reviewer_zkt6 · 2024-11-01

**Soundness:** 4
**Presentation:** 4
**Contribution:** 4
**Rating:** 8
**Confidence:** 4

**Summary:**

The paper submits a measure to understand the linear separability of hidden layers in deep neural network.

**Strengths:**

A measure for detecting the linear separability of hidden layer outputs.
The measure makes sense and is solidly done, with theoretical intuition and depth.
Good and detailed experimental sections.
Good Theoretical results.

**Weaknesses:**

While the paper is solidly written, I would suggest certaint improvements.

Please detail the notion of major side and minor. These two ideas later become really important in defining your metric. So, we need more intuition below Definition 2.4

prior to going to an approximate calculation of MD-LSM, I would suggest dicsussing the implications/intuition when a linear network of one layer is not used. In practice we use, a multi layer network, an intuition on how the ideas for one layer linear network would translate to this case are important and necessary.

I really think, that the discussion on the synchronicity phenomenon should be indicated somewhere in the front of the paper. Because, I went most of the paper trying to understand, why do I need to care about linear separability of hidden layers.

**Questions:**

The notion of multi-point sets, how is this translated to a data set in a standard ML literature.

Most ML methods involve a nonlinear function in the middle. how does the measure (relative or absolute) deal with that, what is the notion of separability in these contexts?

---

> ### Author Response · Authors · 2024-11-22
> **Reply to Reviewer # zkt6**
>
> We really appreciate your positive comments on this paper. The following are the detailed response to your comments
>
> **(1) [RE: “So, we need more intuition below Definition 2.4”]: ** Thanks. In the revised version, we have added the following sentences to give more intuition on the concepts of major and minor sides as well as the idea on how to obtain the the maximum linearly-separable subset:
>
>
> > "*\...\...According to Theorem 2.2, the points $m_{ij}\in minor_{{\omega}}(\mathrm{MD}({\cal A},{\cal B}))$ can be eliminated by removing the relevant points ${\bf a}_i$ from ${\cal A}$ or ${\bf b}_j$ from ${\cal B}$, and the rests turn out to be linearly separable.\...\...* \"
>
> **(2) [RE: “I would suggest discussing the implications /intuition when a linear network of one layer is not used”]:** Thanks. The proposed MD-LSM is a tool for objectively evaluating the linear separability degree of two sets, and thus it can be freely and independently used to detect the linear separability of arbitrary two sets. For a network with multiple hidden layers, we have calculated the MD-LSMs (including ${\rm LS}_*$, ${\rm LS}_0$, and ${\rm LS}_1$) of the output sets of each hidden layer, and the resultant MD-LSM values directly reflect the current linear separability degree of each hidden layer with the weights updated after the previous epoch. This is also the main purpose of providing the MD-LSMs tool for real-time monitoring the hidden-layer behavior.
>
> In the original version, because of the page limitation, we only illustrate the MD-LSM results of five hidden-layer networks and ten hidden-layer networks. In fact, the similar experimental phenomena have also appeared in the networks with other numbers of hidden layers. As a supplement, in **Table 8 of Appendix E.1**, we have added the experiments on the networks, the number of whose hidden layers ranges from $1$ to $5$, to illustrate the characteristics of hidden layers more clearly.
>
> **(3) [RE: “the discussion on the synchronicity phenomenon should be indicated somewhere in the front of the paper”]:** Thanks. In the revised version, we have moved the relevant content on synchronicity to Section 1.2.
>
> **(4) [RE: “The notion of multi-point sets, how is this translated to a data set in a standard ML literature.”]:** Thanks. The following replies are based on our understanding of this comment:
>
> > "Can the notion of multi-point sets considered in this paper be generalizable to the common scenarios of ML literature?\"
>
> If your opinion has been misunderstood, please point it out and we will correct it accordingly. The following is the reply to this comment.
>
> -   The answer to this question is \"Yes\", because all concepts proposed in this paper are built in the Euclidean space, where the datasets of most ML problems belong to. Since this paper mainly concerns with the classification tasks, we only take the multi-point sets as examples. In fact, the findings of this paper are suitable to various kinds of datasets, including real-world data, text data and graph data. In the revised version, we have added the experiments on some real-world classification tasks including **the text datasets (including Cora and IMDB) and the UCI datasets (including Diagnostic, Ionosphere, Maintenance, and Marketing).** Accordingly, the discussion on these new experimental results is also provided in **Table 2 and Appendix E.1, E.2, E.4** of the revised version. In addition, we have also considered the application of MD-LSMs on the graph neural network (GNN) (see Appendix E.4 of the revised version).
>
> **(5) [RE: Most ML methods involve a nonlinear function in the middle. How does the measure (relative or absolute) deal with that, what is the notion of separability in these contexts?]:** Thanks for the instructive comment. Linear separability measures (LSMs) aim to evaluate the linear separability degree of the two sets, this setting originally is to match with the common activation functions (such as sigmoid and tanh) which actually act as a hyperplane to separate the dataset. For the scenario mentioned in this comment, we can introduce the kernel trick into the definition of the proposed MD-LSM, and the notion of separability in this scenario can be converted into that of the linear separability in the feature space. This is an very interesting research issue, and we have added it into our future work list. In the conclusion section, we have added the following sentence:
>
> > "Since the high-dimensional vectors appearing in the expressions of MD-LSMs are of the inner-product form, we will introduce the kernel trick into them and then develop the tools of evaluating the degree of non-linear separability between two sets."

---

> > ### Comment · Reviewer_zkt6 · 2024-11-22
> > **Thank you for responding**
> >
> > The only thing, that is still not clear to me is "what is the meaning of multi-point sets" from the purview of general datasets. While, what you responded makes sense, it does not make clear to me the meaning of multi-point sets.

---

> > > ### Author Response · Authors · 2024-11-23
> > > **Respones to the common on “the notion of multi-point sets"**
> > >
> > > Upon the previous comment, we have found that the notion of "multi-point sets" should the one mentioned in the sentence of "In addition, we define the MD-LSMs for multiple point sets." (see line 295 of the revised version). I am so sorry that there is a typo in this sentence, i.e., the words "multiple point sets" should be "multiple-class sets". We have corrected it in the revised version. We thank you again for your careful reading.

---

### Official Review · Reviewer_VYjq · 2024-11-04

**Soundness:** 4
**Presentation:** 3
**Contribution:** 3
**Rating:** 8
**Confidence:** 3

**Summary:**

Large neural networks are difficult to analyze directly. Instead, the authors analyze the measure of linear separability of hidden layers. In the paper, a new measure for estimating linear separability is proposed, which satisfies two of the three requirements described by the authors. After that, the paper proposes an approximate extension that satisfies all the properties. The authors demonstrate the effectiveness of the proposed separability measures for classification problems in the image domain. The authors conduct an experiment on a wide class of image classification models. Also, the paper provides a deep mathematical analysis and comparison with other linear separability measures.

**Strengths:**

As the authors claim, the proposed measure and its extension can really improve monitoring of deep model training. Separately, I would like to note that the work presents a good mathematical justification for all conclusions, which allows for a much better analysis of the applicability limits of the proposed ideas, their limitations and greatly increases the significance of the work. Also, the authors provide experiments on a wide class of models, which demonstrates and emphasizes the universality of the proposed ideas. In general, the work has excellent quality and clarity of the formulated ideas and high originality of the proposed ideas and mathematical calculations, which have high theoretical value and have potentially good practical value.

**Weaknesses:**

Despite the large number of advantages voiced above. I would like to highlight some points that I would like to see in the work on this topic.

The experiments show monitoring of linear separability for a pair of classes from the CIFAR-10 dataset. In conclusion, the authors point out that experiments on other datasets were not conducted citing limitations on the volume of text and high computational costs. Nevertheless, I would like to see, albeit small, but experiments on data closer to real industry tasks. Also, the proposed measure is positioned as quite universal. However, if in the limitations the authors indicate that the work focuses only on the classification problem, then no discussions about the applicability of the proposed measure to other classification problems, for example, texts or graphs, are indicated in the article.

It is worth noting that the indicated shortcomings are not significant and are rather a suggestion for future work. A good theoretical basis for the proposed ideas gives confidence in the possibility of their generalization.

**Questions:**

Perhaps I missed the description of the restriction of the proposed ideas to the domain of images in text. How will the proposed measure behave for text classification problems, graph classification problems, and other classification problems?

---

> ### Author Response · Authors · 2024-11-22
> **Reply to Reviewer # VYjq**
>
> We really appreciate your valuable comments on this paper. The following are the detailed response to your comments.
>
> **[RE: “Nevertheless, I would like to see, albeit small, but experiments on data closer to real industry tasks.”]:** Thanks. In the revised version, we have added experimental results of MD-LSM on the UCI datasets, text classification and graph classification tasks to examine the effectiveness of the proposed MD-LSM on different types of data.
>
> **[RE: “no discussions about the applicability of the proposed measure to other classification problems, for example, texts or graphs, are indicated in the article.”]:** In the revised version, we have added the experiments on some real-world classification tasks including **the text datasets (including Cora and IMDB) and the UCI datasets (including Diagnostic, Ionosphere, Maintenance, and Marketing).** Accordingly, the discussion on these new experimental results is also provided in **Table 2 and Appendix E.1, E.2, E.4** of the revised version.
>
> **[RE: “How will the proposed measure behave for text classification problems, graph classification problems, and other classification problems?”]:**  In the revised version, we have added the experiments on some real-world classification tasks including **the text datasets (including Cora and IMDB) and the UCI datasets (including Diagnostic, Ionosphere, Maintenance, and Marketing).** In addition, we have also considered the application of MD-LSMs on the graph neural network (GNN) (see Appendix E.4 of the revised version).

---

> > ### Comment · Reviewer_VYjq · 2024-11-25
> >
> > Thanks for your comment. I have read the changes. You answered my main questions. Now the work fully corresponds to the rating of 8 (accept) in my opinion.

---

> > > ### Author Response · Authors · 2024-11-25
> > >
> > > Thank you very much for your support to our work!

---

### Official Review · Reviewer_N1VB · 2024-11-04

**Soundness:** 3
**Presentation:** 4
**Contribution:** 3
**Rating:** 6
**Confidence:** 3

**Summary:**

The newly introduced Minkowski Difference-based Linear Separability Measure (MD-LSM) is a tool designed to evaluate and understand the mechanisms of hidden layers in deep neural networks. This measure assesses the ability of a neural network to linearly separate two sets of data by analyzing their distribution through a hyperplane. Using the Minkowski difference, MD-LSM calculates the spatial relationship between data points from different classes, essentially determining if they can be linearly divided at a certain layer. By examining this separability, MD-LSM provides insights into how well the network’s hidden layers contribute to distinguishing between different classes, helping researchers and practitioners analyze the effectiveness of these layers in complex models. MD-LSM achieves this by solving two key factors: maintaining an absolute measure and being insensitive to outliers.

**Strengths:**

The strength of this paper lies in maintaining the efficiency and absoluteness in the linear separability of the hidden layers through the proposed method, MD-LSM. The measures LS*, LS0, and LS1 achieve efficiency scores, indicating that the model classifies all examples with straightforward accuracy, leaving no ambiguity regarding which points belong to which group. By emphasizing this exact efficiency, the paper enhances our understanding of how deep neural networks operate and underscores the effectiveness of these measures in quantifying linear separability.

**Weaknesses:**

1. High computational cost may make it difficult to use MD-LSM in large-scale or real-time applications, which could prevent it from being widely adopted for evaluating linear separability in deep neural networks.
2. In Table 2, high variance in results among LS*, LS0, and LS1, especially LS0's poor performance, shows that these models lack reliability and effectiveness in handling highly overlapping data in classification tasks. The high variability of LS2 within the MD-LSM values demonstrates its sensitivity to the analyzed data. This sensitivity can result in instability, complicating the interpretation of results across different scenarios. Furthermore, the LS2 model is prone to overfitting, which can lead to biased outcomes when applied to new data.
3. The evaluation of the proposed theory is incomplete, as factors such as increasing network size and utilizing state-of-the-art activation functions must be considered during the assessment of linear separability in hidden layers; an evaluation with a single dataset is insufficient, and larger datasets like ImageNet should be included for a more comprehensive analysis.

**Questions:**

see above in weakness section

---

> ### Author Response · Authors · 2024-11-22
> **Response to Reviewer # N1VB**
>
> Thanks for your valuable comments. The following are the detailed replies to all the comments you made. **If we have clarified your confusion on this paper, please reconsider the score of this paper.**
>
> **(1) [RE: “High computational cost”]:** Thanks. In Table 1 of the original version, we have listed the existing methods for evaluating the linear separability degree of dataset. According to the way they are computed, these methods can be divided into two categories: one is to build a classifier, called "Probe\"; and the other is to find mathematical terms calculated by using the data points in the dataset, such as GRQ, GDV, linear divisible angle, and smallest thickness which are listed in the table.
>
> The former generally needs a training process, and thus their efficiency and performance could be influenced by the choices of hyperparameters or the setting of termination condition, especially when there is no priori knowledge on the data distribution. That is reason of why we think the "Probe\" method based on the state-of-the-art classifiers could not \"efficiently\" handle the real-time monitoring of hidden-layer behavior of deep network during its training process.
>
> In contrast, the calculation of the mathematical terms (such as GRQ, GDV, linear divisible angle, and smallest thickness) does not need a training process. However, it is still challenging to develop an feasible linear separability measure that can be efficiently calculated especially when the data size is large. For example, as shown in Algorithm 1 and Algorithm 2 of Appendix C, the computational costs of GDQ and GDV are both higher than the approximate manner of calculating the proposed MD-LSM. **We also refer to Table 2 of the revised version for the comparison among the time costs of different LSMs on the UCI datasets.**
>
> The proposed MD-LSM is still far from being truly efficient one, but it is the best way we know of so far. Compared with the existing LSMs listed in Table 1, it has been able to meet the current technical requirements on the real-time monitoring of the behavior of each hidden layer after each training epoch.
>
> **In the revised version, we have added the following sentences to highlight this facts:**
>
> *“…\...Some state-of-the-art classifiers (such as logistic regression or naive Bayes) have the potential to act as feasible “probes” because of their low desired computational complexities. However, if there is no priori knowledge on the data distribution, the efficiency and the performance of these classifiers could be heavily influenced by some unavoidable factors such as the choice of hyperparameters and the setting of termination conditions, and thus their desired complexities are usually hard to be achieved in practice. Consequently, the “probe” method is unsuitable (at least cannot be directly applied) to detecting the mapping behavior of each hidden layer after each training epoch.\...\...”*
>
> **(2) [RE: “high variance in results”]:** Thanks. I think there are some misunderstanding for the results shown in Table 2:
>
> -   The statement \"especially LS0's poor performance\" is not true. As addressed in Line 227-228, it is true that $major_{\omega_*}({MD}(A,B)) = major_{\omega_0}({MD}(A,B))$, which means that ${\rm LS}_*$ and ${\rm LS}_0$ are exactly equivalent.
>
> -   Although there are some difference among the separating lines associated with $LS_*$, $LS_0$, and $LS_1$, it is because that they are derived from different optimization problems.
>
> -   It is noteworthy that $LS_2$ is not treated as a linear separability measure in this paper. In fact, $LS_2$ is actually a quadratic version of the proposed MD-LSMs (including $LS_{*}$, $LS_0$, and $LS_1$), and is used to provide a feasible way of approximately solving the optimization problems associated with them.
>
> **(3) [RE: “The evaluation of the proposed theory is incomplete”]:** Due to device limitation, it is difficult for us to implement the real-time monitoring on large-scale datasets. However, in the revised version, we have added new experiments on the UCI datasets (including Diagnostic, Ionosphere, Maintenance, and Marketing) and the text datasets (including Cora and IMDB) to examine the effectiveness of the proposed MD-LSMs and meanwhile to explore the hidden-layer behaviors of deep networks on different types of classification tasks. Please refer to Table 2, Appendix E.1, E.2 and E.4 of the revised version for the new experimental results

---

### Meta-Review · Area_Chair_fQZe · 2024-12-21

**Metareview:**

The paper proposes a method, denoted as  Minkowski Difference-based Linear Separability Measure (MD-LSM)
for measuring  linear separability of hidden layers in deep neural networks.
Reviews are  of mix feelings and points to some  strengths and weaknesses.

Strengths include a  theoretical foundation and a nice presentation, with extensive
experimental validation. However, weaknesses have been pointed regarding
the method's efficiency. Also, some reviewers have found the experimental analysis incomplete, particularly concerning
real-world applications and large-scale datasets.
However, the main point about this paper is that there is a strong concern about appropriateness of
comparisons with logistic regression, and concerns about  method's reliance on
approximations cast doubts on its theoretical soundness.

Based on this latter point to which I concur, the reviews, the rebuttal and my own reading of the paper,
I can not recommend acceptance and suggest the authors to address the raised issues before resubmission
to another venue.

**Additional Comments On Reviewer Discussion:**

There was a long discussion about the above point. While I agree with the authors about importance/relevance of results
provided in the rebuttal, my own understandings of the paper make me agree with reviewer that comparisons with
linear probes may be necessary and that bridging the gap between the theory and the proposed algorithm is necessary.

---

### Decision · Program_Chairs · 2025-01-22

Reject